# Learning No-Regret Sparse Generalized Linear Models with Varying Observation(s)

**Diyang Li[1], Charles X. Ling[2], Zhiqiang Xu[3], Huan Xiong[3] & Bin Gu[3,∗]**
[1]Cornell University
[2]Western University
[3]Mohamed bin Zayed University of Artificial Intelligence
`diyang01@cs.cornell.edu, charles.ling@uwo.ca`
`{zhiqiang.xu,huan.xiong,bin.gu}@mbzuai.ac.ae`

## Abstract

Generalized Linear Models (GLMs) encompass a wide array of regression and classification models, where prediction is a function of a linear combination of the input variables. Often in real-world scenarios, a number of observations would be added into or removed from the existing training dataset, necessitating the development of learning systems that can efficiently train optimal models with varying observations in an online (sequential) manner instead of retraining from scratch. Despite the significance of data-varying scenarios, most existing approaches to sparse GLMs concentrate on offline batch updates, leaving online solutions largely underexplored. In this work, we present the first algorithm without compromising accuracy for GLMs regularized by sparsity-enforcing penalties trained on varying observations. Our methodology is capable of handling the addition and deletion of observations simultaneously, while adaptively updating data-dependent regularization parameters to ensure the best statistical performance. Specifically, we recast sparse GLMs as a bilevel optimization objective upon varying observations and characterize it as an explicit gradient flow in the underlying space for the inner and outer subproblems we are optimizing over, respectively. We further derive a set of rules to ensure a proper transition at regions of non-smoothness, and establish the guarantees of theoretical consistency and finite convergence. Encouraging results are exhibited on real-world benchmarks.

## 1 Introduction

**Sparse GLMs**  Generalized Linear Models (GLMs) (Nelder & Wedderburn, 1972; McCullagh & Nelder, 2019; Massias et al., 2020) constitute a comprehensive extension of ordinary least squares, profoundly impacting the communities of machine learning (Kumar et al., 2015; Emami et al., 2020; Kulkarni et al., 2021), data mining (Zhang et al., 2016) and computer vision (Kim et al., 2014). These models have been extensively utilized as statistical estimators for diverse regression and classification tasks, wherein the output is postulated to adhere to an exponential family distribution whose mean is a linear combination of the input variables. Renowned instances of GLMs consist of logistic regression (Wright, 1995), Poisson regression (Frome, 1983), Gamma regression (Prentice, 1974), and proportional odds model (Bennett, 1983). In the realm of high-dimensional learning, sparsity-inducing regularizations have emerged as powerful tools with substantial theoretical underpinnings (Tibshirani, 1996; Bach et al., 2011; 2012), facilitating the concurrent execution of feature selection and model prediction. By substantially diminishing the quantity of active predictors (variables) involved, sparsity-inducing penalization yields interpretable models and expedite computational efficiency during the prediction phase.

**GLMs with Varying Observations**  Nowadays, a vast majority of existing algorithms designed for sparse GLMs have predominantly been trained offline in batch-mode (Fercoq & Richtárik, 2015; Karimireddy et al., 2019), which necessitates the availability of all training observations at the onset

---

∗Corresponding author.

of the learning process. However, such a prerequisite may not always be realistic or applicable, particularly in real-time scenarios or large-scale problems. Indeed, in numerous real-world machine learning applications, *e.g.*, edge or cloud computing (Pathak et al., 2018), dynamic pricing (Levina et al., 2009), and fraud detection (Dhieb et al., 2020), the observations (samples) of input can dynamically change, thereby rendering it computationally impractical to train over the entire dataset multiple times. Contrastingly, the paradigm of data-varying (also referred to as incremental and decremental) algorithms has proven its efficiency and computational feasibility across various contexts (Burkov, 2020). These methods enable models to incrementally update their knowledge without necessitating complete re-training with *each data alteration* (also referred to as one online round), such as the addition or removal of observation(s) from the training set. Consequently, algorithms for varying data have garnered increasing attention from both academic and industrial research communities in recent years (Hoi et al., 2021; Zhou, 2022).

**State-of-the-art**  The literature on data-varying algorithms has grown significantly, aiming to achieve comparable accuracy to batch re-training without incurring regret (*i.e.*, the regret is zero, or no-regret). However, these works are specifically tailored to certain learning models and are not readily extensible to other loss functions and regularizers in sparse GLMs[1]. The contributions to this domain encompass online $k$-NN (Rodríguez et al., 2002), logistic regression (Tsai et al., 2014), (group) Lasso (Garrigues & Ghaoui, 2008; Chen & Hero, 2012; Li & Gu, 2022; Hofleitner et al., 2013), and a plethora of Support Vector Machines (SVMs) (Laskov et al., 2006; Gu et al., 2014; 2015; 2018; Kashef, 2021). It is pertinent to highlight that nearly all of the aforementioned algorithms are exclusively capable of adjusting the model weights (*i.e.*, learnable parameters) and lack the ability to dynamically update predefined hyperparameters like regularization, during successive online rounds.

**Technical Challenges**  In this paper, we aim to present a sophisticated no-regret learning framework for varying observations accompanied by a pertinent algorithm, which not only yields identical solutions to batch retraining algorithms but also facilitate model selection by dynamically selecting data-dependent regularization parameter(s) at each round, ensuring the optimal statistical performance of the estimator. From data perspective, the proposed methodology accommodates simultaneous addition and deletion of observations while efficiently processing multiple data points in a single execution (*i.e.*, chunk updating), as opposed to many single-point algorithms (*e.g.*, Garrigues & Ghaoui (2008); Gu et al. (2018); Yang et al. (2010)). However, the devising of innovative and high-quality data-varying algorithms remains an arduous task. Firstly, these algorithms often stem from a standard learning approach (Tsai et al., 2014), which may result in exceedingly intricate procedures that are challenging to manage. Secondly, the design of data-varying learning algorithms may diverge considerably depending on the concrete loss functions or regularizers (Hoi et al., 2021), rendering the establishment of a universally applicable framework technically demanding. As a result, the creation of an efficient algorithm for sparse GLMs with varying observations continues to be a highly sought-after yet formidable task for researchers (Burkov, 2020; Luo & Song, 2020).

**Proposed Method**  To work around these bottlenecks, we commence by reparameterizing the data-varying learning task, which entails transforming one single online round into a bilevel optimization framework. Within this refined construct, we characterize its dynamics as an explicit gradient flow in the underlying space we are optimizing over. Consequently, the inner problem is proficiently modeled and tackled through the utilization of a system of Partial Differential Equations (PDE), whereas the outer problem capitalizes on an Ordinary Differential Equations (ODE) system for its effective numerical resolution. Recognizing the inherent sparsity property, we leverage set control methodologies that circumvent non-differentiable regions and establish threshold conditions for the active set inclusion. Such an approach facilitates the training of a data-varying S̲p̲A̲rse G̲LMs in an O̲nline manner, namely SAGO algorithm. Our comprehensive empirical evaluations substantiate the superior efficiency and accuracy of SAGO in comparison to prevalent batch re-training techniques.

**Our Contributions**  The main contributions brought by this paper are delineated as follows.

- We present a sophisticated framework and propose a cutting-edge algorithm for a more general form of sparse GLMs, which offers valuable insights into the dynamics of data-varying paradigm. Under mild assumptions, our analysis demonstrates that SAGO converges to the stationarity point on new set and attains convergence within a finite number of steps.

---

[1]In Appendix C.1, we discuss why existing methods cannot be readily extended to GLMs and delineate the connections between our study and conventional online learning with bounded regret.

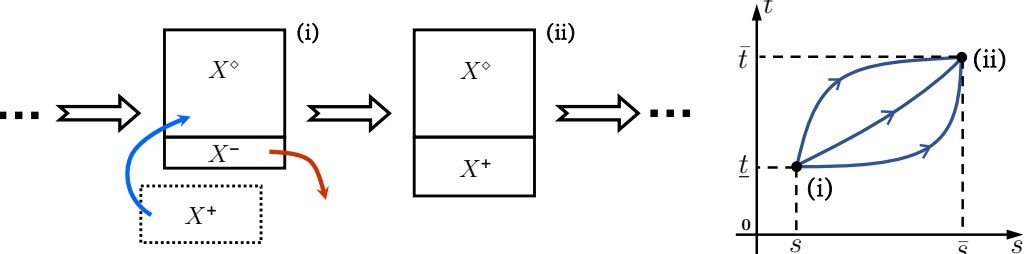

Figure 1: Depiction of the alterations in the training set during online rounds. A single black arrow signifies one learning round, while red and blue arrows indicate the (incoming) operations conducted on data. The status (i), (ii) represent various model conditions (*i.e.*, trained on different dataset).

Figure 2: Distinct blue directional lines represent varied paths from (i) to (ii). The solutions of (2) at $[\underline{s}, \underline{t}]$ (*resp.* $[\bar{s}, \bar{t}]$) is associated with the status (i) (*resp.* (ii)) in Fig. 1.

- Contrasting with existing frameworks for data-varying sparse GLMs, SAGO enables the incorporation and removal of multiple observations in a single execution without sacrificing accuracy, while concurrently updating the regularization parameter(s) in each round.

- We elucidate how SAGO paves the way to no-regret online training for multiple well-known GLMs, including Poisson (Frome, 1983) and gamma regression (Prentice, 1974). To the best of our knowledge, this constitutes the first exact data-varying algorithm for these models.

## 2 PRELIMINARIES

**Basic Notation**    For any integer $d \in \mathbb{N}$, we denote by $[d]$ the set $\{1, \ldots, d\}$. The vector of size $n$ with all entries equal to $0$ (*resp.* 1) is denoted by $\mathbf{0}_n$ (*resp.* $\mathbf{1}_n$). The design matrix $X \in \mathbb{R}^{n \times m}$ is composed of observations $\mathbf{x}_i^\top \in \mathbb{R}^m$ stored row-wise; the vector $y \in \mathbb{R}^n$ (*resp.* $\{-1, 1\}^n$) signifies the response vector for regression (*resp.* binary classification). For a model weights vector $\beta$ and a subset $\mathcal{A} \subset [m]$, $\beta_{\mathcal{A}}$ and $X_{\mathcal{A}}$ are $\beta$ and $X$ restricted to features in $\mathcal{A}$. Subscripts (*e.g.*, $\beta_j$) denote vector entries or matrix columns. The standard Euclidean norm for vectors reads $\| \cdot \|$; the $\ell_1$-norm is symbolized by $\| \cdot \|_1$. We provide a comprehensive compilation of all notations in Appendix A.

### 2.1 FORMALIZED SETTING

Considering each online round, the data is structured (partitioned) as

$$X^\diamond = \{\{\mathbf{x}_1, y_1\}, \ldots, \{\mathbf{x}_{n_0}, y_{n_0}\}\}, \quad X^- = \{\{\mathbf{x}_{n_0+1}, y_{n_0+1}\}, \ldots, \{\mathbf{x}_{n_0+n_-}, y_{n_0+n_-}\}\},$$
$$X^+ = \{\{\mathbf{x}_i, y_i\} \mid \forall i \in [n_0 + n_- + 1, \ n_0 + n_- + n_+]\},$$

where $X^\diamond$ indicates the set of observations to be retained (*i.e.*, unchanged in a single online round), $X^-$ (*resp.* $X^+$) represents the set of observations to be eliminated (*resp.* incorporated). Figure 1 illustrates the schematic diagram of the data-varying setup. Let us assume that we presently possess $n_0 + n_-$ labelled instances $X^{(i)} = \{X^\diamond \cup X^-\}$ and an optimal machine learning model trained using $X^{(i)}$, in which the parameter is denoted by $\hat{\beta}^{(i)} \in \mathbb{R}^m$ (*already known*). In certain practical contexts, it may be desirable to incorporate new samples $X^+$ into $X^{(i)}$, while concurrently discarding selected samples $X^-$ from $X^{(i)}$. This implies the requirement for an optimal learning model that trained on $X^{(ii)} = \{X^\diamond \cup X^+\}$, and the learnable parameter of this optimal model on the new (changed) dataset is signified as $\hat{\beta}^{(ii)} \in \mathbb{R}^m$, which is *currently unknown*. In essence, the resultant sample size should be $n_0 + (n_+ - n_-)$ following this online update. Methodologies in conventional paradigm would only involve re-training a new model on $\{X^\diamond \cup X^+\}$. However, our proposed approach SAGO seeks to compute $\hat{\beta}^{(ii)}$ directly, meaning it bypasses the need for re-training from the ground up, and instead leverages the information encapsulated within initial $\hat{\beta}^{(i)}$.

## 2.2 OFFLINE SPARSE GLMS

To commence, we introduce the class of optimization problems under consideration.

**Definition 1** (Sparse GLMs). *We call sparse generalized linear models the following problem*

$$\min_{\beta \in \mathbb{R}^m} \sum_i \ell_i \left( y_i, f \left( \beta^\top \mathbf{x}_i \right) \right) + \sum_{k=1}^d \sum_{j \in \mathcal{S}_k} P_\eta \left( |\beta_j|, \lambda_k \right), \tag{1}$$

*where each $\ell_i(\cdot)$ represents a piecewise (or elementwise) twice-differentiable convex loss [2] and $d$ is the number of regularizers. We have $\mathcal{S} = \{\mathcal{S}_1, \mathcal{S}_2, ..., \mathcal{S}_k, \mathcal{S}_{k+1}, ..., \mathcal{S}_d\}$, where $\mathcal{S}_k \subset [m]$ denotes the subset of coefficients being penalized ($\mathcal{S}$ for Shrink). $P_\eta \left( |\beta_j|, \lambda_k \right)$ is a scalar penalty function (regularizer), with $\eta$ representing potential parameter(s) for a penalty family and $\lambda_k \in \mathbb{R}^{\geq 0}$ being a non-negative hyperparameter.*

We define $\lambda = [\lambda_1, ..., \lambda_d]^\top$, controlling the trade-off between data fidelity and regularization in (1). To achieve model selection, we typically employ $K$-fold cross-validation. In its simplest form, cross-validation involves the use of a second, independent set of data $\{\tilde{\mathbf{x}}_i, \tilde{y}_i\}_{i=1}^N$ (commonly referred to as a **validation set** (Shao, 1993)), which serves to evaluate model performance and select appropriate regularization parameters. Additionally, we impose several mild conditions on $P_\eta \left( |\beta_j|, \lambda_k \right)$.

**Assumption 1.** *Let $P_\eta(\rho, \xi)$ be a shorthand notation for $P_\eta \left( |\beta_j|, \lambda_k \right)$. We assume $P_\eta(\rho, \xi)$ satisfies the following regularity conditions: (i) symmetric about 0 in $\rho$, (ii) $P_\eta(0, \xi) > -\infty$, for $\forall \xi \geq 0$, and $\xi$ is separable as a multiplier from $P_\eta(\rho, \xi)$, (iii) monotone increasing in $\xi \geq 0$ for any fixed $\rho$, (iv) non-decreasing in $\rho \geq 0$ for any fixed $\xi$, (v) the first two derivatives w.r.t. $\rho$ exist and are finite.*

Most commonly used penalties satisfy the aforementioned Assumption 1, including but not limited to the power family $P_\eta(|\beta|, \lambda_k) = \lambda_k |\beta|^\eta, \eta \in (0, 2]$ (Frank & Friedman, 1993), log penalty $P_\eta(|\beta|, \lambda_k) = \lambda_k \ln(\eta + |\beta|), \eta > 0$ (Armagan et al., 2013), along with the non-convex SCAD regularizer (Fan & Li, 2001) and MC+ penalty (Zhang, 2010), *etc.*

## 2.3 FROM OFFLINE TO ONLINE

Given a GLM as described in (1), we present a framework for learning it with varying observations.

**Definition 2** (Parameterizer). *The parameterizers $\mu(t), \nu(s) \in \mathbb{R} \backslash \{\infty\}$ must satisfy the following conditions: (i) The inverse of parameterizers, denoted as $\mu^{-1}$ or $\nu^{-1}$, exists and should be continuous. (ii) The first-order derivative of $\mu(t)$ (resp. $\nu(s)$) w.r.t. $t$ (resp. $s$) is smooth in the closed interval $[\underline{t}, \bar{t}]$ (resp. $[\underline{s}, \bar{s}]$), where $\underline{t} = \mu^{-1}(0)$, $\bar{t} = \mu^{-1}(1)$, $\underline{s} = \nu^{-1}(0)$ and $\bar{s} = \nu^{-1}(1)$. (iii) The reparameterized objective retains its original (dis)continuity.*

The parameterizer in Definition 2 can be employed to recast the original data-varying learning, which could be implemented in two manners, *i.e.*, *function-wise* or *observation-wise*. Mathematically, let $\blacklozenge$ denote either $\mu(t)$ or $\nu(s)$, the corresponding reparameterization could take the form of $\blacklozenge \cdot \ell_i \left( y_i, f \left( \beta^\top \mathbf{x}_i \right) \right)$ or $\ell_i \left( \blacklozenge \cdot y_i, f \left( \blacklozenge \cdot \beta^\top \mathbf{x}_i \right) \right)$. For simplicity, we denote $\ell^\diamond(\beta) := \sum_{\{i | \{\mathbf{x}_i, y_i\} \in X^\diamond\}} \ell_i \left( y_i, f \left( \beta^\top \mathbf{x}_i \right) \right)$, $\ell^+(\beta) := \sum_{\{i | \{\mathbf{x}_i, y_i\} \in X^+\}} \ell_i \left( y_i, f \left( \beta^\top \mathbf{x}_i \right) \right)$ and similarly, $\ell^-(\beta)$. Then we introduce the following class of function-wise reparameterized optimization problem

$$\min_{\lambda \in \Lambda} \mathcal{L}_{val} := \sum_{i=1}^N \ell_i \left( \tilde{y}_i, f \left( \hat{\beta}^\top \tilde{\mathbf{x}}_i \right) \right)$$

$$s.t. \quad \hat{\beta} \in \arg \min_{\beta \in \mathbb{R}^m} \ell^\diamond(\beta) + \mu(t) \ell^+(\beta) + \nu(s) \ell^-(\beta) + \sum_{k=1}^d \sum_{j \in \mathcal{S}_k} P_\eta \left( |\beta_j|, \lambda_k \right)$$

$$\underline{t} \leqslant t \leqslant \bar{t}, \quad \underline{s} \leqslant s \leqslant \bar{s}. \tag{2}$$

The (2) characterizes a bilevel optimization problem, in which the inner-level problem addresses the changes of varying data, while the outer-level problem exploits validation set to determine the regularization parameter(s) in each online round. Here, $s$ and $t$ serve as variables within the reparameterization (*i.e.*, parameterizer incorporating). At the onset of each round, we initialize $s$

---

[2]Our choice of terminology may be considered an abuse of language since for some choice of $\ell_i$'s, *e.g.*, an Huber loss, there is no underlying statistical GLMs.

and $t$ as $\underline{s}$ and $\underline{t}$, respectively. To deduce an optimal solution subsequent to one round of online data changes, we could progressively increase $s$ and $t$, while concurrently computing (tracking) the dynamics on $\hat{\beta}$ in association with $(s, t)$, until they attain the upper bound of $s = \bar{s}$ and $t = \bar{t}$.

**Remark 1.** *A naive example is $\mu(t) \stackrel{def}{=} t$, $\nu(s) \stackrel{def}{=} 1 - s$. Then $(s, t)$ transitions from $(0, 0)$ to $(1, 1)$. Intuitively, $\mu$ parameterizes an increment and $\nu$ parameterizes a decrement.*

**Theorem 1** (Equivalence Guarantee (informal)). *Let $\mathbf{A}$ be a batch algorithm that trains on $X$ and outputs a model $f \in \mathcal{H}$, i.e., $\mathbf{A}(X) : X \to \mathcal{H}$. The symbol $\leftarrow$ signifies online updating as per (2). Suppose we desire to incorporate $X^+$ and remove $X^-$, from the existing model. Then $\forall \mathcal{T} \subseteq \mathcal{H}$,*

$$P\left\{\left(\mathbf{A}(X^\diamond \cup X^-) \xleftarrow[Online]{Adding} X^+ \xleftarrow[Online]{Deleting} X^-\right) \in \mathcal{T}\right\} = P\left\{\mathbf{A}(X^\diamond \cup X^+) \in \mathcal{T}\right\}.$$

Formal statements and proofs are in Appendix B.1. The underlying intuition of Theorem 1 is that the output of SAGO for (2) is indeed equivalent to the solution that would be obtained if trained from scratch using $\{X^\diamond \cup X^+\}$, and the regret of solution produced by our SAGO algorithm at $[\bar{s}, \bar{t}]$, is provably zero after each online round (*a.k.a.*, no-regret). This equivalence establishes a bridge that links two disparate model statuses (*c.f.* Figure 2). By applying the aforementioned procedure, we are empowered to train in an online manner when the set of observations is not fixed, further affirming that the first-order optimal solution can be achieved, akin to the capabilities of batch algorithms.

## 3 ELABORATED SAGO

In this section, we introduce an innovative approach, employing the PDE and ODE systems, referred to as the PDE-ODE procedure, to trace the trajectory of $\left(\hat{\beta}(s, t), \mathcal{L}_{val}(\lambda)\right)$, or $(\hat{\beta}, \mathcal{L}_{val})$ for short, until we reach the terminal points $[\bar{s}, \bar{t}]$ and $\lambda_{\max}$. Set $\mathcal{S}_* = \bigcup_{k=1}^d \mathcal{S}_k$. For a parameter vector $\beta \in \mathbb{R}^m$, we use $\mathcal{S}_{\mathcal{Z}}(\beta) = \{j \in \mathcal{S}_* \mid \beta_j = 0\}$ to represent the set of penalized parameters that are zero and correspondingly, $\mathcal{S}_{\bar{\mathcal{Z}}}(\beta) = \{j \in \mathcal{S}_* \mid \beta_j \neq 0\}$ demarcates the set of penalized parameters with non-zero values. In light of the fact that typically only a fraction of the coefficients are non-zero in the optimal solution, a prevalent strategy to speed up solver efficiency is to narrow down the optimization problem's magnitude by excluding features that do not contribute to the solution $\beta$. We define the **active set** $\mathcal{A} = \mathcal{S}_{\bar{\mathcal{Z}}} \cup \overline{\mathcal{S}_*}$ indexes the current active predictors having either unpenalized or non-zero penalized coefficients, with cardinality $\|\beta\|_0$.

**Assumption 2.** *We presuppose that the first-order optimal solutions of (1) and (2) can be accomplished (via batch algorithms), and the active set $\mathcal{A}$ is non-empty for $\forall t \in [\underline{t}, \bar{t}]$, $\forall s \in [\underline{s}, \bar{s}]$, $\forall \lambda \in \Lambda$.*

**Theorem 2** (Continuity Guarantee). *Assume that there exists a positive constant $M$ such that the inequalities $\nabla_\lambda^2 \hat{\beta} \preceq MI$, $\partial_s^2 \hat{\beta} \preceq M\mathbf{1}_m$, $\partial_t^2 \hat{\beta} \preceq M\mathbf{1}_m$, $\widetilde{\mathcal{L}}^{-1} \preceq MI$ holds universally for all $\hat{\beta}$, where $\widetilde{\mathcal{L}}$ will be defined in (3). Then provided a fixed active set $\mathcal{A}$, the solution $\hat{\beta}$ demonstrates continuity w.r.t. the variables of the inner problem and the variable(s) of the outer problem, respectively.*

### 3.1 INNER PROBLEM SOLVING

We define $\ell_{i,\mathcal{A}} = \ell_i\left(y_i, f\left(\beta_{\mathcal{A}}^\top \mathbf{x}_{i,\mathcal{A}}\right)\right)$. Theorem 3 aids in tracking the flow to inner-level soluions.

**Theorem 3** (Inner-level Dynamics). *Suppose $\hat{\beta}$ is the solution to the inner problem of (2) for given $s$ and $t$, and the collection $\mathcal{A}$ is fixed, the dynamics of gradient flow on $\hat{\beta}_{\mathcal{A}}(s, t)$ comply with*

$$\begin{cases} \dfrac{d\mu(t)}{dt} \cdot \nabla_{\beta_{\mathcal{A}}} \ell_{\mathcal{A}}^+(\hat{\beta}) + \widetilde{\mathcal{L}}\left(\hat{\beta}, \lambda\right) \cdot \dfrac{\partial \hat{\beta}_{\mathcal{A}}(s, t)}{\partial t} = \mathbf{0} \\[2ex] \dfrac{d\nu(s)}{ds} \cdot \nabla_{\beta_{\mathcal{A}}} \ell_{\mathcal{A}}^-(\hat{\beta}) + \widetilde{\mathcal{L}}\left(\hat{\beta}, \lambda\right) \cdot \dfrac{\partial \hat{\beta}_{\mathcal{A}}(s, t)}{\partial s} = \mathbf{0} \\[2ex] \widetilde{\mathcal{L}}\left(\hat{\beta}, \lambda\right) = \sum_{k=1}^d \left[\nabla_{\beta_{\mathcal{A}}}^2 \sum_{j \in \mathcal{S}_{k,\mathcal{A}}} P_\eta\left(\left|\hat{\beta}_j\right|, \lambda_k\right)\right] + \nabla_{\beta_{\mathcal{A}}}^2 \left[\ell_{\mathcal{A}}^\diamond(\hat{\beta}) + \mu(t) \ell_{\mathcal{A}}^+(\hat{\beta}) + \nu(s) \ell_{\mathcal{A}}^-(\hat{\beta})\right] \\[2ex] \hat{\beta}(s, t)\Big|_{s=\underline{s}, t=\underline{t}} = \hat{\beta}^{(i)}. \end{cases} \quad (3)$$

The (3) represents a first-order homogeneous PDE system. By employing our proposed procedure, the resolution of the inner problem transforms into a standard form initial-value problem (IVP) (Fatunla,

2014) in the context of numerical PDE. The solving to (3) enables tracking of the entire spectrum of solutions for $\left\{ \hat{\beta}\left(s,t\right)|(s,t)\in[\underline{s},\bar{s}]\times[\underline{t},\bar{t}]\right\}$. As our primary interest lies in the specific solutions when $s=\bar{s}$ and $t=\bar{t}$, we can select an integral path $\Omega(s,t)=0$ connecting $[\underline{s},\underline{t}]$ and $[\bar{s},\bar{t}]$, and then perform a line integral along $\Omega$ (*c.f.* Figure 2) to attain $\hat{\beta}(\bar{s},\bar{t})$. During the integration, careful management of the set of non-zero coefficients according to the thresholding conditions enables the application of our algorithm within the lower-dimensional subspace spanned solely by active coefficients. We emphasize that $\hat{\beta}$ is piecewise smooth as a function of $s$ and $t$, *i.e.*, there exists $\underline{t}=t_0<t_1<t_2<\cdots<t_{k_{max}}=\bar{t}$ and $\underline{s}=s_0<s_1<s_2<\cdots<s_{k_{max}}=\bar{s}$, such that the solution $\hat{\beta}\left(s,t\right)$ can be precisely represented via

$$\hat{\beta}\left(s_{k-1},t_{k-1}\right)-\int_\Omega \widetilde{\mathcal{L}}(\hat{\beta},\lambda)^{-1}\cdot\left\langle\left[\frac{d\nu(s)}{ds}\nabla_{\beta_\mathcal{A}}\ell_\mathcal{A}^\diamond(\hat{\beta}),\frac{d\mu(t)}{dt}\nabla_{\beta_\mathcal{A}}\ell_\mathcal{A}^+(\hat{\beta})\right],\left[ds,dt\right]\right\rangle, \qquad (4)$$

where $\forall s\in[s_{k-1},s_k],\forall t\in[t_{k-1},t_k]$. The $s_k$ and $t_k$ essentially represent event points and will be formally introduced in Section 3.3.

## 3.2 OUTER PROBLEM SOLVING

In each round of online phase, Theorem 4 facilitates the approximation to $\lambda^*$ for the outer problem.

**Theorem 4** (Outer-level Dynamics). *Assume that the set $\mathcal{A}$ is fixed. Under varying $\lambda$ values in the outer problem, the dynamics of gradient flow on $\hat{\beta}_\mathcal{A}(\lambda)$ in the outer $\mathcal{L}_{val}$ can be expressed as*

$$\frac{d\hat{\beta}_\mathcal{A}}{d\lambda}=-\sum_{k=1}^d\left[\nabla_{\beta_\mathcal{A}}^2\left(\ell_\mathcal{A}^\diamond(\hat{\beta})+\ell_\mathcal{A}^+(\hat{\beta})\right)+\sum_{k=1}^d\left(\nabla_{\beta_\mathcal{A}}^2\sum_{j\in\mathcal{S}_{k,\mathcal{A}}}P_\eta\left(\left|\hat{\beta}_j\right|,\lambda_k\right)\right)\right]^{-1}\cdot\nabla_{\beta_\mathcal{A}}\sum_{j\in\mathcal{S}_{k,\mathcal{A}}}P_\eta\left(\left|\hat{\beta}_j\right|,\lambda_k\right).$$
$$(5)$$

In analogy with the inner problem, $\hat{\beta}$ exhibits piecewise smoothness as a function of $\lambda$. Consequently, it is possible to numerically solve the IVP for the first-order ODE system (5). The Picard–Lindelöf theorem (Siegmund et al., 2016) unequivocally establishes the uniqueness of solutions,[3] facilitating the calculation of $\{\nabla_\lambda\mathcal{L}_{val}\left(\lambda\right)|\lambda\in\Lambda\}$, given that $\nabla\mathcal{L}_{val}$ is a direct function of $\hat{\beta}\left(\lambda\right)$, represented as

$$\nabla_\lambda\mathcal{L}_{val}=\sum_{i=1}^N\nabla_\beta\ell_i\left(\tilde{y}_i,f\left(\hat{\beta}^\top\tilde{\mathbf{x}}_i\right)\right)\cdot\frac{d\hat{\beta}}{d\lambda}, \qquad (6)$$

where $d\hat{\beta}/d\lambda$ could be computed via (5). To identify the optimal regularization parameter(s), one can evaluate the condition $\nabla_\lambda\mathcal{L}_{val}\stackrel{?}{=}\mathbf{0}$. In practical scenarios, an alternate approach is to examine the hyper-gradient, accepting $\|\nabla_\lambda\mathcal{L}_{val}\|\leqslant\epsilon$ as a loose condition (relaxation) to approximately select the regularization parameter(s), where $\epsilon\in\mathbb{R}^{\geq0}$ is an user-defined parameter.

**Theorem 5** (Approximation Error). *Assume that the outer-level loss function exhibits $\kappa$-strong quasi-convexity over the optimal solutions set of the inner-level problem, the difference between the optimal parameter $\lambda^*$ and its estimated counterpart $\hat{\lambda}$ is upper-bounded by $\|\hat{\lambda}-\lambda^*\|\leqslant 2\epsilon\kappa^{-1}$.*

## 3.3 EVENT POINTS

For illustrative purposes, we consider the case of $\ell_1$ penalty and let $\mathcal{S}=[m]$. While we focus on a specific case for clarity, the analysis presented in Section 3.3 is generalizable to any regularization term satisfying Assumption 1 (*c.f.* Appendix C.2). The thresholding conditions for an $\ell_1$ regularized problem, as put forth by Rosset & Zhu (2007), are reproduced as follows.

**Lemma 1.** *(c.f. (Rosset & Zhu, 2007)) For any loss function $L(y,f(X,\beta))$ that is differentiable w.r.t. $\beta$, any stationary point $\hat{\beta}$ of the optimization problem*

$$\min_\beta\ L(y,f(X,\beta)),\ \ s.t.\ \|\beta\|_1\leq D$$

*ensures the following properties: (a) $\hat{\beta}_i\neq 0\Rightarrow\left|\frac{\partial L}{\partial\beta_i}\right|=\max_j\left|\frac{\partial L}{\partial\beta_j}\right|$, (b) $\mathrm{sign}\left(\hat{\beta}_i\right)=-\mathrm{sign}\left(\frac{\partial L}{\partial\beta_i}\right)$.*

---

[3]We refer here to the uniqueness of the solution to an IVP (5), whereas non-uniqueness for (1) is permissible.

Denote $\alpha = \max_i \left| \frac{\partial L}{\partial \beta_i} \right|$, and suppose the current active set is $\{k_1, k_2\}$. Lemma 1 suggests that $\alpha = \left| \frac{\partial L}{\partial \beta_{k_1}} \right| = \left| \frac{\partial L}{\partial \beta_{k_2}} \right| \geq \left| \frac{\partial L}{\partial \beta_i} \right|$. In the inner-level (*resp.* outer-level), we start at $s = \underline{s}$, $t = \underline{t}$ (*resp.* initial $\lambda$) to compute the trajectory via the PDE-ODE procedure. It can be inferred that the trajectory remains smooth until the occurrence of one of the two possible events: **E1**: Another variable index $k_i$ ($i \neq 1, 2$) joins the active set as $\left| \frac{\partial L}{\partial \beta_{k_i}} \right| = \alpha$; **E2**: Any of the $\beta_{k_i}$ becomes 0 (*c.f.* Figure 3). The solution $\hat{\beta}$ related to $s$, $t$ and $\lambda$ can be computed swiftly before any alterations occur in $\mathcal{A}$. We term the points where an event occurs as **event points**, and the smooth trajectory (path) between two event points as a **segment**. Theorem 6 offers insights into the behaviour of path segmentation. At an event point, we reset $\mathcal{A}$ according to the index violator(s) and proceed to the subsequent segment.

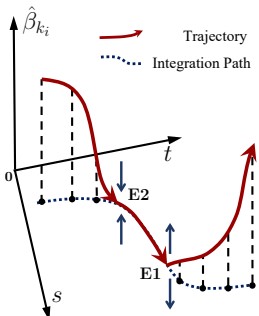

Figure 3: Demonstration of different categories of events. **E1**: $\beta_{k_i}$ enters the trajectory, with its new sign being determined as per Lemma 1 (***b***); **E2**: $\beta_{k_i}$ leaves the path, leading to the removal of $k_i$ from $\mathcal{A}$ and the invalidation of Lemma 1 (***a***).

**Theorem 6** (Segmentation End). *For $(\hat{\beta}, \lambda)$ that satisfies the KKT system of (2), suppose the active set is $\mathcal{P} \cup \{g\}$, but $g$ is just eliminated from the active set. Let $\alpha = \max_i \left| \frac{\partial \ell}{\partial \beta_i} \right|, i \in \mathcal{P}$. If the Jacobian*

$$
\mathcal{J} = \begin{bmatrix} \partial_\beta^2 \left( \ell^\diamond + \ell^+ \right) \{\mathcal{P}, g\} + \sum_{k=1}^d \partial_\beta^2 \sum_{j \in \mathcal{S}_k} P_\eta \left( \left| \hat{\beta}_j \right|, \lambda_k \right) \{\mathcal{P}, g\} & \sum_{k=1}^d \partial_\lambda \sum_{j \in \mathcal{S}_k} P_\eta \left( \left| \hat{\beta}_j \right|, \lambda_k \right) \\ \sum_{k=1}^d \partial_\beta \sum_{j \in \mathcal{S}_k} P_\eta \left( \left| \hat{\beta}_j \right|, \lambda_k \right) \{\mathcal{P}, g\} & \mathbf{0} \end{bmatrix} \tag{7}
$$

*is full rank, then $\hat{\beta}$ or $\lambda$ cannot be the end point of a path segment. Here $\partial_\beta [\cdot] \{\mathcal{P}, g\}$ extracts only the entries $\frac{\partial [\cdot]}{\partial \beta_i}$ with index $i \in \mathcal{P} \cup \{g\}$, $\partial_\beta^2 [\cdot] \{\mathcal{P}, g\}$ extracts $\frac{\partial [\cdot]}{\partial \beta_i \partial \beta_j}$, with index $i, j \in \mathcal{P} \cup \{g\}$.*

**Remark 2** (Discontinuities). *Our extensive empirical studies suggest that when utilizing non-convex penalties, outer-level path discontinuities may arise when predictors enter the model at a non-zero magnitude or vice versa. This phenomenon is attributable to the jumping between local minima, wherein we can employ a warm-start strategy to compute the initial value of the ensuing segment.*

## 4 DISCUSSIONS

### 4.1 SAGO IMPLEMENTATION

The comprehensive SAGO algorithm computes the PDE-ODE procedure on a segment-by-segment basis, which is delineated in Algorithm 1, serves as a promising fitting strategy for online scenarios. At junctures where the events **E1** and **E2** occur, we halt and update the system, subsequently resuming the trajectory until the traversal of $\Omega$ and $\Lambda$ is complete.

**Remark 3** (Extension on Non-linear Pattern). *While we presuppose a linear model in (1), this assumption is far less restrictive than it appears. Indeed, the model can be linear in basis expansions of the original predictors, thereby enabling Algorithm 1 to encompass kernel methods (Shawe-Taylor et al., 2004), wavelets (Antoniadis et al., 1997), boosting techniques (Schapire, 2003) and more.*

### 4.2 COMPLEXITY & CONVERGENCE

Any PDE/ODE solver recurrently evaluates the (partial) derivative in (3) and (5), requiring approximately $\mathcal{O}\left( (n_0 + n_- + n_+) |\mathcal{A}| \right)$ flops to compute the requisite linear systems. When updating $\nabla_{\beta_\mathcal{A}^2}$ or its inverse, one can use the low-rank updating or Woodbury formulae (Ben-Israel & Greville, 2003), with the computational cost capped at $\mathcal{O}\left( |\mathcal{A}|^2 \right)$. The identification of active and inactive penalized predictor by thresholding demands $\mathcal{O}\left( |\mathcal{A}| \right)$ flops. The detailed computational complexity of the entire SAGO Algorithm is contingent upon the explicit form of the loss, the number of smooth segments (events), and the chosen method (solver) for executing the PDE-ODE procedure. This procedure chiefly leverages a numerical library for first-order PDE and ODE, which has undergone extensive investigation within the applied mathematics community (Rhee et al., 2014).

Table 1: Datasets description. The *BC* denotes Binary Classification and *R* for Regression.

| Dataset | Samples | Dim. | Task |
|---|---|---|---|
| creditcard | 284807 | 31 | |
| MiniBooNE | 130064 | 51 | |
| higgs | 98050 | 29 | *BC* |
| numerai28.6 | 96320 | 22 | |
| 2dplanes | 40768 | 11 | |
| ACSIncome | 1664500 | 12 | |
| Buzzinsocialmedia | 583250 | 78 | |
| fried | 40768 | 11 | *R* |
| OnlineNewsPopularity | 39644 | 60 | |
| house_16H | 22784 | 17 | |

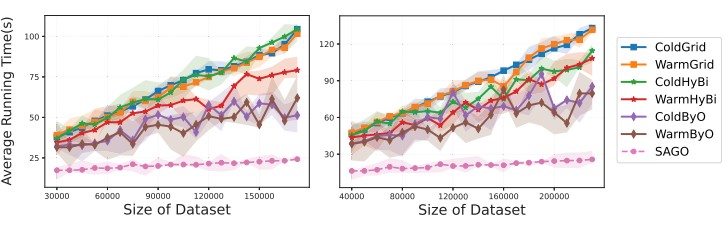

Figure 4: Comparative analysis of algorithmic efficiency. The subfigures on the left and right are GLMs (**i**) and (**ii**), respectively.

The efficiency of Algorithm 1 lies in the fact that SAGO can trace the optimal solutions *w.r.t.* the variables and parameter(s). Throughout this procedure, the solver adaptively selects step sizes to capture all events, thereby eliminating the need for repeated iterations within each round. This stands in contrast to conventional methods that necessitate numerous loops over all samples or coordinates until ultimate convergence. The convergence of SAGO is ensured by forthcoming Theorem 7 and Theorem 8, and more related discussions are given in Appendix C.2.

**Theorem 7** (Successive Adjustments). *On the solving interval, if an index $j'$ is incorporated into the set $\mathcal{A}$, $j'$ will not be removed from $\mathcal{A}$ in the immediate next adjustment cycle; if an index $j''$ is expelled from the set $\mathcal{A}$, $j''$ will not be reintroduced into $\mathcal{A}$ in the immediate next round of adjustment.*

**Theorem 8** (Finite Convergence). *Given the pre-established $\underline{s}$, $\bar{s}$, $\underline{t}$, $\bar{t}$ and $\Lambda$, the SAGO can fit the problem delineated in (2) within a finite number of steps upon Assumption 2.*

---

**Algorithm 1** SAGO Algorithm

**Require:** Initial solution $\hat{\beta}^{(0)}$, online rounds $T$, training sets $X^{\diamond}$, $X^{+}$, $X^{-}$ in each round, tolerance $\epsilon$ for $\lambda$, $max\_iter$.
1: **for** $\tau = 1, \cdots, T$ **do**
2:     Initialize $\mathcal{A} \leftarrow \overline{\mathcal{S}} \cup \mathcal{S}_{\overline{\mathcal{Z}}}$ according to $\hat{\beta}^{(\tau-1)}$.
3:     $s \leftarrow \underline{s}, t \leftarrow \underline{t}$
4:     **while** $s \leq \bar{s}, t \leq \bar{t}$ **do**
5:        Solve the PDE system (3).    ▷ Inner-level problem
6:        **if** any $k \in \mathcal{A}$ has $\hat{\beta}_k = 0$ **then**
7:          Exclude $k$ from $\mathcal{A}$.    ▷ **E2** occurred
8:        **else if** $\hat{\beta}_k(k \in \mathcal{S}_{\mathcal{Z}})$ meets thresholding conditions **then**
9:          Add $k$ into $\mathcal{A}$.    ▷ **E1** occurred
10:        **end if**
11:        Update $\hat{\beta}_{\mathcal{A}}^{(\tau)}$ and $\ell_{i,\mathcal{A}}$ by the modified $\mathcal{A}$.
12:     **end while**
13:     $iters \leftarrow 0$
14:     **while** $\|\nabla_{\lambda}\mathcal{L}_{val}\| > \epsilon$ and $iters \leq max\_iter$ **do**
15:        Solve the ODE system (5).    ▷ Outer-level problem
16:        **if** any $k \in \mathcal{A}$ has $\hat{\beta}_k = 0$ **then**
17:          Exclude $k$ from $\mathcal{A}$.    ▷ **E2** occurred.
18:        **else if** $\hat{\beta}_k(k \in \mathcal{S}_{\mathcal{Z}})$ meets thresholding conditions **then**
19:          Add $k$ into $\mathcal{A}$.    ▷ **E1** occurred.
20:        **end if**
21:        Compute $\nabla_{\lambda}\mathcal{L}_{val}$ via (6).
22:        Update $\hat{\beta}_{\mathcal{A}}^{(\tau)}$ and $\ell_{i,\mathcal{A}}$ by the modified $\mathcal{A}$, and warm-start if necessary.
23:        $iters \leftarrow iters + 1$
24:     **end while**
25:     $\lambda^{(\tau)} \leftarrow \arg\min \nabla_{\lambda}\mathcal{L}_{val}$
26: **end for**
**Ensure:** $\{\hat{\beta}^{(\tau)}\}, \{\lambda^{(\tau)}\}$

---

## 5 EMPIRICAL STUDY

**Sparse GLMs** For evaluation, here we employ two specific GLMs: (**i**) Sparse Poisson Regression (Frome, 1983) given by $\min_{\beta} \frac{1}{n} \sum_i [y_i \log \frac{y_i}{\exp(\beta^{\top}\mathbf{x}_i)} - (y_i - \exp(\beta^{\top}\mathbf{x}_i))] + \lambda_1 \|\beta\|_1 + \lambda_2 \|\beta\|^2$, and (**ii**) Logistic Group Lasso (Meier et al., 2008) (for classification) expressed as $\min_{\beta} \frac{1}{n} \sum_i -y_i \log(h_{\beta}(\mathbf{x}_i)) - (1 - y_i) \log(1 - h_{\beta}(\mathbf{x}_i)) + \lambda \sum_{g \in G} \sqrt{d_g} \|\beta^g\|_2$, wherein $h_{\beta}(\mathbf{x}_i) = (1 + \exp(-\beta^{\top}\mathbf{x}_i))^{-1}$ is hypothesis function. There has a total of $G$ groups, $d_g$ is the number of features in $g$-th group.

**Dataset** We employ real-world datasets from OpenML (Vanschoren et al., 2014) and UCI repository (Asuncion & Newman, 2007) for our simulations. Table 1 summarizes the dataset information. We randomly partition the datasets into training, validation, and testing sets, with 70%, 15%, and 15% of the total samples, respectively.

**Baselines** Given the lack of exact no-regret algorithms for these models prior to SAGO, we compare against the following baselines: (**i**) ColdGrid (*resp.* WarmGrid) employs cold-start (*resp.* warm-

Table 2: Numerical results for validation loss with standard deviation. The best results are shown in bold. We employ GLM (**i**) and GLM (**ii**) for Regression and Binary Classification tasks, respectively.

| Method | creditcard | MiniBooNE | higgs | numerai28.6 | ACSIncome | Buzzinsocialmedia | fried | house_16H |
|---|---|---|---|---|---|---|---|---|
| | | | | **Round #1** | | | | |
| ColdGrid | $0.714 \pm 0.017$ | $0.439 \pm 0.066$ | $0.644 \pm 0.071$ | $0.708 \pm 0.012$ | $256.2 \pm 0.12$ | $12.98 \pm 0.81$ | $3.817 \pm 0.53$ | $27.91 \pm 0.29$ |
| ColdHyBi | $0.693 \pm 0.069$ | $0.393 \pm 0.017$ | $0.669 \pm 0.027$ | $0.711 \pm 0.014$ | $256.0 \pm 0.71$ | $12.88 \pm 0.62$ | $3.994 \pm 0.76$ | $27.43 \pm 0.64$ |
| ColdByO | $0.699 \pm 0.016$ | $0.422 \pm 0.025$ | $0.659 \pm 0.006$ | $0.712 \pm 0.022$ | $256.9 \pm 0.59$ | $13.05 \pm 0.93$ | $3.808 \pm 0.94$ | $26.48 \pm 0.48$ |
| SAGO | $\textbf{0.692} \pm \textbf{0.004}$ | $\textbf{0.389} \pm \textbf{0.009}$ | $\textbf{0.638} \pm \textbf{0.001}$ | $\textbf{0.691} \pm \textbf{0.002}$ | $\textbf{255.9} \pm \textbf{0.09}$ | $\textbf{12.56} \pm \textbf{0.03}$ | $\textbf{3.751} \pm \textbf{0.14}$ | $\textbf{26.25} \pm \textbf{0.07}$ |
| | | | | **Round #2** | | | | |
| ColdGrid | $0.701 \pm 0.057$ | $0.289 \pm 0.041$ | $0.652 \pm 0.043$ | $0.744 \pm 0.043$ | $258.6 \pm 0.08$ | $13.11 \pm 0.94$ | $\textbf{3.680} \pm \textbf{0.41}$ | $27.14 \pm 0.77$ |
| ColdHyBi | $0.698 \pm 0.006$ | $0.299 \pm 0.002$ | $0.666 \pm 0.093$ | $0.756 \pm 0.096$ | $256.1 \pm 0.35$ | $12.68 \pm 0.92$ | $3.844 \pm 0.15$ | $27.62 \pm 0.69$ |
| ColdByO | $0.708 \pm 0.014$ | $0.321 \pm 0.075$ | $0.651 \pm 0.081$ | $0.759 \pm 0.008$ | $257.8 \pm 0.18$ | $12.70 \pm 0.85$ | $3.727 \pm 0.74$ | $27.77 \pm 0.75$ |
| SAGO | $\textbf{0.694} \pm \textbf{0.008}$ | $\textbf{0.287} \pm \textbf{0.021}$ | $\textbf{0.638} \pm \textbf{0.006}$ | $\textbf{0.737} \pm \textbf{0.006}$ | $\textbf{255.7} \pm \textbf{0.48}$ | $\textbf{12.53} \pm \textbf{0.04}$ | $\textbf{3.680} \pm \textbf{0.24}$ | $\textbf{26.59} \pm \textbf{0.14}$ |
| | | | | **Round #3** | | | | |
| WarmGrid | $0.715 \pm 0.043$ | $0.300 \pm 0.014$ | $0.652 \pm 0.093$ | $0.712 \pm 0.087$ | $256.5 \pm 0.88$ | $12.96 \pm 0.49$ | $4.021 \pm 0.85$ | $27.05 \pm 0.11$ |
| WarmHyBi | $0.706 \pm 0.036$ | $\textbf{0.280} \pm \textbf{0.096}$ | $0.645 \pm 0.026$ | $0.673 \pm 0.011$ | $\textbf{255.5} \pm \textbf{0.37}$ | $13.06 \pm 0.66$ | $3.698 \pm 0.45$ | $26.97 \pm 0.34$ |
| WarmByO | $0.708 \pm 0.053$ | $0.307 \pm 0.007$ | $0.657 \pm 0.036$ | $0.688 \pm 0.095$ | $255.9 \pm 0.98$ | $12.94 \pm 0.43$ | $3.925 \pm 0.76$ | $27.25 \pm 0.28$ |
| SAGO | $\textbf{0.693} \pm \textbf{0.006}$ | $\textbf{0.280} \pm \textbf{0.005}$ | $\textbf{0.640} \pm \textbf{0.014}$ | $\textbf{0.668} \pm \textbf{0.003}$ | $\textbf{255.5} \pm \textbf{0.12}$ | $\textbf{12.52} \pm \textbf{0.05}$ | $\textbf{3.679} \pm \textbf{0.39}$ | $\textbf{26.63} \pm \textbf{0.04}$ |

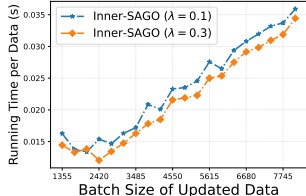

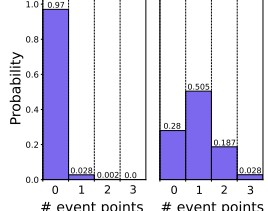

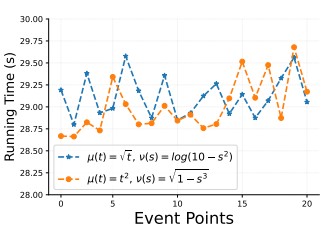

Figure 5: Normalized updating time per data sample (total updating time divided by batch size) for the inner-level problem.

Figure 6: Histograms of event points. The left (or right) panel corresponds to the inner (or outer) problem.

Figure 7: Running time as a function of event point numbers.

start) batch retraining for online updating and utilizes grid search to determine the optimal $\lambda$. (**ii**) ColdHyBi (*resp.* WarmHyBi) applies cold-start (*resp.* warm-start) batch retraining for online updating and leverages hypergradient-based bilevel framework (Feng & Simon, 2018) to find $\lambda^*$. (**iii**) ColdByO (*resp.* WarmByO) adopts cold-start (*resp.* warm-start) batch retraining for online updating and employs Bayesian optimization (Snoek et al., 2012) to identify the optimal $\lambda$. For the grid search, we rely on Scikit-learn (Pedregosa et al., 2011), and for Bayesian optimization, we use the Hyperopt library (Bergstra et al., 2015). For batch re-training, we leverage the popular FISTA optimiser (Beck & Teboulle, 2009) integrated with a gradient-based adaptive restarting scheme (O'donoghue & Candes, 2015).

**Setup** To rigorously assess the validity of our conclusions and the practical efficacy of our SAGO algorithm, we simulate an online environment with varying observations as detailed in Section 2.1. We use 70% of the training set to train an initial model, in which we assign half of the observations with randomly assigned erroneous labels. In each round, the model is updated by progressively eliminating these incorrectly-labeled training data and simultaneously introducing the remaining 30% of unutilized clean data. All experiments are conducted over 50 trials and *more details* are listed in Appendix E.

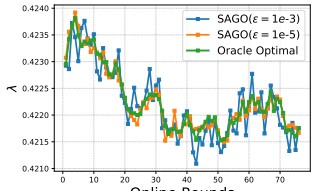

Figure 8: Dynamics of $\lambda$.

**Results & Analysis** At first we perform a single round update (maintaining a constant $n_+ + n_-$) on datasets of varying scales to contrast the runtime of diverse algorithms. The results depicted in Figure 4 underscore the computational advantage of proposed SAGO. We present our numerical results in Table 2, suggesting that the accuracy of our algorithm parallels that of existing methodologies. Moreover, we fix the dataset size $n_0$ and vary the number of dynamically updated samples per round, as illustrated in Figure 5. The running time is observed to scale linearly with the growth of $n_+ + n_-$, which means our method is scalable with the increasing number of varying observations. Figure 6 displays the histograms of event points, revealing a near-absence of event points in the inner-level problem, while most of them are present in the outer-layer problem. Subsequently, we forcibly alter the optimality conditions during a certain round, causing a change in the number of event points. Figure 7 illustrates that the influence of event point count on overall runtime is almost negligible. Lastly, Figure 8 validates SAGO's capability to approximate optimal regularization parameter when the samples are varying in each online round.

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

# Supplemental Material

## Table of Contents

## A  GLOSSARY

| Term/Symbol | Explanation |
| --- | --- |
| SAGO | Algorithm for training a data-varying SpArse GLMs in an Online manner. |
| $\mathbf{A}$ | A corresponding batch algorithm. |
| $d$ | The total number of regularizer(s) in (1). |
| $\Lambda$ | Predefined searching space of $\lambda$. |
| $\lambda_k$ | The regularization parameter of the $k$-th regularizer, or the $k$-th component of $\lambda$. |
| $\beta$ | The learnable parameters / model weights vector. |
| $\beta_j$ | The $j$-th component of vector $\beta$. |
| $\hat{\beta}$ | The optimal solution of $\beta$. |
| $\beta^g$ | The parameter(s) of $g$-th group in group Lasso. |
| $\mathcal{S}_k$ | The subset of coefficients being penalized of the $k$-th regularizer. |
| $X^\diamond, X^-, X^+$ | The (sub)dataset include observation(s) to be kept, removed and incorporated in each round, respectively. |
| $n_0, n_-, n_+$ | The number of observation(s) in $X^\diamond, X^-, X^+$, respectively. |
| $m$ | The size of feature space / number of features in observation. |
| $\{\mathbf{x}_i, y_i\}$ | A pair of $i$-th observation and its label. |
| $\hat{\beta}^{(\mathrm{i})}$ | The optimal solution before each online round (*i.e.*, initial value of (3)). |
| $\hat{\beta}^{(\mathrm{ii})}$ | The optimal solution after each online round. |
| $\ell_i$ | Loss function. |
| $\mathcal{L}_{val}$ | Validation loss. |
| $\{\tilde{\mathbf{x}}_i, \tilde{y}_i\}_{i=1}^N$ | Validation set with $N$ observations. |
| $\underline{t}, \underline{s}$ | Calculated lower bound for $t$ and $s$. |
| $\bar{t}, \bar{s}$ | Calculated upper bound for $t$ and $s$. |
| $\mu(t), \nu(s)$ | The defined parameterizers. |
| $\eta$ | The potential parameter(s) for a penalty family. |
| $T$ | Number of rounds. |
| $\tau$ | Index for iterating over $T$ rounds. |
| $iters$ | Number of current iterations when running SAGO. |
| $max\_iter$ | Maximum allowed number of iterations. |
| $\mathcal{A}$ | Active set. |
| $M$ | A positive constant in Theorem 2. |
| elementwise twice-differentiable | Twice-differentiable, if limited to certain elements / coordinates (in $\mathcal{A}$). |
| reparameterization | The process of reparameterize using parameterizers. |
| knot | The point in regularizer that is not well-defined. |

(Continued)

| Term/Symbol | Explanation |
|---|---|
| $\partial f(x)$ | The subdifferential of $f$. For a proper function $f : \mathbb{R}^n \to \mathbb{R} \cup \{\infty\}$, its subdifferential at $x \in \mathbb{R}^n$ is $\partial f(x) = \{u \in \mathbb{R}^n : \forall y \in \mathbb{R}^n, f(y) \geq f(x) + u^\top (y - x)\}$. |
| $\mathrm{sign}(\cdot)$ | The sign function, defined as $x \mapsto x/|x|$ with the convention $0/0 = 0$. |
| $\exp(\cdot)$ | The exponential function with $e$ as base. Applied to vectors, sign, $\exp(\cdot)$ act element-wise. |
| $\mathbf{1}_{\{j \in \mathcal{S}_k\}}$ | The indicator function, defined as $\mathbf{1}_{\{j \in \mathcal{S}_k\}} := \begin{cases} 1 & \text{if } j \in \mathcal{S}_k \\ 0 & \text{if } j \notin \mathcal{S}_k \end{cases}$. |
| $\kappa$ | To describe $\kappa$-strong quasi-convexity. |
| $(s_k, t_k)$ | The $k$-th event point. |
| $\langle \cdot, \cdot \rangle$ | Inner product. |
| $\nabla_\lambda^2 \hat{\beta}$ | Calculate second-order derivative. |
| $\partial_s^2 \hat{\beta}$ | Calculate second-order partial derivative. |
| $\Omega$ | An integral path $\Omega(s, t) = 0$. |
| $\int_\Omega$ | Curvilinear integral over $\Omega$. |
| $\mathcal{S}_{k, \mathcal{A}}$ | Intersection of the set $\mathcal{S}_k$ and $\mathcal{A}$. |
| $P\{\cdot\}$ | Probability of events. |
| $\epsilon$ | Tolerance for hyperparameter $\lambda$. |
| $\lambda^*$ | Optimal hyperparameter $\lambda$ in the space. |
| $D$ | Some predefined constant in Lemma 1. |
| $\alpha$ | Maximum absolute value of directional derivative in Lemma 1. |
| $\mathcal{O}(\cdot)$ | The Big-O notation of complexity. |
| $h_\beta(\cdot)$ | Hypothesis function. |

# B  OMITTED PROOFS

## B.1  PROOF OF THEOREM 1

We present the formal statement of Theorem 1 as Theorem 9 and subsequently provide its proof.

**Theorem 9** (Equivalence Guarantee). *Let $\mathbf{A}$ be a batch algorithm that trains on the dataset $X \in \mathcal{X}$ and outputs a model $f \in \mathcal{H}$, i.e., $\mathbf{A}(X) : \mathcal{X} \to \mathcal{H}$. Now, assume that there is a requirement to incorporate new observations in $X^+$ and to simultaneously exclude certain existing observations $X^-$, from the trained model using algorithm $\mathbf{A}$. Then $\forall \mathcal{T} \subseteq \mathcal{H}$, we have*

$$P\left\{ \mathtt{SAGO}\left[\mathbf{A}(X^\diamond \cup X^-), X^+, X^-\right] \in \mathcal{T} \right\} = P\left\{ \mathbf{A}(X^\diamond \cup X^+) \in \mathcal{T} \right\}, \tag{8}$$

*where $\cup$ represents standard set operations. The operator $\mathtt{SAGO}[\cdot]$ executes an online update based on the reformulation given by (2). Within the arguments of $\mathtt{SAGO}[\cdot]$, the first term is the output of $\mathbf{A}(\cdot)$, the second term corresponds to the observations to be added, and the third term signifies the observations to be removed.*

*Proof.* We begin by considering the case where $t = \underline{t}$ and $s = \underline{s}$. According to Definition 2, we observe that $\mu(t) = 0$ and $\nu(s) = 1$. This leads to the following representation of (2):

$$
\begin{aligned}
\min_{\lambda \in \Lambda} \; & \mathcal{L}_{val} := \sum_{i=1}^{N} \ell_i \left( \tilde{y}_i, f \left( \hat{\beta}^\top \tilde{\mathbf{x}}_i \right) \right) \\
s.t. \quad & \hat{\beta} \in \arg \min_{\beta \in \mathbb{R}^m} \ell^\diamond(\beta) + 0 + \ell^-(\beta) + \sum_{k=1}^{d} \sum_{j \in \mathcal{S}_k} P_\eta \left( |\beta_j|, \lambda_k \right).
\end{aligned}
\tag{9}
$$

Equation (9) is associated with the model state **(i)** depicted in Figure 1, wherein $\hat{\beta}(s, t)$ in (9) is obtained by minimizing the loss of observations in $X$ and $X_-$. Assuming that (9) is properly solved by a batch algorithm $\mathbf{A}$, the solution is indeed $\mathbf{A}(X^\diamond \cup X^-)$ in (8). We now consider varying $s$ and $t$ to $\bar{s}$ and $\bar{t}$, implying that $\mu(t) = 1$ and $\nu(s) = 0$. This results in

$$
\begin{aligned}
\min_{\lambda \in \Lambda} \; & \mathcal{L}_{val} := \sum_{i=1}^{N} \ell_i \left( \tilde{y}_i, f \left( \hat{\beta}^\top \tilde{\mathbf{x}}_i \right) \right) \\
s.t. \quad & \hat{\beta} \in \arg \min_{\beta \in \mathbb{R}^m} \ell^\diamond(\beta) + \ell^+(\beta) + 0 + \sum_{k=1}^{d} \sum_{j \in \mathcal{S}_k} P_\eta \left( |\beta_j|, \lambda_k \right).
\end{aligned}
\tag{10}
$$

Equation (10) corresponds to the model status **(ii)**, as illustrated in Figure 1, which is equivalent to the model trained by minimizing the loss of observations in $X$ and $X_+$. Assuming that (10) is correctly solved by a batch algorithm $\mathbf{A}$, the solution is denoted as $\mathbf{A}(X^\diamond \cup X^+)$ in (8). Since we employ our $\text{SAGO}[\cdot]$ to transition from (9) to (10), the equivalence of our $\text{SAGO}$ and batch retraining methodology is thereby established. $\square$

## B.2 PROOF OF THEOREM 2

Before presenting our proof of Theorem 2, it is essential to establish the existence of the path of solutions $\left\{ \hat{\beta}(s, t) \mid (s, t) \in [\underline{s}, \bar{s}] \times [\underline{t}, \bar{t}] \right\}$ for the inner problem, and $\left\{ \hat{\beta}(\lambda) \mid \lambda \in \Lambda \right\}$ for the outer problem as a critical prerequisite.

**Lemma 2.** *The solution paths $\hat{\beta}$ w.r.t. (2) exist for $\forall (s, t) \in [\underline{s}, \bar{s}] \times [\underline{t}, \bar{t}]$ for the inner problem, and for $\forall \lambda \in \Lambda$ for the outer problem.*

*Proof.* Our proof is inspired by ideas in probability-one homotopy methods (Chow et al., 1978). We make use of the $1$-$D$ manifold classification theorem in differential topology (Milnor & Weaver, 1997), which states that a 1-dimensional smooth manifold must be homeomorphic either to a line segment or to a circle. Therefore, we need to show two things: *(i)* first, the path is (close to) a $1$-$D$ manifold (it does not self-intersect); *(ii)* second, this manifold cannot be a circle.

The first is a local property, and the implicit function theorem (Krantz & Parks, 2002) is used to prove it. To prove that the manifold of solutions cannot be a circle, we make sure that the path starts at $(\underline{s}, \underline{t})$ in a single direction. When this holds, it cannot be a circle, since a circle will imply two directions to choose from at each point. To see this, suppose that the manifold of solutions is a circle, and let $\mathbb{C}$ be the circle. Let $v$ be a unit tangent vector at a point $p$ on $\mathbb{C}$, which gives us a well-defined direction at $p$. Since the manifold of solutions is a path, it intersects the circle at two points $p_1$ and $p_2$. Let $p_1$ be the starting point of the path, i.e., $p_1 = (\underline{s}, \underline{t})$. Without loss of generality, we can assume that $p_2$ is in the counterclockwise direction from $p_1$. Then, let $v_1$ be the unit tangent vector to $\mathbb{C}$ at $p_1$, and let $v_2$ be the unit tangent vector to $\mathbb{C}$ at $p_2$. Since $\mathbb{C}$ is a circle, $v_1$ and $v_2$ are opposite directions. However, since the manifold of solutions is a path, it cannot change direction suddenly, so it must continue in the same direction as $v_1$ as it leaves $p_1$. But this is a contradiction since $v_1$ and $v_2$ are opposite directions, so the manifold of solutions cannot be a circle.

The proof for the existence of solution paths in the outer problem is analogous to that of the inner problem; therefore, we omit it here. $\square$

We proceed to prove Theorem 2, first demonstrating the solution continuity of the outer problem, followed by a proof of the continuous path *w.r.t.* the inner problem. For the sake of simplicity and

uniformity, we designate the Hessian matrix $\nabla_\lambda^2 \hat{\beta}(\lambda)$ as $\boldsymbol{H}_\lambda(\hat{\beta})$. Similarly, we use the notation $\boldsymbol{H}_{\widetilde{\mathcal{L}}}(\hat{\beta})$ to represent $\widetilde{\mathcal{L}}^{-1}(\cdot)$. We also introduce the shorthand $\boldsymbol{H}_s$ and $\boldsymbol{H}_t$ to denote the second partial derivatives *w.r.t.* $s$ and $t$, respectively, of $\hat{\beta}$, that is $\partial_s^2 \hat{\beta}(s,t)$ and $\partial_t^2 \hat{\beta}(s,t)$.

*Proof.* Let $\hat{\beta}(\lambda)$ denote the solution corresponding to the hyperparameter value $\lambda$ of the outer problem, and let $\hat{\beta}(\lambda + \Delta\lambda)$ represent the solution associated with the hyperparameter value $\lambda + \Delta\lambda$. Our aim is to demonstrate that $\left\| \hat{\beta}(\lambda) - \hat{\beta}(\lambda + \Delta\lambda) \right\| \to 0$ as $\Delta\lambda \to 0$. By employing the fundamental theorem of calculus, we can write[4]

$$\begin{aligned}
\hat{\beta}(\lambda + \Delta\lambda) - \hat{\beta}(\lambda) &= \int_\lambda^{\lambda + \Delta\lambda} \frac{d\hat{\beta}(\lambda')}{d\lambda'} d\lambda' \\
&= \frac{1}{\Delta\lambda} \int_\lambda^{\lambda + \Delta\lambda} \frac{d\hat{\beta}(\lambda')}{d\lambda'} \Delta\lambda d\lambda'.
\end{aligned} \tag{11}$$

Taking the norm of both sides yields

$$\left\| \hat{\beta}(\lambda + \Delta\lambda) - \hat{\beta}(\lambda) \right\| \le \frac{1}{|\Delta\lambda|} \int_\lambda^{\lambda + \Delta\lambda} \left\| \frac{d\hat{\beta}(\lambda')}{d\lambda'} \right\| d\lambda'. \tag{12}$$

Utilizing Taylor's theorem, we arrive at

$$\hat{\beta}(\lambda + \Delta\lambda) - \hat{\beta}(\lambda) = \mathbf{v}^\top \Delta\lambda + \frac{1}{2}(\Delta\lambda)^\top \boldsymbol{H}_\lambda(\hat{\beta}')\Delta\lambda, \tag{13}$$

where $\hat{\beta}'$ is a point on the line segment between $\hat{\beta}(\lambda)$ and $\hat{\beta}(\lambda + \Delta\lambda)$. Rearranging terms, we get

$$\left\| \hat{\beta}(\lambda + \Delta\lambda) - \hat{\beta}(\lambda) - \mathbf{v}^\top \Delta\lambda \right\| \le \frac{1}{2}M \|\Delta\lambda\|^2. \tag{14}$$

We need to demonstrate that the right-hand side of (12) goes to zero as $\Delta\lambda \to 0$. To achieve this, we shall use the assumption that $\boldsymbol{H}_{\widetilde{\mathcal{L}}}(\hat{\beta}) \preceq MI$ for some positive constant $M$ and for every $\hat{\beta}$. Let $\mathbf{v} = \frac{d\hat{\beta}(\lambda')}{d\lambda'}$ be a vector. Firstly, we can bound $\mathbf{v}$ as

$$\begin{aligned}
\left\| \frac{d\hat{\beta}(\lambda')}{d\lambda'} \right\| &= \left\| \sum_{k=1}^d \left[ \widetilde{\mathcal{L}}\left( \hat{\beta}(\bar{s}, \bar{t}), \lambda \right) \right]^{-1} \cdot \nabla_\beta \sum_{j \in \mathcal{S}_k} P_\eta\left( \left| \hat{\beta}_j \right|, \lambda_k \right) \right\| \\
&\le \left\| \left[ \widetilde{\mathcal{L}}\left( \hat{\beta}(\bar{s}, \bar{t}), \lambda \right) \right]^{-1} \right\| \cdot \left\| \sum_{k=1}^d \nabla_\beta \sum_{j \in \mathcal{S}_k} P_\eta\left( \left| \hat{\beta}_j \right|, \lambda_k \right) \right\| \\
&\le \frac{1}{m} \cdot \frac{\sqrt{d}}{\widetilde{\lambda}_{\max}(\boldsymbol{H}_{\widetilde{\mathcal{L}}}(\hat{\beta}))} \cdot \left\| \nabla_\beta \sum_{j \in \mathcal{S}_k} P_\eta\left( \left| \hat{\beta}_j \right|, \lambda_k \right) \right\| \\
&\le \frac{1}{m} \cdot \frac{\sqrt{d}}{\sqrt{M}} \cdot \left\| \nabla_\beta \sum_{j \in \mathcal{S}_k} P_\eta\left( \left| \hat{\beta}_j \right|, \lambda_k \right) \right\|,
\end{aligned} \tag{15}$$

where $\widetilde{\lambda}_{\max}(\boldsymbol{H}_{\widetilde{\mathcal{L}}}(\hat{\beta}))$ is the maximum eigenvalue of the Hessian matrix $\boldsymbol{H}_{\widetilde{\mathcal{L}}}(\hat{\beta})$ at $\hat{\beta}$. By employing the Cauchy-Schwarz inequality, we can also derive the following inequality

$$\begin{aligned}
\left\| \frac{d\hat{\beta}(\lambda')}{d\lambda'} \right\| &= \frac{1}{\|\mathbf{v}\|} \|\mathbf{v}\|^2 \\
&\le \frac{1}{\sqrt{\lambda' - \lambda}} \left( \int_\lambda^{\lambda'} \|\mathbf{v}\|^2 d\lambda' \right)^{\frac{1}{2}} \\
&\le \frac{1}{\sqrt{|\Delta\lambda|}} \left( \int_\lambda^{\lambda + \Delta\lambda} \|\mathbf{v}\|^2 d\lambda' \right)^{\frac{1}{2}}.
\end{aligned} \tag{16}$$

---

[4]We employ the shorthand notation $\hat{\beta}$ (or $\ell_i$) in place of $\hat{\beta}_{\mathcal{A}}$ (or $\ell_{i,\mathcal{A}}$) whenever the context allows for such simplification.

Substituting this inequality into (12) and then using (14) and (15), we get

$$
\begin{aligned}
\left\| \hat{\beta}\left(\lambda + \Delta\lambda\right) - \hat{\beta}(\lambda) \right\| &\leq \frac{1}{|\Delta\lambda|} \int_{\lambda}^{\lambda + \Delta\lambda} \left\| \frac{d\hat{\beta}(\lambda')}{d\lambda'} \right\| d\lambda' \\
&\leq \frac{1}{|\Delta\lambda|^{\frac{3}{2}}} \int_{\lambda}^{\lambda + \Delta\lambda} \left( \int_{\lambda}^{\lambda + \Delta\lambda} |\mathbf{v}|^2 d\lambda' \right)^{\frac{1}{2}} d\lambda' \\
&\leq \frac{1}{|\Delta\lambda|^{\frac{3}{2}}} \int_{\lambda}^{\lambda + \Delta\lambda} \left( \int_{\lambda}^{\lambda + \Delta\lambda} M^2 (\Delta\lambda)^2 d\lambda' \right)^{\frac{1}{2}} d\lambda' \\
&\leq \frac{M|\Delta\lambda|^2}{|\Delta\lambda|^{\frac{3}{2}}} |\Delta\lambda| \\
&= M|\Delta\lambda|^{\frac{3}{2}}.
\end{aligned}
\tag{17}
$$

Since $M$ is a positive constant, it is evident that the right-hand side of this inequality goes to zero as $\Delta\lambda \to 0$. Consequently, we have established that $\left\| \hat{\beta}\left(\lambda + \Delta\lambda\right) - \hat{\beta}(\lambda) \right\| \to 0$ as $\Delta\lambda \to 0$, which implies that $\hat{\beta}$ is continuous *w.r.t.* the variables of the outer problem.

For the inner problem, we revisit the system of PDE (3) as follows.

$$
\begin{cases}
\dfrac{d\mu(t)}{dt} \cdot \nabla_\beta \ell^+(\hat{\beta}) + \widetilde{\mathcal{L}}\left(\hat{\beta}, \lambda\right) \cdot \dfrac{\partial \hat{\beta}(s,t)}{\partial t} = \mathbf{0} \\[2mm]
\dfrac{d\nu(s)}{ds} \cdot \nabla_\beta \ell^-(\hat{\beta}) + \widetilde{\mathcal{L}}\left(\hat{\beta}, \lambda\right) \cdot \dfrac{\partial \hat{\beta}(s,t)}{\partial s} = \mathbf{0} \\[2mm]
\widetilde{\mathcal{L}}\left(\hat{\beta}, \lambda\right) = \displaystyle\sum_{k=1}^{d} \left[ \nabla_{\beta^2} \sum_{j \in \mathcal{S}_k} P_\eta\left( \left|\hat{\beta}_j\right|, \lambda_k \right) \right] + \nabla_{\beta^2} \left[ \ell^\diamond(\hat{\beta}) + \mu(t)\, \ell^+(\hat{\beta}) + \nu(s)\, \ell^-(\hat{\beta}) \right]
\end{cases}
\tag{18}
$$

Our objective is to demonstrate that the solution $\hat{\beta}(s,t)$ is continuous *w.r.t.* both $s$ and $t$. To this end, let $\beta_1(s,t)$ and $\beta_2(s,t)$ represent two solutions to the system in (18). Define the difference $\Delta\beta(s,t) = \beta_1(s,t) - \beta_2(s,t)$. Our goal is to show that $\Delta\beta(s,t)$ can be made arbitrarily small by selecting $s$ and $t$ sufficiently close, which implies the continuity of $\hat{\beta}(s,t)$. To accomplish this, consider the equation for $\Delta\beta(s,t)$, derived by subtracting the two equations in (18):

$$
\begin{aligned}
&\frac{d\mu(t_1)}{dt} \cdot \nabla_\beta \ell^+(\hat{\beta}_1) - \frac{d\mu(t_2)}{dt} \cdot \nabla_\beta \ell^+(\hat{\beta}_2) + \widetilde{\mathcal{L}}\left(\hat{\beta}_1, \lambda\right) \cdot \frac{\partial \hat{\beta}_1(s_1, t_1)}{\partial t} - \widetilde{\mathcal{L}}\left(\hat{\beta}_2, \lambda\right) \cdot \frac{\partial \hat{\beta}_2(s_2, t_2)}{\partial t} \\
&+ \frac{d\nu(s_1)}{ds} \cdot \nabla_\beta \ell^-(\hat{\beta}_1) - \frac{d\nu(s_2)}{ds} \cdot \nabla_\beta \ell^-(\hat{\beta}_2) + \widetilde{\mathcal{L}}\left(\hat{\beta}_1, \lambda\right) \cdot \frac{\partial \hat{\beta}_1(s_1, t_1)}{\partial s} - \widetilde{\mathcal{L}}\left(\hat{\beta}_2, \lambda\right) \cdot \frac{\partial \hat{\beta}_2(s_2, t_2)}{\partial s} = \mathbf{0}.
\end{aligned}
\tag{19}
$$

By rearranging terms and employing the Taylor's Theorem with remainder estimates (Folland, 1990), we obtain

$$
\begin{aligned}
&\Delta\hat{\beta}(s_2, t_2) \\
&= \Delta\hat{\beta}(s_1, t_1) + \nabla_{s,t} \Delta\hat{\beta}(s_1, t_1) \cdot \begin{pmatrix} s_2 - s_1 \\ t_2 - t_1 \end{pmatrix} + \frac{1}{2!} \begin{pmatrix} s_2 - s_1 \\ t_2 - t_1 \end{pmatrix}^\top \cdot \nabla_{s,t}^2 \Delta\hat{\beta}(s_1, t_1) \cdot \begin{pmatrix} s_2 - s_1 \\ t_2 - t_1 \end{pmatrix} + \mathcal{O}\left( \|(s_2 - s_1, t_2 - t_1)\|^3 \right) \\
&\leq \Delta\hat{\beta}(s_1, t_1) + \nabla_{s,t} \Delta\hat{\beta}(s_1, t_1) \cdot \begin{pmatrix} s_2 - s_1 \\ t_2 - t_1 \end{pmatrix} \\
&+ \frac{1}{2} \int_0^1 (1 - \gamma) \cdot \begin{pmatrix} s_2 - s_1 \\ t_2 - t_1 \end{pmatrix}^\top \cdot \left[ \frac{\partial \nabla_{s,t} \Delta\hat{\beta}(\cdot, t)}{\partial s} \Bigg|_{s = \gamma s_2 + (1-\gamma) s_1} (s_2 - s_1) + \frac{\partial \nabla_{s,t} \Delta\hat{\beta}(s, \cdot)}{\partial t} \Bigg|_{t = \gamma t_2 + (1-\gamma) t_1} (t_2 - t_1) \right] d\gamma,
\end{aligned}
\tag{20}
$$

where we utilize the fact that, according to (18), the functions $\frac{\partial \hat{\beta}(s,t)}{\partial s}$ and $\frac{\partial \hat{\beta}(s,t)}{\partial t}$ are continuous *w.r.t.* $s$ and $t$, respectively. Observe that the integrand in the final term of (20) can be bounded using the

Cauchy-Schwarz inequality and triangle inequality, as follows

$$\left\| \begin{pmatrix} s_2 - s_1 \\ t_2 - t_1 \end{pmatrix}^\top \left[ \left. \frac{\partial \nabla_{s,t} \Delta \hat{\beta}(\cdot, t)}{\partial s} \right|_{s=\gamma s_2 + (1-\gamma)s_1} (s_2 - s_1) + \left. \frac{\partial \nabla_{s,t} \Delta \hat{\beta}(s, \cdot)}{\partial t} \right|_{t=\gamma t_2 + (1-\gamma)t_1} (t_2 - t_1) \right] \right\|$$

$$\leq \left\| \begin{pmatrix} s_2 - s_1 \\ t_2 - t_1 \end{pmatrix} \right\| \cdot \left\| \left. \frac{\partial \nabla_{s,t} \Delta \hat{\beta}(\cdot, t)}{\partial s} \right|_{s=\gamma s_2 + (1-\gamma)s_1} (s_2 - s_1) + \left. \frac{\partial \nabla_{s,t} \Delta \hat{\beta}(s, \cdot)}{\partial t} \right|_{t=\gamma t_2 + (1-\gamma)t_1} (t_2 - t_1) \right\|$$

$$\leq \left\| \begin{pmatrix} s_2 - s_1 \\ t_2 - t_1 \end{pmatrix} \right\| \cdot \left( \left\| \left. \frac{\partial \nabla_{s,t} \Delta \hat{\beta}(\cdot, t)}{\partial s} \right|_{s=\gamma s_2 + (1-\gamma)s_1} \right\| \cdot |s_2 - s_1| + \left\| \left. \frac{\partial \nabla_{s,t} \Delta \hat{\beta}(s, \cdot)}{\partial t} \right|_{t=\gamma t_2 + (1-\gamma)t_1} \right\| \cdot |t_2 - t_1| \right), \tag{21}$$

Substituting this bound into (20) leads to

$$\left\| \Delta \hat{\beta}(s_2, t_2) \right\|$$

$$\leq \left\| \Delta \hat{\beta}(s_1, t_1) \right\| + \left\| \nabla_{s,t} \Delta \hat{\beta}(s_1, t_1) \cdot \begin{pmatrix} s_2 - s_1 \\ t_2 - t_1 \end{pmatrix} \right\|$$

$$+ \frac{1}{2} \left\| \int_0^1 (1 - \gamma) \begin{pmatrix} s_2 - s_1 \\ t_2 - t_1 \end{pmatrix}^\top \cdot \left[ \left. \frac{\partial \nabla_{s,t} \Delta \hat{\beta}(\cdot, t)}{\partial s} \right|_{s=\gamma s_2 + (1-\gamma)s_1} (s_2 - s_1) + \left. \frac{\partial \nabla_{s,t} \Delta \hat{\beta}(s, \cdot)}{\partial t} \right|_{t=\gamma t_2 + (1-\gamma)t_1} (t_2 - t_1) \right] d\gamma \right\|$$

$$\leq \left\| \Delta \hat{\beta}(s_1, t_1) \right\| + \left\| \nabla_{s,t} \Delta \hat{\beta}(s_1, t_1) \cdot \begin{pmatrix} s_2 - s_1 \\ t_2 - t_1 \end{pmatrix} \right\|$$

$$+ \frac{1}{2} \left\| \begin{pmatrix} s_2 - s_1 \\ t_2 - t_1 \end{pmatrix} \right\| \cdot \left( \| \nabla_{s,t} \Delta \beta(s_1, t_1) \| + \sqrt{2} M \cdot \sqrt{(s_2 - s_1)^2 + (t_2 - t_1)^2} \right)$$

$$\leq \left\| \Delta \hat{\beta}(s_1, t_1) \right\| + \frac{3}{2} \left\| \nabla_{s,t} \Delta \hat{\beta}(s_1, t_1) \right\| \cdot \left\| \begin{pmatrix} s_2 - s_1 \\ t_2 - t_1 \end{pmatrix} \right\| + M \cdot \left( (s_2 - s_1)^2 + (t_2 - t_1)^2 \right), \tag{22}$$

where we employ the assumption that $\boldsymbol{H}_s \preceq M \boldsymbol{1}_m$ and $\boldsymbol{H}_t \preceq M \boldsymbol{1}_m$ for every $\hat{\beta}$. Without loss of generality, we assume that $\Delta \hat{\beta}(s_1, t_1) = \boldsymbol{0}$, *i.e.*, $\hat{\beta}_1(s_1, t_1) = \hat{\beta}_2(s_1, t_1)$. Then, we find

$$\left\| \Delta \hat{\beta}(s_2, t_2) \right\| \leq \frac{3}{2} \left\| \nabla_{s,t} \Delta \hat{\beta}(s_1, t_1) \right\| \cdot \left\| \begin{pmatrix} s_2 - s_1 \\ t_2 - t_1 \end{pmatrix} \right\| + M \cdot \left( (s_2 - s_1)^2 + (t_2 - t_1)^2 \right). \tag{23}$$

Let $\epsilon > 0$ be given. Since $\frac{\partial \hat{\beta}(s,t)}{\partial s}$ and $\frac{\partial \hat{\beta}(s,t)}{\partial t}$ are continuous *w.r.t.* $s$ and $t$, there exists $\delta > 0$ such that for all $s, t$ with distance $\sqrt{(s_2 - s_1)^2 + (t_2 - t_1)^2} < \delta$, we have $\left\| \nabla_{s,t} \Delta \hat{\beta}(s_1, t_1) \right\| < \frac{\epsilon}{3}$. Now, let $\Delta = \min \left\{ \frac{\delta}{2}, \frac{\epsilon}{3M} \right\}$. Then, for any $s_2, t_2$ with $\sqrt{(s_2 - s_1)^2 + (t_2 - t_1)^2} < \Delta$, it can be inferred that the following inequalities

$$\left\| \Delta \hat{\beta}(s_2, t_2) \right\|$$

$$\leq \frac{3}{2} \left\| \nabla_{s,t} \Delta \hat{\beta}(s_1, t_1) \right\| \cdot \left\| \begin{pmatrix} s_2 - s_1 \\ t_2 - t_1 \end{pmatrix} \right\| + M \cdot \left( (s_2 - s_1)^2 + (t_2 - t_1)^2 \right)$$

$$< \frac{\epsilon}{2} \cdot \left\| \begin{pmatrix} s_2 - s_1 \\ t_2 - t_1 \end{pmatrix} \right\| + M \cdot \Delta^2$$

$$\leq \epsilon. \tag{24}$$

holds. This result demonstrates that $\Delta \hat{\beta}(s, t)$ is a continuous function of $s$ and $t$. Therefore, $\hat{\beta}(s, t)$ is also a continuous function of $s$ and $t$, as it can be obtained by adding $\Delta \hat{\beta}(s, t)$ to $\hat{\beta}_1(s, t)$, which concludes the proof. □

### B.3 PROOF OF THEOREM 3

We commence by establishing Lemma 3, and thereafter, we present a complete proof of Theorem 3.

**Lemma 3** (Necessary Optimality Condition). *Given $(s,t)$, if $\hat{\beta}$ is a local minimum of the inner problem of (2), then $\hat{\beta}$ fulfills the stationarity condition of the inner problem, expressed as*

$$\mu(t)\cdot\nabla_{\beta_j}\ell^+(\hat{\beta})+\nu(s)\cdot\nabla_{\beta_j}\ell^-(\hat{\beta})+\nabla_{\beta_j}\ell^\diamond(\hat{\beta})+\sum_{k=1}^{d}\frac{\partial P_\eta\left(\left|\hat{\beta}_j\right|,\lambda_k\right)}{\partial\left|\beta_j\right|}\omega_j\mathbf{1}_{\{j\in\mathcal{S}_k\}}=\mathbf{0},\quad\forall j\in[m],$$

(25)

*where the coefficient $\omega_j$ meets*

$$\omega_j\in\begin{cases}\{-1\}&\text{if }\hat{\beta}_j<0\\[-1,1]&\text{if }\hat{\beta}_j=0\\\{1\}&\text{if }\hat{\beta}_j>0\end{cases}.$$

(26)

*Proof.* Two potential cases are contemplated. (**i**) When the penalty function $P_\eta\left(|\beta_j|,\lambda_k\right)$ is convex, this naturally corresponds to the first-order optimality condition for unconstrained convex minimization (Boyd et al., 2004). (**ii**) Alternatively, when $P_\eta\left(|\beta_j|,\lambda_k\right)$ is non-convex, the optimality condition is examined on a coordinate-wise basis. For $j\in\{j:\hat{\beta}_j\neq0\}$, this is self-evident. When $\hat{\beta}_j=0$, the implication of $\hat{\beta}_j$ being a local minimum is that the two directional derivatives[5] are non-negative (Danilova et al., 2022). We denote the inner-level objective of the parameterized problem (2) by $\Theta(\beta)$. Let $\boldsymbol{e}_j$ represent a steering vector. Then, we obtain

$$d_{\boldsymbol{e}_j}\Theta(\beta)=\lim_{\varepsilon\downarrow0}\frac{\Theta\left(\beta+\varepsilon\boldsymbol{e}_j\right)-\Theta(\beta)}{\varepsilon}=\nabla_j\left[\ell^\diamond(\beta)+\mu(t)\ell^+(\beta)+\nu(s)\ell^-(\beta)\right]+\sum_{k=1}^{d}\frac{\partial P_\eta\left(|\beta_j|,\lambda_k\right)}{\partial|\beta_j|}\geq0$$

$$d_{-\boldsymbol{e}_j}\Theta(\beta)=\lim_{\varepsilon\uparrow0}\frac{\Theta\left(\beta+\varepsilon\boldsymbol{e}_j\right)-\Theta(\beta)}{\varepsilon}=-\nabla_j\left[\ell^\diamond(\beta)+\mu(t)\ell^+(\beta)+\nu(s)\ell^-(\beta)\right]+\sum_{k=1}^{d}\frac{\partial P_\eta\left(|\beta_j|,\lambda_k\right)}{\partial|\beta_j|}\geq0,$$

(27)

which is equivalent to (25) with $\omega_j\in[-1,1]$. □

In light of Lemma 3, we now proceed to delineate the proof of Theorem 3.

*Proof.* To infer the inner-level dynamics, we treat $\hat{\beta}$ as a bivariate function of $s$ and $t$. Initially, we compute the partial derivative *w.r.t.* $t$, which adheres to the subsequent equation

$$\frac{d\mu(t)}{dt}\nabla_{\beta_{\mathcal{A}}}\ell^+_{\mathcal{A}}\left(\hat{\beta}(s,t)\right)+\mu(t)\nabla^2_{\beta_{\mathcal{A}}}\ell^+_{\mathcal{A}}\left(\hat{\beta}(s,t)\right)\frac{\partial\hat{\beta}_{\mathcal{A}}(s,t)}{\partial t}+\nu(s)\nabla^2_{\beta_{\mathcal{A}}}\ell^-_{\mathcal{A}}\left(\hat{\beta}(s,t)\right)\frac{\partial\hat{\beta}_{\mathcal{A}}(s,t)}{\partial t}+$$

$$\nabla^2_{\beta_{\mathcal{A}}}\ell^\diamond_{\mathcal{A}}\left(\hat{\beta}(s,t)\right)\frac{\partial\hat{\beta}_{\mathcal{A}}(s,t)}{\partial t}+\sum_{k=1}^{d}\left[\nabla^2_{\beta_{\mathcal{A}}}\sum_{j\in\mathcal{S}_{k,\mathcal{A}}}P_\eta\left(\left|\hat{\beta}_j\right|,\lambda_k\right)\right]\frac{\partial\hat{\beta}_{\mathcal{A}}(s,t)}{\partial t}=\mathbf{0}.$$

(28)

Likewise, the partial derivative *w.r.t.* $s$ is established as follows

$$\mu(t)\nabla^2_{\beta_{\mathcal{A}}}\ell^+_{\mathcal{A}}\left(\hat{\beta}(s,t)\right)\frac{\partial\hat{\beta}_{\mathcal{A}}(s,t)}{\partial s}+\frac{d\nu(s)}{ds}\nabla_{\beta_{\mathcal{A}}}\ell^-_{\mathcal{A}}\left(\hat{\beta}(s,t)\right)+\nu(s)\nabla^2_{\beta_{\mathcal{A}}}\ell^-_{\mathcal{A}}\left(\hat{\beta}(s,t)\right)\frac{\partial\hat{\beta}_{\mathcal{A}}(s,t)}{\partial s}+$$

$$\nabla^2_{\beta_{\mathcal{A}}}\ell^\diamond_{\mathcal{A}}\left(\hat{\beta}(s,t)\right)\frac{\partial\hat{\beta}_{\mathcal{A}}(s,t)}{\partial s}+\sum_{k=1}^{d}\left[\nabla^2_{\beta_{\mathcal{A}}}\sum_{j\in\mathcal{S}_{k,\mathcal{A}}}P_\eta\left(\left|\hat{\beta}_j\right|,\lambda_k\right)\right]\frac{\partial\hat{\beta}_{\mathcal{A}}(s,t)}{\partial s}=\mathbf{0}.$$

(29)

---

[5]Considering the path illustrated in Figure 2, we can observe that at any given point on $\Omega$, a maximum of two distinct directions can be identified.

In situations where the expression does not engender ambiguity, we employ $\hat{\beta}$ as a shorthand for $\hat{\beta}(s,t)$. By reordering the terms in (28) and (29), we acquire

$$
\mathbf{0} = \begin{cases}
\dfrac{d\mu(t)}{dt}\nabla_{\beta_{\mathcal{A}}}\ell_{\mathcal{A}}^{+}(\hat{\beta}) + \underbrace{\left[\nabla_{\beta_{\mathcal{A}}}^{2}\left(\ell_{\mathcal{A}}^{\diamond}(\hat{\beta}) + \mu(t)\ell_{\mathcal{A}}^{+}(\hat{\beta}) + \nu(s)\ell_{\mathcal{A}}^{-}(\hat{\beta})\right) + \sum_{k=1}^{d}\left(\nabla_{\beta_{\mathcal{A}}}^{2}\sum_{j\in\mathcal{S}_{k,\mathcal{A}}}P_{\eta}\left(\left|\hat{\beta}_{j}\right|,\lambda_{k}\right)\right)\right]}_{\text{a common term, denotes as }\widetilde{\mathcal{L}}(\hat{\beta},\lambda)}\dfrac{\partial\hat{\beta}_{\mathcal{A}}(s,t)}{\partial t} \\[4ex]
\dfrac{d\nu(s)}{ds}\nabla_{\beta_{\mathcal{A}}}\ell_{\mathcal{A}}^{-}(\hat{\beta}) + \left[\nabla_{\beta_{\mathcal{A}}}^{2}\left(\ell_{\mathcal{A}}^{\diamond}(\hat{\beta}) + \mu(t)\ell_{\mathcal{A}}^{+}(\hat{\beta}) + \nu(s)\ell_{\mathcal{A}}^{-}(\hat{\beta})\right) + \sum_{k=1}^{d}\left(\nabla_{\beta_{\mathcal{A}}}^{2}\sum_{j\in\mathcal{S}_{k,\mathcal{A}}}P_{\eta}\left(\left|\hat{\beta}_{j}\right|,\lambda_{k}\right)\right)\right]\dfrac{\partial\hat{\beta}_{\mathcal{A}}(s,t)}{\partial s}.
\end{cases}
$$
(30)

Incorporating the initial conditions of the system, we arrive at

$$
\begin{cases}
\dfrac{d\mu(t)}{dt}\cdot\nabla_{\beta_{\mathcal{A}}}\ell_{\mathcal{A}}^{+}(\hat{\beta}) + \widetilde{\mathcal{L}}\left(\hat{\beta},\lambda\right)\cdot\dfrac{\partial\hat{\beta}_{\mathcal{A}}(s,t)}{\partial t} = \mathbf{0} \\[2ex]
\dfrac{d\nu(s)}{ds}\cdot\nabla_{\beta_{\mathcal{A}}}\ell_{\mathcal{A}}^{-}(\hat{\beta}) + \widetilde{\mathcal{L}}\left(\hat{\beta},\lambda\right)\cdot\dfrac{\partial\hat{\beta}_{\mathcal{A}}(s,t)}{\partial s} = \mathbf{0} \\[2ex]
\hat{\beta}(s,t)\Big|_{s=\underline{s},t=\underline{t}} = \hat{\beta}_{0},
\end{cases}
$$
(31)

wherein the $\widetilde{\mathcal{L}}\left(\hat{\beta},\lambda\right)$ has been pre-defined in (30). $\qquad\square$

**Remark 4.** *Furthermore, the $\widetilde{\mathcal{L}}\left(\hat{\beta},\lambda\right)$ is verified to be positive semi-definite. The verification of the positive semidefiniteness w.r.t. $\widetilde{\mathcal{L}}\left(\hat{\beta},\lambda\right)$ is drawn from the second-order necessary optimality condition when constrained to coordinates within the set $\mathcal{A}$.*

### B.4 PROOF OF THEOREM 4

**Lemma 4** (*c.f.* Lemma 2.1 in Ghadimi & Wang (2018))**.** *Assuming for any $\lambda\in\Lambda$, the $\partial_{\omega^{2}}G\left(\lambda,\omega_{\lambda}\right)$ is invertible. We consider*

$$
\min_{\lambda\in\Lambda}f(\lambda) := F\left(\lambda,\omega_{\lambda}\right) \quad s.t.\ \omega_{\lambda} = \arg\min_{\omega}G(\lambda,\omega)
$$
(32)

*and the hyper-gradient w.r.t $\lambda$ takes the form*

$$
\nabla_{\lambda}f = \partial_{\lambda}F\left(\lambda,\omega_{\lambda}\right) - \partial_{\omega\lambda}G\left(\lambda,\omega_{\lambda}\right)\partial_{\omega}^{2}G\left(\lambda,\omega_{\lambda}\right))^{-1}\partial_{\omega}F\left(\lambda,\omega_{\lambda}\right).
$$
(33)

*Herein, F and G represent the outer and inner problems, respectively, and $\lambda$, $\omega$ are the corresponding outer and inner variables.*

Employing Lemma 4, we will demonstrate the proof for Theorem 4.

*Proof.* Grounded in the PDE associated with the inner problem solving, we obtain the following condition

$$
\mathbf{0} \in \nabla_{\beta_{\mathcal{A}}}\ell^{\diamond}(\hat{\beta}) + \nabla_{\beta_{\mathcal{A}}}\ell_{\mathcal{A}}^{+}(\hat{\beta}) + \partial\sum_{k=1}^{d}\sum_{j\in\mathcal{S}_{k,\mathcal{A}}}P_{\eta}\left(\left|\hat{\beta}_{j}\right|,\lambda_{k}\right),
$$
(34)

where $\partial$ signifies the subdifferential (*i.e.*, set of subgradients). Subsequently, by differentiating both sides, apply the chain rule, and invoke the implicit function theorem we derive

$$
\mathbf{0} \in \nabla_{\beta_{\mathcal{A}}}^{2}\ell^{\diamond}(\hat{\beta})\dfrac{d\hat{\beta}_{\mathcal{A}}}{d\lambda} + \nabla_{\beta_{\mathcal{A}}}^{2}\ell_{\mathcal{A}}^{+}(\hat{\beta})\dfrac{d\hat{\beta}_{\mathcal{A}}}{d\lambda} + \sum_{k=1}^{d}\left(\nabla_{\beta}\sum_{j\in\mathcal{S}_{k,\mathcal{A}}}P_{\eta}\left(\left|\hat{\beta}_{j}\right|,\lambda_{k}\right)\right) + \nabla_{\beta_{\mathcal{A}}}^{2}\sum_{k=1}^{d}\sum_{j\in\mathcal{S}_{k,\mathcal{A}}}P_{\eta}\left(\left|\hat{\beta}_{j}\right|,\lambda_{k}\right)\dfrac{d\hat{\beta}_{\mathcal{A}}}{d\lambda}.
$$
(35)

Upon extracting the factor *w.r.t.* $\frac{d\hat{\beta}_{\mathcal{A}}}{d\lambda}$, we deduce

$$\left[\nabla^2_{\beta_{\mathcal{A}}}\left(\ell^{\diamond}_{\mathcal{A}}(\hat{\beta}) + \ell^+_{\mathcal{A}}(\hat{\beta})\right) + \sum_{k=1}^{d}\left(\nabla^2_{\beta_{\mathcal{A}}}\sum_{j\in\mathcal{S}_{k,\mathcal{A}}}P_{\eta}\left(\left|\hat{\beta}_j\right|,\lambda_k\right)\right)\right]\frac{d\hat{\beta}_{\mathcal{A}}}{d\lambda} = -\sum_{k=1}^{d}\left(\nabla_{\beta}\sum_{j\in\mathcal{S}_{k,\mathcal{A}}}P_{\eta}\left(\left|\hat{\beta}_j\right|,\lambda_k\right)\right). \tag{36}$$

By reorganizing the above expression, we arrive at

$$\frac{d\hat{\beta}_{\mathcal{A}}}{d\lambda} = -\sum_{k=1}^{d}\left[\nabla^2_{\beta_{\mathcal{A}}}\left(\ell^{\diamond}_{\mathcal{A}}(\hat{\beta}) + \ell^+_{\mathcal{A}}(\hat{\beta})\right) + \sum_{k=1}^{d}\left(\nabla^2_{\beta_{\mathcal{A}}}\sum_{j\in\mathcal{S}_{k,\mathcal{A}}}P_{\eta}\left(\left|\hat{\beta}_j\right|,\lambda_k\right)\right)\right]^{-1}\cdot\nabla_{\beta_{\mathcal{A}}}\sum_{j\in\mathcal{S}_{k,\mathcal{A}}}P_{\eta}\left(\left|\hat{\beta}_j\right|,\lambda_k\right). \tag{37}$$

Moreover, Lemma 4 supplies the derivative as given in (38).

$$\begin{aligned}\nabla_{\lambda}\mathcal{L}_{val} &= \partial_{\lambda}\sum_{i=1}^{N}\ell_i\left(\tilde{y}_i, f\left(\hat{\beta}^{\top}\tilde{\mathbf{x}}_i\right)\right) + \nabla_{\beta}\sum_{i=1}^{N}\ell_i\left(\tilde{y}_i, f\left(\hat{\beta}^{\top}\tilde{\mathbf{x}}_i\right)\right)\cdot\frac{d\hat{\beta}}{d\lambda}\\ &= \sum_{i=1}^{N}\nabla_{\beta}\ell_i\left(\tilde{y}_i, f\left(\hat{\beta}^{\top}\tilde{\mathbf{x}}_i\right)\right)\cdot\frac{d\hat{\beta}}{d\lambda}\end{aligned} \tag{38}$$

$\square$

## B.5 PROOF OF THEOREM 5

**Lemma 5** (Implication of $\kappa$-strong Quasi-Convexity). *(c.f. (Vial, 1982)) By definition, a differentiable function $\mathcal{L}_{val}$ is strongly quasi-convex if the following inequality holds*

$$\mathcal{L}_{val}\left(\lambda^*\right) \geqslant \mathcal{L}_{val}\left(\lambda_x\right) + \nabla\mathcal{L}_{val}\left(\lambda_x\right)^{T}\left(\lambda^* - \lambda_x\right) + \frac{\kappa}{2}\|\lambda^* - \lambda_x\|^2 \tag{39}$$

*for some $\kappa > 0$ and all $\lambda_x \in \Lambda$.*

Subsequently, we provide our proof of Theorem 5.

*Proof.* Utilizing the property articulated in Lemma 5 and invoking the triangle inequality, we ascertain that

$$\begin{aligned}\mathcal{L}_{val} &\geqslant \sum_{i=1}^{N}\ell_i\left(\tilde{y}_i, f\left(\hat{\beta}^{\top}(\lambda^*)\tilde{\mathbf{x}}_i\right)\right)\\ &\geqslant \sum_{i=1}^{N}\ell_i\left(\tilde{y}_i, f\left(\hat{\beta}^{\top}(\lambda)\tilde{\mathbf{x}}_i\right)\right) + \left\langle\sum_{i=1}^{N}\nabla_{\beta}\ell_i\left(\tilde{y}_i, f\left(\hat{\beta}^{\top}(\lambda)\tilde{\mathbf{x}}_i\right)\right)\cdot\nabla_{\lambda}\hat{\beta}(\lambda), \lambda^* - \lambda\right\rangle + \frac{\kappa}{2}\|\lambda^* - \lambda\|^2\\ &= \sum_{i=1}^{N}\ell_i\left(\tilde{y}_i, f\left(\hat{\beta}^{\top}(\lambda)\tilde{\mathbf{x}}_i\right)\right) + \frac{\kappa}{2}\|\lambda - \lambda^*\|^2 + \left\|\sum_{i=1}^{N}\nabla_{\beta}\ell_i\left(\tilde{y}_i, f\left(\hat{\beta}^{\top}(\lambda)\tilde{\mathbf{x}}_i\right)\right)\cdot\nabla_{\lambda}\hat{\beta}(\lambda)\right\|\cdot\|\lambda^* - \lambda\|\cdot\cos\psi\\ &\geqslant \sum_{i=1}^{N}\ell_i\left(\tilde{y}_i, f\left(\hat{\beta}^{\top}(\lambda)\tilde{\mathbf{x}}_i\right)\right) - \left\|\sum_{i=1}^{N}\nabla_{\beta}\ell_i\left(\tilde{y}_i, f\left(\hat{\beta}^{\top}(\lambda)\tilde{\mathbf{x}}_i\right)\right)\cdot\nabla_{\lambda}\hat{\beta}(\lambda)\right\|\cdot\|\lambda^* - \lambda\| + \frac{\kappa}{2}\|\lambda^* - \lambda\|^2,\end{aligned} \tag{40}$$

where

$$\psi = \arg\cos\frac{\left\langle\sum_{i=1}^{N}\nabla_{\beta}\ell_i\left(\tilde{y}_i, f\left(\hat{\beta}^{\top}(\lambda)\tilde{\mathbf{x}}_i\right)\right)\cdot\nabla_{\lambda}\hat{\beta}(\lambda), \lambda^* - \lambda\right\rangle}{\left\|\sum_{i=1}^{N}\nabla_{\beta}\ell_i\left(\tilde{y}_i, f\left(\hat{\beta}^{\top}(\lambda)\tilde{\mathbf{x}}_i\right)\right)\cdot\nabla_{\lambda}\hat{\beta}(\lambda)\right\|\cdot\|\lambda^* - \lambda\|}. \tag{41}$$

Following the reorganization, we deduce

$$\frac{\kappa}{2}\|\lambda - \lambda^*\|^2 \leqslant \cdot\|\lambda^* - \lambda\|\cdot\left\|\sum_{i=1}^{N}\nabla_{\beta}\ell_i\left(\tilde{y}_i, f\left(\hat{\beta}^{\top}(\lambda)\tilde{\mathbf{x}}_i\right)\right)\cdot\nabla_{\lambda}\hat{\beta}(\lambda)\right\|. \tag{42}$$

For the case where $\lambda = \lambda^*$, the proof is self-evident. We now consider the scenario wherein $\lambda \neq \lambda^*$, thereby arriving at

$$\frac{\kappa}{2} \|\lambda - \lambda^*\| \leqslant \left\| \sum_{i=1}^N \nabla_\beta \ell_i \left( \tilde{y}_i, f\left(\hat{\beta}^\top(\lambda)\,\tilde{\mathbf{x}}_i\right)\right) \nabla_\lambda \hat{\beta}(\lambda) \right\|. \tag{43}$$

This in turn leads us to our ultimate conclusion, expressed as

$$\|\lambda - \lambda^*\| \leqslant 2\kappa^{-1} \left\| \sum_{i=1}^N \nabla_\beta \ell_i \left( \tilde{y}_i, f\left(\hat{\beta}^\top(\lambda)\,\tilde{\mathbf{x}}_i\right)\right) \nabla_\lambda \hat{\beta}(\lambda) \right\| \leqslant 2\epsilon\kappa^{-1}. \tag{44}$$

$\square$

## B.6 Proof of Theorem 6

*Proof.* We can complete this proof by contradiction. Suppose that $\lambda$ lies at the end point of one segment of a solution path, and the corresponding Jacobian $\mathcal{J}$ is full rank. Let $\Delta\beta = \hat{\beta}' - \hat{\beta}$ and $\Delta\lambda = \lambda' - \lambda$, where the elements in $\beta$ we considered are indexed by the activity set. Since $\hat{\beta}$ satisfies the KKT system of (2), we have

$$\nabla_\beta \ell^\diamond(\hat{\beta}) + \mu(t)\nabla_\beta \ell^+(\hat{\beta}) + \nu(s)\nabla_\beta \ell^-(\hat{\beta}) + \sum_{k=1}^d \sum_{j \in \mathcal{S}_k} \partial_\beta P_\eta \left( \left|\hat{\beta}_j\right|, \lambda_k \right) = \mathbf{0} \tag{45}$$

$$\partial_\lambda \sum_{i=1}^N \ell_i \left( \tilde{y}_i, f\left(\hat{\beta}^\top(\lambda)\,\tilde{\mathbf{x}}_i\right)\right) = \mathbf{0}.$$

From the above assumption, we know that $\lambda'$ is not on the solution path. This implies that there exists some $\hat{\beta}'$ such that $(\hat{\beta}', \lambda')$ satisfies the KKT system, where $\hat{\beta}'$ is different from $\hat{\beta}$. Then we have

$$\nabla_\beta \ell^\diamond(\hat{\beta}') + \mu(t)\nabla_\beta \ell^+(\hat{\beta}') + \nu(s)\nabla_\beta \ell^-(\hat{\beta}') + \sum_{k=1}^d \sum_{j \in \mathcal{S}_k} \partial_\beta P_\eta \left( \left|\hat{\beta}'_j\right|, \lambda'_k \right) = \mathbf{0} \tag{46}$$

$$\partial_\lambda \sum_{i=1}^N \ell_i \left( \tilde{y}_i, f\left(\hat{\beta}'^\top(\lambda')\,\tilde{\mathbf{x}}_i\right)\right) = \mathbf{0}.$$

Subtracting the two sets of KKT conditions, we obtain

$$\nabla_\beta \left( \ell^\diamond(\hat{\beta}') - \ell^\diamond(\hat{\beta}) \right) + \mu(t)\nabla_\beta \left( \ell^+(\hat{\beta}') - \ell^+(\hat{\beta}) \right)$$
$$+ \nu(s)\nabla_\beta \left( \ell^-(\hat{\beta}') - \ell^-(\hat{\beta}) \right) + \sum_{k=1}^d \sum_{j \in \mathcal{S}_k} \left[ \partial_\beta P_\eta \left( \left|\hat{\beta}'_j\right|, \lambda'_k \right) - \partial_\beta P_\eta \left( \left|\hat{\beta}_j\right|, \lambda_k \right) \right] = \mathbf{0}. \tag{47}$$

Note that $\hat{\beta}$ and $\hat{\beta}'$ have *different* active sets, which means that some of the gradient(s) above may not exist. Let $\alpha_i = \left| \frac{\partial \ell}{\partial \beta_i} \right|$ for $i \in \mathcal{P}$. Then we have

$$\mathbf{0} \in \nabla_\beta \left( \ell^\diamond(\hat{\beta}') - \ell^\diamond(\hat{\beta}) \right) + \mu(t)\nabla_\beta \left( \ell^+(\hat{\beta}') - \ell^+(\hat{\beta}) \right) + \nu(s)\nabla_\beta \left( \ell^-(\hat{\beta}') - \ell^-(\hat{\beta}) \right)$$

$$+ \sum_{i \in \mathcal{P}} \alpha_i \left( \text{sign}(\hat{\beta}'_i) - \text{sign}(\hat{\beta}_i) \right) + \sum_{k=1}^d \sum_{j \in \mathcal{S}_k} \left[ \partial_\beta P_\eta \left( \left|\hat{\beta}'_j\right|, \lambda'_k \right) - \partial_\beta P_\eta \left( \left|\hat{\beta}_j\right|, \lambda_k \right) \right] \tag{48}$$

$$+ \partial_\lambda \sum_{i=1}^N \left[ \ell_i \left( \tilde{y}_i, f\left(\hat{\beta}'^\top(\lambda')\,\tilde{\mathbf{x}}_i\right)\right) - \ell_i \left( \tilde{y}_i, f\left(\hat{\beta}^\top(\lambda)\,\tilde{\mathbf{x}}_i\right)\right) \right].$$

Without loss of generality, we set $t = \bar{t}$, $s = \bar{s}$ to simplify notations. Now we can convert the above system as

$$\mathcal{J} \cdot \begin{bmatrix} \Delta\hat{\beta}_\mathcal{P} \\ \Delta\lambda \end{bmatrix} = \mathbf{0}, \tag{49}$$

where $\mathcal{J}$ is the Jacobian matrix defined in the theorem statement. Since $\mathcal{J}$ is full rank by assumption, we have $\Delta\hat{\beta} = \mathbf{0}$ and $\Delta\lambda = \mathbf{0}$. This contradicts the assumption that $\hat{\beta}'$ and $\lambda'$ are different from $\hat{\beta}$ and $\lambda$, respectively. Therefore, the end point cannot have a full rank Jacobian matrix. Hence the theorem is proved.

**Geometric Intuition.** There also exists a geometric understanding towards Theorem 6. Consider a perturbation $(\Delta\hat{\beta}, \Delta\lambda)$ *w.r.t.* the variables $\hat{\beta}$ and $\lambda$. The perturbed optimality condition, linearized around $(\hat{\beta}, \lambda)$, can be written as

$$\left[ \partial_\beta^2 (\ell^\diamond + \ell^+) \cdot \Delta\hat{\beta} + \sum_{k=1}^d \partial_\beta^2 \sum_{j \in \mathcal{S}_k} P_\eta \left( \left| \hat{\beta}_j \right|, \lambda_k \right) \cdot \Delta\hat{\beta} \right] \{\mathcal{P}, g\} + \sum_{k=1}^d \partial_\lambda \sum_{j \in \mathcal{S}_k} P_\eta \left( \left| \hat{\beta}_j \right|, \lambda_k \right) \cdot \Delta\lambda = \mathbf{0},$$

$$\sum_{k=1}^d \partial_\beta \sum_{j \in \mathcal{S}_k} P_\eta \left( \left| \hat{\beta}_j \right|, \lambda_k \right) \Delta\hat{\beta} \{\mathcal{P}, g\} = \mathbf{0}, \quad \alpha = \max_i \left| \frac{\partial\ell}{\partial\beta_i} \right|, i \in \mathcal{P}.$$

$$(50)$$

If matrix $\mathcal{J}$ is full rank, then the only solution to the linearized optimality condition is the trivial solution $\begin{bmatrix} \Delta\hat{\beta}_\mathcal{P} \\ \Delta\lambda \end{bmatrix} = \mathbf{0}$, which implies that no nontrivial perturbations in the variables $\beta$ and $\lambda$ can satisfy the optimality conditions near $(\hat{\beta}, \lambda)$. Suppose $(\Delta\hat{\beta}, \Delta\lambda)$ span the null space of $\mathcal{J}$. Since the Jacobian $\mathcal{J}$ is non-degenerate, there must be a unique direction $(\Delta\hat{\beta}, \Delta\lambda)$ in the null space of $\mathcal{J}$. Taking a small step along this direction will not violate the KKT conditions given the active set, and thus, the path continues in that direction. Therefore, $(\hat{\beta}, \lambda)$ cannot be the end point of one segment of the KKT path in this case. □

### B.7 PROOF OF THEOREM 7

*Proof.* We commence by examining the case where an index $j'$ is appended to the set $\mathcal{A}$ at the point $(t, s)$ within the interval $[\underline{t}, \overline{t}] \times [\underline{s}, \overline{s}]$. Let $\hat{\beta}(t, s)$ represent the vector of solutions at $(t, s)$, and $\hat{\beta}_{\mathcal{A}(t,s)}$ denote the subvector pertinent to the active set $\mathcal{A}(t, s)$. By derivation, the PDE system illustrating the trajectory of $\hat{\beta}$ is given by

$$\widetilde{\mathcal{L}}\left( \hat{\beta}_{\mathcal{A}(t,s)}, \lambda \right) \cdot \begin{bmatrix} \dfrac{\partial\hat{\beta}_{\mathcal{A}}(t,s)}{\partial t} \\ \dfrac{\partial\hat{\beta}_{\mathcal{A}}(t,s)}{\partial s} \end{bmatrix} = - \begin{bmatrix} \dfrac{d\mu(t)}{dt} \cdot \nabla_{\beta_\mathcal{A}} \ell_{\mathcal{A}(t,s)}^+ (\hat{\beta}) \\ \dfrac{d\nu(s)}{ds} \cdot \nabla_{\beta_\mathcal{A}} \ell_{\mathcal{A}(t,s)}^- (\hat{\beta}) \end{bmatrix}, \quad (51)$$

where $\widetilde{\mathcal{L}}\left( \hat{\beta}_{\mathcal{A}(t,s)}, \lambda \right)$ is defined as per (3). Hence, for any small $\delta t > 0$ and $\delta s > 0$, there exists a $\delta\hat{\beta}$ such that

$$\hat{\beta}_{\mathcal{A}(t,s)} + \delta\hat{\beta} = \hat{\beta}_{\mathcal{A}(t+\delta t, s+\delta s)}. \quad (52)$$

On integrating (52) over the interval $[t, t + \delta t]$ and $[s, s + \delta s]$, we derive

$$\delta\hat{\beta} = - \int_t^{t+\delta t} \int_s^{s+\delta s} \widetilde{\mathcal{L}}^{-1}\left( \hat{\beta}_{\mathcal{A}(t',s')}, \lambda \right) \cdot \begin{bmatrix} \dfrac{d\mu(t')}{dt} \cdot \nabla_{\beta_\mathcal{A}} \ell_{\mathcal{A}(t',s')}^+ (\hat{\beta}) \\ \dfrac{d\nu(s')}{ds} \cdot \nabla_{\beta_\mathcal{A}} \ell_{\mathcal{A}(t',s')}^- (\hat{\beta}) \end{bmatrix} ds' dt'. \quad (53)$$

Upon synthesizing (51) and (53), it becomes evident that the change in $\hat{\beta}_{j'}$ is nonzero. Thus, we obtain

$$\hat{\beta}_{j'}(t, s) \neq 0 \quad \text{and} \quad \hat{\beta}_{j'}(t + \delta t, s + \delta s) \neq 0, \quad (54)$$

which signifies that $j'$ is included in $\mathcal{A}$ at $(t, s)$ and would remain active for a small $\delta t > 0$ and $\delta s > 0$. This further implies that $j'$ will not be excluded from $\mathcal{A}$ in the immediate next time of segmentation.

Utilizing a comparable line of reasoning, we assume the existence of a point $(t', s')$ in the interval $[\underline{t}, \overline{t}] \times [\underline{s}, \overline{s}]$ such that $j''$ is expelled from $\mathcal{A}$ at $(t', s')$. Given that $j''$ is inactive, the corresponding

entry of $\hat{\beta}_{\mathcal{A}(t',s')}$ becomes zero. Additionally, the coefficient $\omega_j$ associated with $\hat{\beta}_{j''}$ at $(t', s')$ must be less than 1. Otherwise, $j''$ would be active in the segment. Analogously to the prior case, we integrate (52) over the interval $[t', t' + \delta t']$ and $[s', s' + \delta s']$ as

$$\delta \hat{\beta}'' = -\int_{t'}^{t'+\delta t'} \int_{s'}^{s'+\delta s'} \widetilde{\mathcal{L}}^{-1}\left(\hat{\beta}_{\mathcal{A}(t'',s'')}, \lambda\right) \cdot \begin{bmatrix} \dfrac{d\mu(t'')}{dt} \cdot \nabla_{\beta_{\mathcal{A}}} \ell^+_{\mathcal{A}(t'',s'')}(\hat{\beta}) \\ \dfrac{d\nu(s'')}{ds} \cdot \nabla_{\beta_{\mathcal{A}}} \ell^-_{\mathcal{A}(t'',s'')}(\hat{\beta}) \end{bmatrix} ds'' dt''. \quad (55)$$

By combining (51) and (53), it can be observed that the change in $\beta_{j''}$ is zero. Thus, we derive

$$\hat{\beta}_{j''}(t', s') = 0 \quad \text{and} \quad \hat{\beta}_{j''}(t' + \delta t', s' + \delta s') = 0. \quad (56)$$

As $j''$ is excluded from $\mathcal{A}$ at $(t', s')$ and remains inactive for a small $\delta t' > 0$ and $\delta s' > 0$, it can be deduced that $j''$ will not be incorporated into $\mathcal{A}$ during the immediate next time of segmentation. $\quad\square$

## B.8   Proof of Theorem 8

*Proof.* Analogous to the analysis conducted in Le Thi et al. (2008), it can be deduced that $\hat{\beta}_i \neq \pm\infty$ and $\lambda_k \neq \pm\infty$. We define a compact set $\mathcal{T} \times \mathcal{S}$, where $\mathcal{T} = [\underline{t}, \overline{t}]$ and $\mathcal{S} = [\underline{s}, \overline{s}]$. As $t$ and $s$ increase within $(t, s) \in \mathcal{T} \times \mathcal{S}$, the active set $\mathcal{A}$ may alter. Employing Theorem 2 and Definition 2, it can be readily verified that there exist $t^{\max}, s^{\max}$ such that $t^{\max} - \underline{t} > 0$ and $s^{\max} - \underline{s} > 0$, where $t^{\max}$ and $s^{\max}$ denote the maximal $t$ and $s$ fulfilling Lemma 3 given the initial partition $\mathcal{A}$. Consequently, the first segment constitutes a nontrivial interval (*i.e.*, not merely a single point). In a similar vein, one can deduce the existence of a sequence of nontrivial interval(s) within $\mathcal{T} \times \mathcal{S}$. Based on the active set definition, each subinterval must correspond to a unique pair of partitions $\mathcal{A}$ and $\overline{\mathcal{A}}$. Invoking Theorem 7, we ascertain that the sequence of segments is finite, as the combinatorial numbers of $\mathcal{A}$ and $\overline{\mathcal{A}}$ are finite.

We now examine the outer problem. Analogously, as $\lambda$ varies, the path of $\hat{\beta}$ is also piecewise smooth, with each segment associated with a specific active set $\mathcal{A}$. Given that the number of potential combinations of the active set $\mathcal{A}$ is finite, a finite number of piecewise segments exist for the path *w.r.t.* $\lambda$. Hence, we have demonstrated that both the inner and outer problems can be solved in a finite number of steps, thus completing the proof. $\quad\square$

## B.9   Derivation of (4)

To integrate along the $\Omega$, we need to combine the first two equations in the PDE system as follows

$$\frac{d\mu(t)}{dt} \cdot \nabla_{\beta_{\mathcal{A}}} \ell^+_{\mathcal{A}}(\hat{\beta}) + \widetilde{\mathcal{L}}\left(\hat{\beta}, \lambda\right) \cdot \frac{\partial \hat{\beta}_{\mathcal{A}}(s,t)}{\partial t} + \frac{d\nu(s)}{ds} \cdot \nabla_{\beta_{\mathcal{A}}} \ell^-_{\mathcal{A}}(\hat{\beta}) + \widetilde{\mathcal{L}}\left(\hat{\beta}, \lambda\right) \cdot \frac{\partial \hat{\beta}_{\mathcal{A}}(s,t)}{\partial s}$$

$$= \widetilde{\mathcal{L}}\left(\hat{\beta}, \lambda\right) \cdot \left(\frac{\partial \hat{\beta}_{\mathcal{A}}(s,t)}{\partial s} + \frac{\partial \hat{\beta}_{\mathcal{A}}(s,t)}{\partial t}\right) + \frac{d\mu(t)}{dt} \cdot \nabla_{\beta_{\mathcal{A}}} \ell^+_{\mathcal{A}}(\hat{\beta}) + \frac{d\nu(s)}{ds} \cdot \nabla_{\beta_{\mathcal{A}}} \ell^-_{\mathcal{A}}(\hat{\beta}) \quad (57)$$

$$= \mathbf{0}.$$

Given the integration path $\Omega(s, t) = 0$, we calculate a vector line integral to obtain

$$\hat{\beta}\left(s_{k-1}, t_{k-1}\right) + \int_{\Omega} \frac{\partial \hat{\beta}_{\mathcal{A}}(s,t)}{\partial s} ds + \int_{\Omega} \frac{\partial \hat{\beta}_{\mathcal{A}}(s,t)}{\partial t} dt$$

$$= \hat{\beta}\left(s_{k-1}, t_{k-1}\right) - \int_{\Omega} \widetilde{\mathcal{L}}(\hat{\beta}, \lambda)^{-1} \cdot \frac{d\mu(t)}{dt} \nabla_{\beta_{\mathcal{A}}} \ell^+_{\mathcal{A}}(\hat{\beta}) dt - \int_{\Omega} \widetilde{\mathcal{L}}(\hat{\beta}, \lambda)^{-1} \cdot \frac{d\nu(s)}{ds} \nabla_{\beta_{\mathcal{A}}} \ell^-_{\mathcal{A}}(\hat{\beta}) ds$$

$$= \hat{\beta}\left(s_{k-1}, t_{k-1}\right) - \int_{(s,t) \in \Omega} \widetilde{\mathcal{L}}(\hat{\beta}, \lambda)^{-1} \left(\frac{d\mu(t)}{dt} \nabla_{\beta_{\mathcal{A}}} \ell^+_{\mathcal{A}}(\hat{\beta}) dt + \frac{d\nu(s)}{ds} \nabla_{\beta_{\mathcal{A}}} \ell^-_{\mathcal{A}}(\hat{\beta}) ds\right)$$

$$= \hat{\beta}\left(s_{k-1}, t_{k-1}\right) - \int_{(s,t) \in \Omega} \widetilde{\mathcal{L}}(\hat{\beta}, \lambda)^{-1} \left\langle \left[\frac{d\nu(s)}{ds} \nabla_{\beta_{\mathcal{A}}} \ell^-_{\mathcal{A}}(\hat{\beta}), \frac{d\mu(t)}{dt} \nabla_{\beta_{\mathcal{A}}} \ell^+_{\mathcal{A}}(\hat{\beta})\right], \left[ds, dt\right] \right\rangle, \quad (58)$$

which yields the final result (4) for each respective segment.

## C  FURTHER DISCUSSIONS

### C.1  MORE ON PRIOR WORK

#### C.1.1  LIMITATIONS OF STATE-OF-THE-ART

In this subsection, we present an analysis highlighting why prior art *cannot* be trivially extended to sparse GLMs as defined in objective (1). This discussion not only illuminates the connection between our theorem and pre-existing work, but also delves into the limitations of specific data-varying algorithms.

Incremental learning with $k$-Nearest Neighbors ($k$-NN) is derived from the inherent nature of the algorithm (Rodríguez et al., 2002). As an instance-based learning algorithm, $k$-NN retains the entire training dataset instead of parametric modeling. It can incorporate new observation(s) into the existing training set without necessitating a retraining process. This characteristic is due to the absence of internal parameters or states, *i.e.*, only the data itself matters. Analogously, removing observations involves merely deleting the corresponding data. The process for online logistic regression presented by Tsai et al. (2014) transforms the original objective into its dual problem, and outlines the boundaries to estimate solutions after one online round. Using this solution, they apply warm-start training to compute the definitive new solution. However, their primal and dual initial values in (12) are not adaptable to models exceeding simple $\ell_1$ and $\ell_2$ loss, as their dual form would invite added complexity. Therefore, we cannot derive a similar bound as presented in their Equations (17) to (19).

Adopting the procedure of online Lasso (Garrigues & Ghaoui, 2008), employing different losses would not yield a closed-form solution for $\theta_1$ and $w_2$ like their Equations (4) and (5), which are crucial for deriving the analytical updating rules using homotopy reformulation. Analogous constraints are also apparent in the methodologies of the generalized Lasso (Chen & Hero, 2012) (*i.e.*, their Equations (35)-(40)) (Hofleitner et al., 2013) (*i.e.*, their Lemma 1 and Lemma 2). The research concerning online group Lasso (Li & Gu, 2022) utilizes the finite difference and Taylor expansion. Nevertheless, in their Equations (5) and (6), other loss functions (*e.g.*, inverse Gaussian deviance) cannot independently separate $\Delta w^*$ and $\Delta \theta$ as per their derivation. Furthermore, the high-order residual of their regularization term converges to a constant and therefore, cannot be extended to broader situations as explored in our study.

Moreover, numerous studies on online (or incremental and decremental) SVMs (Laskov et al., 2006; Gu et al., 2014; 2015; 2018; Kashef, 2021) are conducted in the dual space as opposed to the original loss. During online adjustments, they typically maintain the KKT conditions of the various classes of support vectors, encompassing both relaxed adiabatic incremental adjustments and strict restoration adjustments. It is evident that the style of these works differs significantly from general sparse GLMs and thus cannot offer technical insight for the development of more generic online GLMs.

Within the context of adjusting the hyperparameter(s) in online scenarios, (Garrigues & Ghaoui, 2008; Yang et al., 2023) attempt to derive many rules for updating the regularization strength in each round. Nevertheless, these methods remain largely heuristic in nature, demonstrating a deficiency in identifying the locally or globally optimal $\lambda$ throughout the full breadth of the parameter space.

#### C.1.2  RELATIONSHIP WITH (CONVENTIONAL) ONLINE LEARNING

The term "no-regret" is used to describe online algorithms that have a property where, as the number of rounds or decisions grows, the average regret per round converges to zero (Zinkevich, 2003). Concerning the regret bound of SAGO framework, based on its definition $Regret = \sum_{\tau=1}^{T} \mathcal{F}_\tau(\hat{\beta}_\tau) - \min_{\beta \in \mathbb{R}^m} \sum_{\tau=1}^{T} \mathcal{F}_\tau(\beta)$ ($\tau$ is the round index and $\mathcal{F}$ denotes the whole objective (1)) and our Theorem 1 (*i.e.*, Equivalence Guarantee), we could simply find that the theoretical regret value is fixed at $\sum_{\tau=1}^{T} 0 = 0$.

Pursuing an alternate vein, drawing upon the framework established by (Zinkevich, 2003; Xiao, 2009; Duchi et al., 2011; Wang et al., 2018) in conventional online learning, numerous contemporary studies (Fan et al., 2018; Luo & Song, 2020; Yang et al., 2023) have ventured into facilitating online learning for sparse GLMs with regret bound. In other words, their algorithm fails to establish an equivalence guarantee similar to that of our Theorem 1.

A salient distinction between these works and ours is the access constraint placed on training data in their algorithms. Specifically, these algorithms are limited to accessing only the "current new" training data. Once a subset of the data has been utilized in a particular round, it becomes inaccessible in subsequent training rounds, due to the stipulated setting which forbids the retention of historical training data. In essence, they cannot store data in an offline manner. While several algorithms are permitted to retain summaries of past observations such as historical average gradients (Shalev-Shwartz et al., 2012; Hoi et al., 2021), theoretically, they still fall short of emulating the effectiveness of batch retraining. Nevertheless, these methods intrinsically offer approximate solutions. Their convergence to the stationarity point of a new or modified dataset remains non-deterministic. This underscores that the models derived from these methodologies are *not* congruent to those trained ab initio as in our `SAGO` approach. Additionally, they frequently exhibit an insufficiency in adaptively adjusting regularization hyperparameters.

### C.1.3 RELATIONSHIP WITH MACHINE UNLEARNING

Recently, concerns related to privacy in machine learning have gained pronounced emphasis. In this realm, machine unlearning (or data deletion) algorithms (Bourtoule et al., 2021; Gupta et al., 2021) exhibit certain parallels with our configuration as detailed in Section 2.1. Such algorithms endeavor to expunge the influence of specific data points from trained models, aiming for a more computationally efficient alternative than a complete model retraining. Fundamentally, these algorithms prioritize an approximative online removal strategy, utilizing the theoretical framework of differential privacy. This stands in contrast to our approach, which concurrently addresses both the addition and removal of samples. Nevertheless, our methodology holds potential to inspire future endeavors in that research.

### C.2 MORE ON ALGORITHM 1

**Solving Interval**  The length of the solving interval (path) for the inner problem has a bearing on different varieties of $\mu(t)$, $\nu(s)$. Remarkably, a significantly shorter interval results in a substantial reduction in the total query times (computing time) of a PDE solver. We will verify this property empirically in Appendix F.2. However, we underscore that an exceedingly short solving interval could potentially instigate numerical instability. Moreover, an overly complex form of parameterization would considerably augment the challenges in theoretical analysis.

**Line Search for Event Points**  To more efficiently identify the location (existence) of the subsequent event, a line search strategy that integrates binary splits with interpolation could also be employed. Starting at an event point, we utilize interpolation to estimate the solution $\hat{\beta}\left(s_{k+1}, t_{k+1}\right)$ for the end of next segment, assuming there are no nearby event points. If the interpolation proves unsuccessful, we either resort on bisection (Moré & Thuente, 1994) between the known maximal point (after an event occurrence) and minimal point (where no event has occurred), or use a step size three times the last one if we do not know any maximal point. Within the line search, if any probed point has already been computed, we directly use the cached solution.

**Generalized Thresholding Conditions**  In Section 3.3, we have discussed the application of the $\ell_1$ regularization term's thresholding condition. Indeed, this is a classic conclusion, but it might struggle with expressions of other more complex regularizers. Here, we point out how to establish a thresholding condition for all regularizers that meet our Assumption 1. For more general cases, you may find our previous Lemma 3 useful, as it is not based on any specific regularizer and provides a set of subgradient conditions for $\beta_j$. In practice, we only need to calculate (or just estimate) the value of $\omega_j$ according to Equation (25), and based on

$$\omega_j \in \begin{cases} \{-1\} & \text{if } \hat{\beta}_j < 0 \\ [-1, 1] & \text{if } \hat{\beta}_j = 0 \\ \{1\} & \text{if } \hat{\beta}_j > 0 \end{cases}, \tag{59}$$

we can determine the occurrence of event **E1** and **E2**. It is worth noting that if the computing architecture allows, each $\omega_j$ can be calculated in a parallel and vectorized manner.

**Remark 5.** *When computing the $\omega_j$ via (25), we should use the loss on the training set when computing the gradient flow for the inner problem, and use the loss on the validation set for the outer problem.*

**Regarding Computational Burdens**   In Section 4.2, we analyzed the algorithmic complexity of the `SAGO`. In the context of computational overhead, it is imperative to elucidate the following points

- Our Equ. (3), (5) and (6) give the explicit expression of gradient when solving PDEs & ODEs, which substantiates the symmetry of some (sub)matrices and leads the fundamental acceleration induced by Numpy & SciPy libraries. While the principal burden of solving lies in matrix inversion, how to utilize the symmetrical characteristic to accelerate inversion has been extensively well-investigated in community (Lim et al., 2014). Additionally, more libraries oriented towards parallel computation are emerging, such as the `pbdR` package in R.

- Assuming that the partition $\mathcal{A}$ is known, even the fastest (rate) second-order Newton's method needs to solve the similar linear system multiple times until final convergence. The efficiency of `SAGO` lies in the fact that no iterations are needed at the overwhelming majority values of and it adaptively chooses step sizes to capture all pertinent events across the intervals.

- The current state-of-the-art in numerical libraries for ODEs and PDEs offers commendable computational efficiency. Furthermore, there exists a relentless endeavor within the research community to unearth even more expeditious solution techniques, as exemplified by works such as Nagy et al. (2022).

**GLMs without the Sparsity**   Given $\mathcal{A}$ to represent the complete set of features within the dataset, our proposed `SAGO` approach can be seamlessly adapted for dense parameter estimation within generalized linear models. Interestingly, the imposition of sparsity often eases the process of theorem derivations, influencing steps within Algorithm 1. We elaborate `SAGO` in the context of GLMs without imposed sparsity in Algorithm 2.

---

**Algorithm 2** `SAGO` Algorithm (Dense Estimating)

---

**Require:** Initial solution $\hat{\beta}^{(0)}$, online rounds $T$, training sets $X^\diamond$, $X^+$, $X^-$ in each round, tolerance $\epsilon$ for $\lambda$, $max\_iter$.
 1: Initialize $\mathcal{A} \leftarrow [m]$.
 2: **for** $\tau = 1, \cdots, T$ **do**
 3:     $s \leftarrow \underline{s}, t \leftarrow \underline{t}$
 4:     **while** $s \leq \bar{s}, t \leq \bar{t}$ **do**
 5:         Solve the PDE system (3).     ▷ Inner-level problem
 6:         Update $\hat{\beta}^{(\tau)}_{\mathcal{A}}$ and $\ell_{i,\mathcal{A}}$.
 7:     **end while**
 8:     $iters \leftarrow 0$
 9:     **while** $\|\nabla_\lambda \mathcal{L}_{val}\| > \epsilon$ and $iters \leq max\_iter$ **do**
10:         Solve the ODE system (5).     ▷ Outer-level problem
11:         Compute $\nabla_\lambda \mathcal{L}_{val}$ via (6).
12:         Update $\hat{\beta}^{(\tau)}_{\mathcal{A}}$ and $\ell_{i,\mathcal{A}}$, and warm-start if necessary.
13:         $iters \leftarrow iters + 1$
14:     **end while**
15:     $\lambda^{(\tau)} \leftarrow \arg\min \nabla_\lambda \mathcal{L}_{val}$
16: **end for**
**Ensure:** $\{\hat{\beta}^{(\tau)}\}, \{\lambda^{(\tau)}\}$

---

### C.3   Algorithmic Drawbacks

In this subsection, we articulate the inherent limitations associated with the proposed `SAGO` algorithm. Currently, these primary drawbacks manifest in two specific areas.

**Memory Challenge.**   The foremost consideration is the storage challenge in the context of online applications. Our framework necessitates the storage of every observation present in both $X^\diamond$ and $X^+$. With the substantial accumulation of historical data or the influx of new data, the required memory space escalates significantly. That is to say, if the quantity of samples consistently increases in each

round without any corresponding reduction (*i.e.*, data deletion), the storage requirements of our algorithm will progressively grow until it surpasses the maximum available capacity. Contrastingly, certain existing online frameworks (Duchi et al., 2011; Yang et al., 2023) offer adaptability to online environments involving *infinite* rounds since they solely store the information extracted from the historical data.

However, within the perspective of information theory, the necessity to guarantee the first-order optimality condition remains a theoretical inevitability for nearly all exact online algorithms, with some SVMs being the exception.

**Dimensionality Challenge.** Another important aspect to note is the challenge posed by high-dimensional data. In scenarios where $|\mathcal{A}| \ggg n$, the running complexity becomes predominantly governed by $|\mathcal{A}|$ rather than the size $n$. Consequently, in cases of exceptionally high-dimensional data, where the order of magnitude of $|\mathcal{A}|$ exceeds that of $n$ (or $n_0 + (n_+ - n_-)$), the algorithm's performance advantage is likely to decline, or possibly underperform when compared to established batch algorithms.

To further improve the scalability of SAGO, we primarily need to reduce the computational burden associated with Hessian inversion. An intuitive approach might involve leveraging low-rank approximations or related methodologies. Another conceivable strategy could entail keeping the Hessian component of (3) and (5) fixed for many queries, modifying only the gradient part, and updating the Hessian intermittently. However, the clear disadvantage would be that the equivalence guarantee would no longer hold. We have not yet explored this avenue but believe it is worth pursuing.

### C.4 OPEN PROBLEMS

Despite the progress accomplished through this study, there remain a multitude of open-ended questions that can act as promising trajectories for future research. We give two examples here.

- In relation to Remark 2, we noted the possibility of path discontinuities arising in the context of non-convex penalties. Formally, let $\boldsymbol{e}_j$ denote a steering vector. Then there exists a $\lambda^{dis}$ such that

$$\lim_{\varepsilon \downarrow 0} \hat{\beta}\left(\lambda^{dis} + \varepsilon \boldsymbol{e}_j\right) \neq \lim_{\varepsilon \uparrow 0} \hat{\beta}\left(\lambda^{dis} + \varepsilon \boldsymbol{e}_j\right).$$

  This phenomenon leads to an interesting question: how can we calculate the value of $\hat{\beta}$ during discontinuities without the reliance on warm-start training? If we can establish equivalent updating rules, we can further improve efficiency for non-convex $P_\eta(\rho, \xi)$.

- Computing the gradient in deep learning architectures can prove to be highly computationally intensive. This leads us to the question of how we could effectively extend the proposed methodology SAGO to accommodate more intricate structures such as neural networks or other machine learning tasks like clustering algorithms? Additionally, exploring how SAGO could be adapted to the context of black-box models presents a significant challenge.

## D SENSITIVITY ANALYSIS

In this section, we turn our focus towards the stability analysis of the inner-level problem, acknowledging that the outer layer problem follows a comparable tract. To carry out a sensitivity analysis on the solution $\hat{\beta}(s,t)$ of the PDE system, we first introduce small perturbations to the independent variables $s$ and $t$. Let $\delta s$ and $\delta t$ represent these perturbations, and $\delta \hat{\beta}(s,t)$ is the corresponding perturbation in the dependent variable $\hat{\beta}(s,t)$. Nonetheless, without specific functional forms, a general sensitivity analysis may be quite abstract. We can now consider the Taylor expansion of the perturbed system around the point $(s,t)$, and we have

$$\begin{aligned} \hat{\beta}(s+\delta s, t+\delta t) &= \hat{\beta}(s,t) + \frac{\partial \hat{\beta}(s,t)}{\partial s}\delta s + \frac{\partial \hat{\beta}(s,t)}{\partial t}\delta t + \mathcal{O}(\delta s^2, \delta t^2) \\ &= \hat{\beta}(s,t) + \delta\hat{\beta}(s,t) + \mathcal{O}(\delta s^2, \delta t^2). \end{aligned} \tag{60}$$

By substituting this expansion into the original PDE system as per (3), we effectuate the linearization of the system concerning the introduced perturbations. This results in

$$\frac{d\mu(t)}{dt} \cdot \left[ \nabla_{\beta_{\mathcal{A}}} \ell_{\mathcal{A}}^+(\hat{\beta}) + \nabla_{\beta_{\mathcal{A}}}^2 \ell_{\mathcal{A}}^+(\hat{\beta}) \delta\hat{\beta} \right] + \left[ \widetilde{\mathcal{L}}(\hat{\beta}, \lambda) + \nabla \widetilde{\mathcal{L}}(\hat{\beta}, \lambda) \delta\hat{\beta} \right] \cdot \frac{\partial \hat{\beta}_{\mathcal{A}}(s, t)}{\partial t} = \mathbf{0}. \qquad (61)$$

Using the base solution

$$\frac{d\mu(t)}{dt} \cdot \nabla_{\beta_{\mathcal{A}}} \ell_{\mathcal{A}}^+(\hat{\beta}) + \widetilde{\mathcal{L}}\left( \hat{\beta}, \lambda \right) \cdot \frac{\partial \hat{\beta}_{\mathcal{A}}(s, t)}{\partial t} = \mathbf{0}, \qquad (62)$$

the first term inside the square brackets in (61) vanishes , leaving us with

$$\frac{d\mu(t)}{dt} \cdot \nabla_{\beta_{\mathcal{A}}}^2 \ell_{\mathcal{A}}^+(\hat{\beta}) \delta\hat{\beta} + \nabla \widetilde{\mathcal{L}}(\hat{\beta}, \lambda) \delta\hat{\beta} \cdot \frac{\partial \hat{\beta}_{\mathcal{A}}(s, t)}{\partial t} = \mathbf{0}. \qquad (63)$$

We can follow a similar process for the second equation. Upon linearizing, we obtain two linearized PDEs for the perturbation $\delta\hat{\beta}(s, t)$ as follows

$$\begin{cases} \dfrac{d\mu(t)}{dt} \cdot \nabla_{\beta_{\mathcal{A}}}^2 \ell_{\mathcal{A}}^+(\hat{\beta}) \delta\hat{\beta} + \nabla \widetilde{\mathcal{L}}(\hat{\beta}, \lambda) \delta\hat{\beta} \cdot \dfrac{\partial \hat{\beta}_{\mathcal{A}}(s, t)}{\partial t} = \mathbf{0} \\[2ex] \dfrac{d\nu(s)}{ds} \cdot \nabla_{\beta_{\mathcal{A}}}^2 \ell_{\mathcal{A}}^-(\hat{\beta}) \delta\hat{\beta} + \nabla \widetilde{\mathcal{L}}(\hat{\beta}, \lambda) \delta\hat{\beta} \cdot \dfrac{\partial \hat{\beta}_{\mathcal{A}}(s, t)}{\partial s} = \mathbf{0}. \end{cases} \qquad (64)$$

Investigating the eigenvalues of the Hessian matrices $\nabla_{\beta_{\mathcal{A}}}^2 \ell_{\mathcal{A}}^\diamond(\hat{\beta})$, $\nabla_{\beta_{\mathcal{A}}}^2 \ell_{\mathcal{A}}^+(\hat{\beta})$ and $\nabla_{\beta_{\mathcal{A}}}^2 \ell_{\mathcal{A}}^-(\hat{\beta})$ helps us inspect the stability of the solution $\hat{\beta}(s, t)$. In practical settings, it becomes imperative to utilize numerical methods for finding the eigenvalues and affirming the system's stability given the intricacies in analytically solving this eigenvalue problem for particular functional forms and parameter values. A solution exhibiting all positive eigenvalues can be deemed stable concerning the perturbations $\delta s$ and $\delta t$, while the presence of any negative eigenvalue can imply potential instability in the solution.

Table 3: The smallest matrix eigenvalue across 50 trials, with the loss function imposed on each dataset identical to that mentioned in Section 5.

| Dataset | creditcard | MiniBooNE | higgs | numerai28.6 | 2dplanes |
|---|---|---|---|---|---|
| $\nabla_{\beta_{\mathcal{A}}}^2 \ell_{\mathcal{A}}^\diamond$ | 452.7 | 2.3 | 466.1 | 2925.7 | 1484.1 |
| $\nabla_{\beta_{\mathcal{A}}}^2 \ell_{\mathcal{A}}^-$ | 57.9 | 8.0 | 945.6 | 2821.5 | 1534.9 |
| $\nabla_{\beta_{\mathcal{A}}}^2 \ell_{\mathcal{A}}^+$ | 38.8 | 1.7 | 307.5 | 1708.5 | 1526.6 |
| $\nabla_{\beta_{\mathcal{A}}}^2 P_\eta$ | 383.4 | 5.0 | 1828.4 | 1071.5 | 1517.9 |

| Dataset | ACSIncome | Buzzinsocialmedia | fried | OnlineNewsPopularity | house_16H |
|---|---|---|---|---|---|
| $\nabla_{\beta_{\mathcal{A}}}^2 \ell_{\mathcal{A}}^\diamond$ | 1313.9 | 0.0422 | 2785.6 | 0.360 | 5891.0 |
| $\nabla_{\beta_{\mathcal{A}}}^2 \ell_{\mathcal{A}}^-$ | 854.8 | 0.0230 | 3160.8 | 0.323 | 5032.7 |
| $\nabla_{\beta_{\mathcal{A}}}^2 \ell_{\mathcal{A}}^+$ | 827.3 | 0.677 | 2982.7 | 0.369 | 5343.1 |
| $\nabla_{\beta_{\mathcal{A}}}^2 P_\eta$ | 662.0 | 0.0826 | 3025.4 | 0.397 | 5240.7 |

Referring to the large amount of positive outcomes in Table 3, we can ascertain that the PDE system exhibits stability in simulations. This implies that small perturbations in the solution will decay over time and will not give rise to unbounded behavior. This inference is instrumental in understanding the robustness and reliability of the inner-level PDE system, along with its solutions in various online data-varying applications.

# E EXPERIMENT DETAILS

In this section, we provide missing details of the experimental settings used in Section 5.

**Data Preprocessing** For datasets regarding both Poisson and Gamma regressions, we take each component in label $y_i$ as its absolute value to ensure it's greater than 0. For $X^-$ (*i.e.*, deleted observations), we generate noises by turning normal samples into poisoning ones by flipping their labels (Frénay & Verleysen, 2014; Ghosh et al., 2017) for classification tasks. For regression tasks, noises are generated following a distribution analogous to the original training data, consistent with the methodology delineated in Jagielski et al. (2018).

**Infrastructure** All experiments presented in this study were conducted on a workstation running the Ubuntu $18.04$ operating system, equipped with Intel $2.30$GHz CPU$\times 200$ and $400.0$GB of RAM. It should be noted that we abstained from employing data-parallel training methodologies during performance speed comparisons.

**Baseline Details** We implement our SAGO algorithm in `Python 3.7`, wherein the numerical solvers exploit the Runge-Kutta method of order $4$ or $5$. We implement the batch training methods in Python, using the NumPy and SciPy libraries. The parameterizers are set to $\mu(t) = 4t^2$, $\nu(s) = \sqrt{1 - s^2}$, respectively. We approximately control each group to have the number of $round(0.1 \times d)$ features. The convergence tolerance $\varepsilon$ for batch training is 1e-7 and the tolerance $\epsilon$ for hyperparameter in the outer-level problem is 1e-4. For logistic loss, we adopted the line search algorithm presented in Beck & Teboulle (2009) to estimate the Lipschitz bound, with an initial guess given by the Frobenius norm of the data matrix. We didn't compare with the baselines for simple $\ell_2$-regularized cases, since their problem structure is less complex than ours. Meanwhile, they cannot be extended to other GLMs like SAGO did here.

**Results Details** The Figure 5, Figure 6, Figure 7, Figure 8 are plotted using Logistic group Lasso on the `higgs` dataset, while the Figure 4 (left) is plotted using sparse Poisson regression on the `ACSIncome` dataset and the Figure 4 (right) is plotted using group Lasso on the `creditcard` dataset. In Figure 7, we manually alter the optimality conditions as shown in our Lemma 3 to trigger the events **E1** or **E2**, so as to test with the varying number of event points.

### E.1 CODE

To ensure the replicability, Python codes corresponding to the pivotal components of the proposed algorithms are incorporated within the supplementary materials.

## F ADDITIONAL RESULTS

### F.1 ADDITIONAL HISTOGRAMS

The histogram of the number of event points when solving the GLM on more datasets are provided in Figure 9 and Figure 10. It should be emphasized that the total number of event points is intricately linked to the length of the solution interval. Consequently, our primary focus is on their distributional characteristics rather than their absolute count.

### F.2 ON THE INNER-LEVEL SOLVING INTERVAL

In order to investigate the total query times when solving the inner-level of SAGO, we adopt the different $\Omega(\cdot)$ and mark their running times as shown in Figure 11. When the interval length gets shorter, the total number of queries also gets smaller, resulting in a decreased total training time. The results of this experiment align with those delineated in Appendix C.2.

### F.3 PERFORMANCE UNDER VARYING LEVELS OF SPARSITY

To investigate the performance of SAGO with different scenarios of $\mathcal{A}$, we simulate the settings in Section 5 while varying the size of feature spaces. The results of accuracy performance as well as the running time are reported in Table 4 and Figure 12, respectively.

As the dimensionality (*i.e.*, the number of features) of observations increases, the efficiency benefit of our SAGO algorithm, as quantified by the performance gain (or time gap) among various methods,

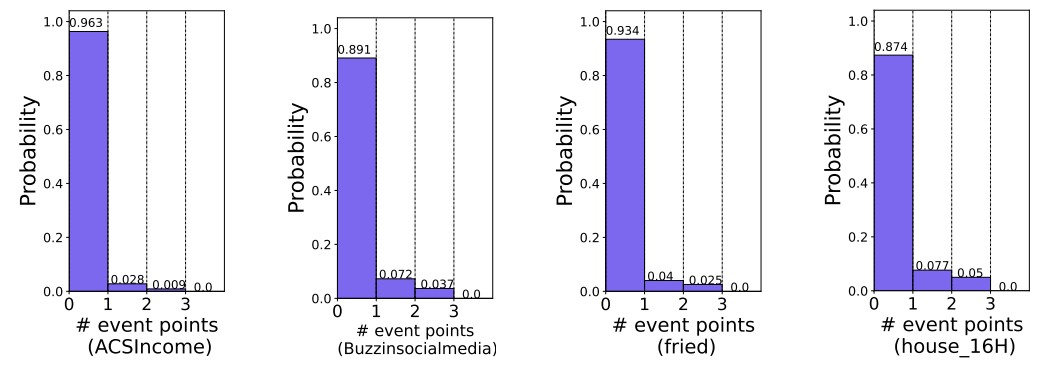

Figure 9: Histograms of the number of event points in *inner-level* problem when training the GLM (**i**) (*i.e.*, Sparse Poisson Regression). The name of the each dataset is enclosed in parentheses.

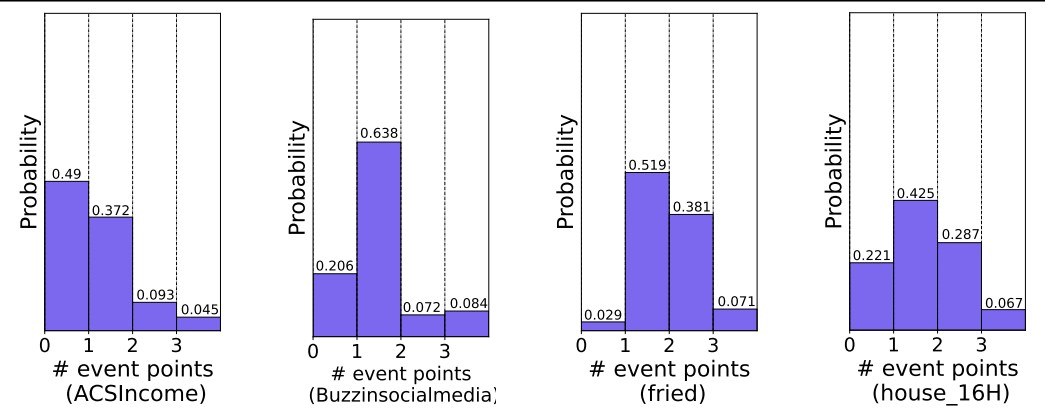

Figure 10: Histograms of the number of event points in *outer-level* problem when training the GLM (**i**) (*i.e.*, Sparse Poisson Regression). The name of the each dataset is enclosed in parentheses.

Table 4: Numerical results for validation loss *w.r.t* different levels of feature space. The best results are shown in bold. In simulations we use sparse Poisson regression (**i**) on `Buzzinsocialmedia` dataset and the number of observations in $X^+$ and $X^-$ is fixed within the trials in each column. In order to increase the quantity of features, we have employed polynomial feature transformation with a degree of 2, and randomly select a subset of $m$ features.

| Method | $m = 21$ | $m = 41$ | $m = 61$ | $m = 81$ | $m = 101$ |
|---|---|---|---|---|---|
| ColdGrid | $42.42 \pm 0.62$ | $24.92 \pm 0.25$ | $16.44 \pm 0.82$ | $12.80 \pm 0.38$ | $11.78 \pm 0.89$ |
| WarmGrid | $42.67 \pm 0.39$ | $24.76 \pm 0.58$ | $16.27 \pm 0.78$ | $12.68 \pm 0.78$ | $11.76 \pm 0.45$ |
| ColdHyBi | $41.83 \pm 0.27$ | $25.23 \pm 0.11$ | $\mathbf{16.04 \pm 0.49}$ | $12.81 \pm 0.48$ | $12.26 \pm 0.52$ |
| WarmHyBi | $41.68 \pm 0.15$ | $25.69 \pm 0.15$ | $16.28 \pm 0.39$ | $12.82 \pm 0.33$ | $12.09 \pm 0.34$ |
| ColdByO | $41.77 \pm 0.28$ | $24.58 \pm 0.27$ | $16.57 \pm 0.56$ | $12.69 \pm 0.45$ | $11.95 \pm 0.24$ |
| WarmByO | $41.44 \pm 0.16$ | $24.40 \pm 0.37$ | $16.56 \pm 0.43$ | $\mathbf{12.52 \pm 0.55}$ | $11.83 \pm 0.26$ |
| SAGO | $\mathbf{40.85 \pm 0.05}$ | $\mathbf{24.36 \pm 0.13}$ | $\mathbf{16.04 \pm 0.07}$ | $\mathbf{12.52 \pm 0.19}$ | $\mathbf{11.63 \pm 0.27}$ |

tends to decrease. On the other hand, the increase in the number of observations does not significantly influence this efficiency and complexity. Consequently, we can conclude that the `SAGO` algorithm exhibits superior performance in scenarios characterized by extensive sample sizes as opposed to those with high dimensional spaces.

## F.4 ON OTHER FORMS OF GLMS

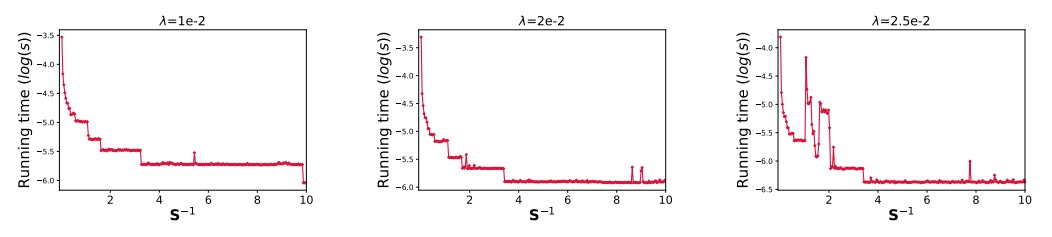

Figure 11: Evaluations for running time *w.r.t.* different $\mathbf{S}$, where $\mathbf{S} = \int_{\underline{s}}^{\bar{s}} \sqrt{1 + t^2(s)}\, ds$ (or equivalently, $\int_{\underline{t}}^{\bar{t}} \sqrt{1 + s^2(t)}\, dt$) is the arc length of $\Omega(\cdot)$. We use Gamma regression on *higgs* dataset and the $\lambda$ is fixed (shown in sugfigure's title) in the inner-level problem solving.

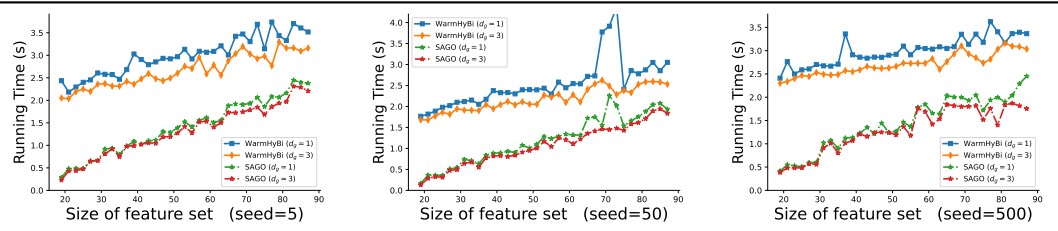

Figure 12: Comparisons of algorithmic efficiency. In experiments we keep the number of observations in $X^\diamond$, $X^+$ and $X^-$ as well as the random seed, while alternating the size of $m$. Other settings are the same in the Table 4.

### F.4.1  SPARSE GAMMA REGRESSION

The Gamma distribution is frequently used to model waiting times. For instance, in life testing, the waiting time until death is a random variable that is frequently modeled with a gamma distribution (Mittlböck & Heinzl, 2002). Similar to Poisson regression, we minimize the gamma deviance (Prentice, 1974) in regression as

$$\min_{\beta \in \mathbb{R}^m} \sum_i \left[ \log \frac{\exp\left(\beta^\top \mathbf{x}_i\right)}{y_i} + \frac{y_i}{\exp\left(\beta^\top \mathbf{x}_i\right)} - 1 \right] + \lambda \left\| \beta \right\|_1. \tag{65}$$

The numerical results obtained from applying `SAGO` to (65) are elucidated in Table 5. A comparative analysis of execution times against various baseline methods can be found in Figure 13.

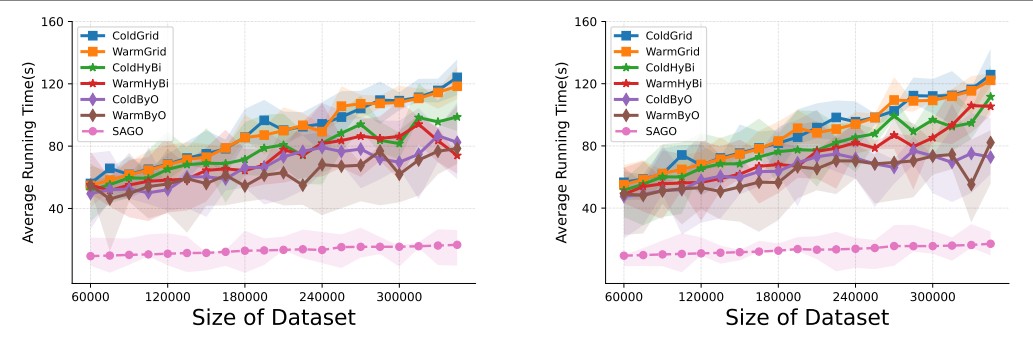

Figure 13: Comparative analysis of algorithmic efficiency. The subfigures on the left and right are sparse GLMs mentioned in Appendix F.4.1 and Appendix F.4.2, respectively. This figure is plotted using `ACSIncome` dataset.

Table 5: Numerical results for validation loss with standard deviation. The best results are shown in bold. We conduct comparative experiments in the same environment and setting as in the main paper.

| Method | ACSIncome | Buzzinsocialmedia | fried | house_16H |
|---|---|---|---|---|
| **Round #1** | | | | |
| ColdGrid | $296.0 \pm 0.45$ | $17.59 \pm 0.59$ | $9.260 \pm 0.52$ | $34.22 \pm 0.81$ |
| ColdHyBi | $295.0 \pm 0.31$ | $17.40 \pm 0.61$ | $9.489 \pm 0.27$ | $34.23 \pm 0.54$ |
| ColdByO | $295.3 \pm 0.34$ | $17.18 \pm 0.45$ | $9.224 \pm 0.34$ | $33.05 \pm 0.47$ |
| SAGO | $\mathbf{294.4 \pm 0.19}$ | $\mathbf{16.04 \pm 0.09}$ | $\mathbf{9.007 \pm 0.11}$ | $\mathbf{32.96 \pm 0.21}$ |
| **Round #2** | | | | |
| ColdGrid | $295.6 \pm 0.15$ | $18.55 \pm 0.68$ | $8.511 \pm 0.45$ | $34.23 \pm 0.85$ |
| ColdHyBi | $294.9 \pm 0.30$ | $16.98 \pm 0.86$ | $8.774 \pm 0.27$ | $\mathbf{33.66 \pm 0.23}$ |
| ColdByO | $295.7 \pm 0.18$ | $16.84 \pm 0.14$ | $8.974 \pm 0.79$ | $33.74 \pm 0.65$ |
| SAGO | $\mathbf{293.1 \pm 0.21}$ | $\mathbf{16.79 \pm 0.14}$ | $\mathbf{8.460 \pm 0.12}$ | $\mathbf{33.66 \pm 0.07}$ |
| **Round #3** | | | | |
| WarmGrid | $295.7 \pm 0.73$ | $17.55 \pm 0.57$ | $9.571 \pm 0.37$ | $34.92 \pm 0.61$ |
| WarmHyBi | $295.8 \pm 0.69$ | $18.86 \pm 0.56$ | $9.126 \pm 0.40$ | $35.96 \pm 0.94$ |
| WarmByO | $294.5 \pm 0.52$ | $17.51 \pm 0.49$ | $8.874 \pm 0.38$ | $34.98 \pm 0.44$ |
| SAGO | $\mathbf{293.7 \pm 0.12}$ | $\mathbf{16.26 \pm 0.17}$ | $\mathbf{8.843 \pm 0.14}$ | $\mathbf{34.86 \pm 0.09}$ |

### F.4.2 SPARSE INVERSE GAUSSIAN GLM

Inverse Gaussian GLM (Giner & Smyth, 2016) considers models for positive continuous data. Variables that take positive and continuous values often measure the amount of some physical quantity that is always present (Amin et al., 2016). We minimize the inverse Gaussian deviance as

$$\min_{\beta \in \mathbb{R}^m} \sum_i \frac{\left(y_i - \exp\left(\beta^\top \mathbf{x}_i\right)\right)^2}{y_i \exp\left(\beta^\top \mathbf{x}_i\right)} + \lambda \|\beta\|_1 . \tag{66}$$

The numerical results of SAGO on (66) is displayed in Table 6. A comparison of running times against various baselines can be found in Figure 13.

Table 6: Numerical results for validation loss with standard deviation. The best results are shown in bold. We conduct comparative experiments in the same environment and setting as in the main paper.

| Method | ACSIncome | Buzzinsocialmedia | fried | house_16H |
|---|---|---|---|---|
| **Round #1** | | | | |
| ColdGrid | $246.2 \pm 0.63$ | $37.79 \pm 0.33$ | $6.608 \pm 0.32$ | $21.22 \pm 0.88$ |
| ColdHyBi | $244.5 \pm 0.45$ | $36.70 \pm 0.29$ | $6.385 \pm 0.44$ | $19.87 \pm 0.76$ |
| ColdByO | $244.9 \pm 0.67$ | $37.03 \pm 0.88$ | $6.208 \pm 0.14$ | $19.74 \pm 0.12$ |
| SAGO | $\mathbf{243.1 \pm 0.59}$ | $\mathbf{36.07 \pm 0.17}$ | $\mathbf{6.176 \pm 0.12}$ | $\mathbf{19.73 \pm 0.05}$ |
| **Round #2** | | | | |
| ColdGrid | $246.8 \pm 0.76$ | $35.98 \pm 0.69$ | $6.994 \pm 0.92$ | $19.81 \pm 0.68$ |
| ColdHyBi | $247.5 \pm 0.38$ | $35.59 \pm 0.61$ | $6.779 \pm 0.72$ | $19.88 \pm 0.18$ |
| ColdByO | $246.7 \pm 0.47$ | $35.52 \pm 0.53$ | $6.860 \pm 0.74$ | $19.92 \pm 0.67$ |
| SAGO | $\mathbf{245.8 \pm 0.35}$ | $\mathbf{35.40 \pm 0.13}$ | $\mathbf{6.773 \pm 0.14}$ | $\mathbf{19.72 \pm 0.24}$ |
| **Round #3** | | | | |
| WarmGrid | $246.8 \pm 0.32$ | $36.13 \pm 0.31$ | $6.895 \pm 0.18$ | $20.74 \pm 0.57$ |
| WarmHyBi | $247.5 \pm 0.96$ | $36.51 \pm 0.57$ | $6.795 \pm 0.13$ | $22.68 \pm 0.35$ |
| WarmByO | $246.7 \pm 0.85$ | $\mathbf{35.60 \pm 0.79}$ | $6.878 \pm 0.53$ | $20.54 \pm 0.42$ |
| SAGO | $\mathbf{245.8 \pm 0.41}$ | $\mathbf{35.60 \pm 0.11}$ | $\mathbf{6.693 \pm 0.18}$ | $\mathbf{20.26 \pm 0.39}$ |

### F.5 ON THE NON-LINEAR PATTERN

As delineated in Remark 3, our approach is adept at handling (some) non-linear patterns. To illustrate it empirically, consider the utilization of random Fourier features.

Given any shift-invariant kernel like the Gaussian kernel, characterized by $k(x, y) = k(x - y)$, Rahimi & Recht (2007) introduced a comprehensive decomposition approach employing Fourier basis

functions $\left\{ \phi_\omega\left(x_i\right) = \sqrt{2}\cos\left(\omega^T x_i + b\right) \right\}_{\omega \in \mathbb{R}^d}$. This allows for the identification of its associated feature mapping. In practical terms, $\omega$ is sampled from $p(\omega)$ and $b$ is uniformly drawn from the interval $[0, 2\pi]$. Notably, the probability distribution $p(\omega)$, when affiliated with the Gaussian kernel, corresponds to a normal distribution, a connection further elaborated in Rahimi & Recht (2007).

In our experiment configuration, the parameter $\gamma$ of the RBF kernel is designated as 2, and the random seed is initialized to 100. We map the native input features into a feature space comprising 200 dimensions. The numerical results of SAGO is displayed in Table 7.

Table 7: Non-linear accuracy results for validation loss with standard deviation. The best results are shown in bold. We conduct comparative experiments in the same environment setting as in the main paper, as well as the GLMs for each specific dataset.

| Method | ACSIncome | Buzzinsocialmedia | fried | house_16H |
|---|---|---|---|---|
| **Round #1** | | | | |
| ColdGrid | $234.9 \pm 0.51$ | $46.55 \pm 0.68$ | $2.995 \pm 0.40$ | $24.71 \pm 0.76$ |
| ColdHyBi | $236.7 \pm 0.52$ | $48.04 \pm 0.86$ | $3.438 \pm 0.84$ | $24.93 \pm 0.62$ |
| ColdByO | $236.2 \pm 0.41$ | $45.79 \pm 0.96$ | $3.017 \pm 0.24$ | $24.77 \pm 0.19$ |
| SAGO | $\mathbf{234.8 \pm 0.27}$ | $\mathbf{44.28 \pm 0.14}$ | $\mathbf{2.728 \pm 0.13}$ | $\mathbf{24.59 \pm 0.17}$ |
| **Round #2** | | | | |
| ColdGrid | $237.0 \pm 0.53$ | $46.07 \pm 0.57$ | $3.046 \pm 0.72$ | $26.21 \pm 0.65$ |
| ColdHyBi | $236.9 \pm 0.63$ | $41.98 \pm 0.38$ | $3.175 \pm 0.58$ | $25.94 \pm 0.64$ |
| ColdByO | $235.7 \pm 0.69$ | $41.16 \pm 0.59$ | $\mathbf{2.892 \pm 0.02}$ | $26.09 \pm 0.89$ |
| SAGO | $\mathbf{235.3 \pm 0.18}$ | $\mathbf{40.35 \pm 0.21}$ | $\mathbf{2.892 \pm 0.12}$ | $\mathbf{25.53 \pm 0.27}$ |
| **Round #3** | | | | |
| WarmGrid | $233.3 \pm 0.70$ | $40.68 \pm 0.42$ | $3.419 \pm 0.69$ | $25.32 \pm 0.58$ |
| WarmHyBi | $237.7 \pm 0.96$ | $40.18 \pm 0.25$ | $3.227 \pm 0.40$ | $25.21 \pm 0.63$ |
| WarmByO | $235.3 \pm 0.82$ | $38.96 \pm 0.70$ | $2.567 \pm 0.62$ | $26.33 \pm 0.39$ |
| SAGO | $\mathbf{233.1 \pm 0.38}$ | $\mathbf{38.87 \pm 0.18}$ | $\mathbf{2.259 \pm 0.53}$ | $\mathbf{24.11 \pm 0.22}$ |

