# OpenReview forum: "Learning No-Regret Sparse Generalized Linear Models with Varying Observation(s)"
_ICLR.cc/2024/Conference — ICLR 2024 spotlight_

### Official Review · Reviewer_244k · 2023-10-22

**Soundness:** 4 excellent
**Presentation:** 2 fair
**Contribution:** 3 good
**Rating:** 8
**Confidence:** 3

**Summary:**

This paper analyzes learning a generalized linear model in online scenarios. The paper proposes an algorithm and analyzes the performance, path of optimizer and regret. Specifically, this paper analyzes the action of adding and eliminating some data by an ODE, and the based on the KKT condition, it proposes the path of optimizer given by the implementation of the proposed algorithm. At the end the empirical study validates the algorithm.

Update:

Thanks for the authors' comment. It makes sense to me, I raised the score to 8.

**Strengths:**

The algorithm is correct and concrete in maths, and the technique that treats the discrete actions of adding and removing data in a continuous space is normal and inspiring. It proposes a mathematical sound and practical algorithm and gives thorough analysis of the algorithm.

**Weaknesses:**

I do appreciate the discussion of the path of the optimizer in both inner loop and outer loop, but I'm still a little bit confused about how the notion of “no-regret” takes place, I would like to see more discussion about the final claim that can highlight the importance of the result.

**Questions:**

Lemma 3, eq 27, Why is there a minus sign in the second equation? However the proof below from eq 28 is correct.

In Assumption 1, shall the last assumption be there? If the function is symmetric about zero and has the first two derivatives, then the first derivative at zero must be 0, otherwise at the point 0 there is only subgradient but no gradient. However the examples, such as $\ell_1$ norm, or the regularizers that encourage sparsity, are all non-smooth at 0, or, if you take out the absolute value thus it is linear, then it is not symmetric about 0, but instead $f(x) = -f(-x)$ right?

Algo 1, line 5, what does “solve” mean? Solve the whole trajectory of $\hat \beta$?

Page 4 and later, I'm a bit confused about the set definitions and the elements in sets. For example, are $S_1,...,S_d$ elements of $S$, or $S = S_1 \cup ... \cup S_d$, is it consistent with the definition of $S_{\bar Z}$ which says $j\in S$?

"Formal statements ... in Appendix B.1", what if place the formal theorem and proof sketch in main body and the detailed proof in appendix?

Should Thm 2. be called continuity or Lipschitz?

---

> ### Author Response · Authors · 2023-11-15
> **Thank you for the in-depth review!**
>
> Dear reviewer 244k,
>
> We are very grateful for the effort you have put into reviewing our work and for recognizing our Contribution as "3 good" and Soundness as "4 excellent"! Your comments have been incredibly beneficial; we have corrected a notational issue and enhanced the discussion of the paper's contributions. **We believe all concerns and questions have been thoroughly addressed, and we are looking forward to further interaction with you.** We respectfully ask you to read our response and consider stronger support based on the contribution of the work. Thank you once again.
>
> ## Minor Notation Misuse
>
> > Page 4 and later, I'm a bit confused about the set definitions ..., are $ S_1, \ldots, S_d $ elements of $S $, or is $S = S_1 \cup \ldots \cup S_d $ consistent with the definition of $\mathcal{S}_{\bar{{Z}}}$ which says $ j \in S$?
>
> Thank you very much for pointing this out! We apologize for confusion caused. **Your first understanding is exactly what we intended in our paper.** Recall that we have defined $\mathcal{S}=\left\lbrace\mathcal{S}\_1, \mathcal{S}\_2, ..., \mathcal{S}\_k, \mathcal{S}\_{k+1}, ..., \mathcal{S}\_d\right\rbrace$, where $\mathcal{S}\_k \subset[m]$ and $m$ is the data dimensionality. We regard $\mathcal{S}_k$ as **an element** of the set $\mathcal{S}$, which conforms to our definition of the regularization term for GLMs (see (1) in page 3). This setting also facilitates subsequent derivations, avoiding unnecessary complexity and ambiguity. However, our definitions of $\mathcal{S}\_{\mathcal{Z}}$ and $\mathcal{S}\_{\bar{\mathcal{Z}}}$
> in Section 3 were indeed *not* rigorous. Specifically, **they should be written as**
>
> $$
> \begin{align*}
> &\mathcal{S}\_{\mathcal{Z}}(\beta)=\left\\{j \in \bigcup\_{k=1}^d\mathcal{S}\_k ~ |~\beta\_j=0\right\\}  \\\\
> &\mathcal{S}\_{\overline{\mathcal{Z}}}(\beta)=\left\\{j \in \bigcup\_{k=1}^d\mathcal{S}\_k ~ | ~ \beta\_j \neq 0\right\\} ~ (\text{or equivalently, ~ } \mathcal{S}\_{\overline{\mathcal{Z}}}(\beta)=\complement\_{\bigcup\_{k=1}^d\mathcal{S}\_k}^{\mathcal{S}\_{\mathcal{Z}}}).
> \end{align*}
> $$
>
> For consistency, we also modified the definition of $\mathcal{A}$ in the subsequent text (**the revised version has been uploaded**). The revised part is shown as follows:
>
> > Set ${\mathcal{S}\_\*}=\bigcup\_{k=1}^d\mathcal{S}\_k$. For a parameter vector $\beta \in \mathbb{R}^m$, we use $\mathcal{S}\_{\mathcal{Z}}(\beta)=\left\\{j \in \mathcal{S}\_\* ~ |~\beta\_j=0\right\\}$ to represent the set of penalized parameters that are zero
> > and correspondingly, $\mathcal{S}\_{\overline{\mathcal{Z}}}(\beta)=\left\\{j \in \mathcal{S}\_\* ~ | ~\beta\_j\neq 0\right\\}$ demarcates the set of penalized parameters with non-zero values.
>
> > We define the **active set** $\mathcal{A}={\mathcal{S}\_{\overline{\mathcal{Z}}} \cup \overline{\mathcal{S}\_\*}}$ indexes the current active predictors having either unpenalized or non-zero penalized coefficients.
>
> Thank you again and we will continue to check symbols. Please feel free to inform us if you have any further questions.
>
> ## Correctness in Theory
>
> > Lemma 3, eq 27, Why is there a minus sign in the second equation? However the proof below from eq 28 is correct.
>
> Thank you for raising a good question. At first **we verified the correctness of Equation (27)** in the proof process of Lemma 3. A minus sign appears multiple times in (27). Here's a step-by-step reasoning for the inclusion of the minus sign:
>
> 1. The **first** term $d_{-\boldsymbol{e}_j} \Theta(\beta)$ is the directional derivative, which is taken in the negative direction of the $j$-th coordinate axis. A minus sign before $e_j$ is to reflect this opposite direction.
>
> 2. Regarding the **second** minus sign, it arises from our calculations. According to the definition of the directional derivative, we obtain
> $$
> \begin{align*}
> &  d\_{-\boldsymbol{e}\_j} \Theta(\beta)  \\\\
> & = -\nabla\_j \left[\rule{0pt}{10pt}  \ell^{\diamond}\hspace{-2pt}\left(\rule{0pt}{9pt}\beta\right)+\mu\left(t\right)\ell^{\textbf{+}}\hspace{-2pt}\left(\rule{0pt}{9pt}\beta\right)+\nu\left( s\right) \ell^{\textbf{--}}\hspace{-2pt}\left(\rule{0pt}{9pt}\beta\right)\right]-(-1)\cdot\sum\_{k=1}^d\frac{\partial P\_\eta \left(\left\|{\beta}\_j\right\|, \lambda\_k\right)}{\partial\left\|\beta\_j\right\|} \\\\
> & =-\nabla\_j \left\[\rule{0pt}{10pt}  \ell^{\diamond}\hspace{-2pt}\left(\rule{0pt}{9pt}\beta\right)+\mu\left(t\right)\ell^{\textbf{+}}\hspace{-2pt}\left(\rule{0pt}{9pt}\beta\right)+\nu\left( s\right) \ell^{\textbf{--}}\hspace{-2pt}\left(\rule{0pt}{9pt}\beta\right)\right\]+\sum\_{k=1}^d\frac{\partial P\_\eta\left(\left|{\beta}\_j\right|, \lambda\_k\right)}{\partial\left|\beta\_j\right|}
> \end{align*}
> $$
>
> For a local minimum, both directional derivatives must be non-negative (the rationale could be found in **[1]**), which is equivalent to (25) with $\omega_j \in[-1,1]$. Thus, our proof should be correct.
>
> ------
>
> (We are running out of allowed characters, please see the second part [2/3])

---

> ### Author Response · Authors · 2023-11-15
> **Author Response [2/3]**
>
> > In Assumption 1, shall the last assumption be there?
>
> That's another constructive question. To construct a concise yet accurate Assumption 1, we went through countless rounds of trial and discussion before finalizing the current five assumptions. For the (***v***),
>
> 1. We **only assume the *existence*** of the first and second derivatives, not their continuity over the domain. That is, we distinguish between "**twice differentiable**" and "**twice continuously differentiable**" (*in the latter case, the first derivative at zero must be $0$*). In fact, we allow the derivatives to be piecewise continuous; take $\ell_1$ as an example, where 0 is their partition point. The non-differentiability at $0$ is one of the key statistical guarantees for sparse estimators to shrink component predictors to zero. Other regularizers (**like SCAD**) have additional manually defined partition points besides $0$.
>
> 2. We also assume that the values of the **derivatives are finite**, as this forms the basis of many analyses and derivations. In fact, this property is observed by almost all regularizers.
>
> > if you take out the absolute value thus it is linear, then it is not symmetric about 0, but instead $f(x)=-f(-x)$ right?
>
> **Yes**, you're correct in your observation. However, when discussing sparse estimation for GLMs, we (and other literature) generally do not take out the absolute value unless necessary.
>
> ## Notion of "no-regret"
>
> > I'm still a little bit confused about how the notion of “no-regret” takes place
>
> **TL;DR:** This is related to our core claim Theorem 1. It asserts that **SAGO maintains consistent accuracy with batch retraining methods** for varying observations, preserving GLM's generalization performance.
>
> ----
>
> The "***regret***" refers to the difference (gap) between the losses incurred by the algorithm in an online manner and the losses incurred by the best predicted action in hindsight [2], which is an important and widely used definition in the realm of online learning and repeated decision making. In our setting, consider a sequence of rounds indexed by $\tau=1,\ldots, T$. In each round $\tau$, we need to add new samples into the current dataset and delete certain selected samples from it (c.f., Section 1, 2). Since retraining the entire model from scratch is infeasible, we would resort to a "shortcut" method to update the model's solution $\hat{\beta}$ efficiently (a.k.a. *online updating*) [3]. This updated solution $\hat{\beta}^{\text{(ii)}}$ using SAGO is then compared to the `ground truth' optimal solution $\beta^*$ on the new (changed) dataset, and the gap between them measured by certain metrics like $\\|\hat{\beta}^{\text{(ii)}}-\beta^*\\|$ is the "regret" in each round.
>
> The term "***no-regret***" is used to describe online algorithms that have a property where, as the number of rounds or decisions grows, the average regret per round converges to zero [4]. Concerning the regret bound of SAGO framework, based on its definition ${Regret} = \sum\_{\tau=1}^{T} \mathcal{F}\_{\tau}(\hat{\beta}\_{\tau}) - \min_{\beta \in \mathbb{R}^m} \sum\_{\tau=1}^{T} \mathcal{F}\_{\tau}(\beta)$ ($\tau$ is the round index and $\mathcal{F}$ denotes the whole objective (1)) and our Theorem 1 (i.e.,  Equivalence Guarantee), we could simply find that the theoretical regret value is fixed at $\sum\_{\tau=1}^{T}0=0$. Therefore, we have highlighted our algorithm as no-regret within the Title and Section 1.
>
> While there may be *slight* terminological differences, especially for experts in the fields of GLMs or differential equations who might be less familiar with this notion, it is, however, a *well-entrenched* concept among researchers in online optimization and incremental (decremental) learning.
>
> ---
>
> >  I would like to see more discussion about the final claim that can highlight the importance of the result.
>
> Thank you for your valuable suggestion! We have made revisions to the paper based on your advice. All changes have been marked in red (**the revised version has been uploaded**). Specifically,
>
> 1. We have reemphasize this "***no-regret***" claim on **Page 2** (*Section 1*) and highlighted the innovative contributions of our work in this regard.
>
> 2. We have revisited the concept of *Regret* on **Page 5** (*Section 2.3*, following Theorem 1), pointing out that its value is fixed at zero using our algorithm.
>
> 3. We have discussed in detail on **Page 28** (*Appendix C.1.2*) why the regret would remain zero in each online round and how this reflects a departure from the traditional online framework.
>
> ----------
>
> (We are running out of allowed characters, please see the third part [3/3])

---

> ### Author Response · Authors · 2023-11-15
> **Author Response [3/3]**
>
> ## Remaining Questions
>
>
> > Algo 1, line 5, what does “solve” mean? Solve the whole trajectory of $\hat{\beta}$?
>
> Yes, your understanding aligns with our intent. The referenced step of our algorithm, as well as the step in Line 15, is to **numerically compute the trajectory of $\hat{\beta}$ in one segment** based on its dynamics. Upon reaching the terminal point of current segment, we proceed to update the active set as delineated in Lines 6 through 10. Subsequently, we solve the trajectory of $\hat{\beta}$ in the next segment, and continues until we have traversed the entire $[\underline{s}, \bar{s}]\times [\underline{t}, \bar{t}]$ (or $\Lambda$). We believe that with the further aid of Appendix C.2 and the provided code documentation, it would be clear to the readers.
>
> > "Formal statements ... in Appendix B.1", what if place the formal theorem and proof sketch in main body and the detailed proof in appendix?
>
> We are grateful for the suggestion to relocate them. We have considered including the complete Theorem 1 and its proof sketch in the main body, and concluded that it would introduce the use of **additional space**. This, in turn, would make us to condense other elements like key equations or experimental tables / figures **due to the page limit** (9 pages), potentially detracting from the overall readability of the paper. Furthermore, those seeking the comprehensive details of the statement and proof can be directed to the appendix, where the proof can be presented in full without interrupting the flow of the main body.
>
> > Should Thm 2. be called continuity or Lipschitz?
>
> We appreciate the reviewers' insightful query regarding the nomenclature. Our Theorem 2, titled "*Continuity Guarantee*", is indeed focused on establishing the continuous dependence of the solution $\hat{\beta}$ on the variables of the inner and outer problems. However, if the theorem specifies a "Lipschitz condition", which is **a stronger statement** that *not only* does $\hat{\beta}$ vary continuously *but also* does so within a bounded rate of change with respect to changes in $s, t$ and other parameters [5]. Hence, we are inclined to maintain the current designation, assuming it accurately reflects the theorem's statement. Nonetheless, we are open to further discussion and will revise the terminology if it enhances the accuracy of Theorem 2.
>
> ----
>
>
> **References**:
>
> [1] Rockafellar, R. Tyrrell. "Generalized directional derivatives and subgradients of nonconvex functions." Canadian Journal of Mathematics (1980): 257-280.
>
> [2] Zinkevich, Martin. "Online convex programming and generalized infinitesimal gradient ascent." ICML 2003.
>
> [3] Anderson, Terry, ed. "The theory and practice of online learning." Athabasca university press, 2008.
>
> [4] Gordon, Geoffrey J. "No-regret algorithms for online convex programs." NIPS 2006.
>
> [5] Hager, William W. "Lipschitz continuity for constrained processes." SIAM Journal on Control and Optimization (1979): 321-338.
>
> **We thank the Reviewer 244k again for your detailed feedback. Please feel free to let us know if you still have any remaining concerns or questions, we will be happy to address them.**

---

> ### Author Response · Authors · 2023-11-16
> **A Thank You Note**
>
> Dear reviewer 244k,
>
> We are glad that our responses have largely addressed your questions. Many thanks for your appreciation and constructive suggestions on our work!

---

### Official Review · Reviewer_MPDs · 2023-10-29

**Soundness:** 3 good
**Presentation:** 3 good
**Contribution:** 3 good
**Rating:** 6
**Confidence:** 3

**Summary:**

An online updating algorithm SAGO for generalized linear models based on solving ODE-PDE systems was proposed. It is proved that the algorithm has no regret. The merit of the SAGO is empirically tested  by comparing with ColdGrid (WarmGrid), ColdHyBi (WarmHyBi) and ColdByO (WarmByO). Improved accuracy was observed in numerical experiments.

**Strengths:**

+ Careful theoretical development

**Weaknesses:**

- The proposed algorithm seems to require high computational resources, e.g. memory, making it incompetent with existing algorithms
- Dimension of the datasets selected for empirical studies are relatively large, but are low in model dimension. It is unclear whether the proposed SAGO algorithm can handle modern datasets of interest

**Questions:**

1. Comparing with existing methods ColdGrid, ColdHyBi, ColdByO:

(a) It would be good to present the computational time SAGO and benchmark with the existing methods

(b) Comparing the active sets f => do the methods agree on which ones are active?

2. Minor: above Eq. (2), it should probably be $\Sigma_{\{i|({\bf x_i}, y_i) \in X^\diamond\}}$ in the definition of $\ell^{\diamond}$. Same applies to $\ell^{-}$ and $\ell^{+}$.

3. In Algorithm 1, line 16-19: the thresholding condition is based on Lemma 1, which seems to only apply on $\ell_1$ penalty. However, it is claimed that Algorithm 1 applies to penalties satisfying Assumption 1. There seems to be a gap and I wonder how the thresholding condition can be generalized to a wider class of penalty functions

4. It's unclear how a coefficient can become active from inactive in SAGO in Algorithm 1, i.e. E1 on line 18-19.

(a) The ODE in Eq. (5) describe the dynamics of coefficients in "active" set, so solving (5) doesn't seem to provide an update for the coefficients in inactive sets?

(b) When applying the thresholding condition, should one replace the $L$ in Lemma 1 by the validation loss $\mathcal L_{val}$? To make things concrete, probably it's better to repeat explicitly the thresholding condition in line 18-19.

5. Minor: line 25 of Algorithm 1: should probably be arg zero.

---

> ### Author Response · Authors · 2023-11-15
> **Thank you for the insightful review!**
>
> Dear reviewer MPDs,
>
> We deeply appreciate the effort you dedicated to reviewing and acknowledging the Soundness of our work as "3 good"! Your insightful feedback has led us to correct a notation error and make our steps description more clear. **We are confident that we have addressed all raised concerns and queries, and we eagerly anticipate further dialogue with you**. We respectfully ask you to read response and consider stronger support based on the soundness and contributions of the work. Our sincere thanks once again.
>
> ## Our Clarification
>
> Perhaps due to some uncontrollable factors, some key points of this work were unexpectedly missed in the presented review. Please kindly allow us to make a quick clarification at the outset:
>
> > 1.(a) present the computational time SAGO and benchmark with the existing methods
>
> **The training time comparisons are shown in Figure 4 and Figure 13**. We compared the efficiency of SAGO with ColdGrid (WarmGrid), ColdHyBi (WarmHyBi) and ColdByO (WarmByO) based on different GLMs including:
>
> 1. `Sparse Poisson Regression`  　　　 3. `Sparse Gamma Regression`
>
> 2. `Logistic Group Lasso`  　　　　　　4. `Inverse Gaussian GLM`
>
> All experiments are conducted $50$ trials and **demonstrate that SAGO achieves significant improvements *in terms of runtime***. We have not specified the hyperparameters in figures since they are automatically adjusted by the outer-level problem (Section 5 and Appendix F.4).
>
> > 1.(b) Comparing the active sets f => do the methods agree on which ones are active?
>
> We can confidently answer **Yes**. In our research, we focused solely on the transformation of existing GLMs from offline training to data-varying training without modifying the statistical properties (see, e.g., [1]) of the model itself.
> We would like to emphasize that our approach does not involve any approximation to the features or weights (***Theorem 1***). i.e., when the $\\{\lambda_k\\}$ is fixed, **the updated solution (including $\mathcal{A}$) using SAGO is exactly identical to the solution of offline batch retraining**.
>
> This consistency has been verified in the numerical results of *Tables 2, 4, 5, 6, and 7*, since any loss of the correct predictor by SAGO would result in a significant performance degradation. **We have revised our Appendix C.2 to highlight this property**.
>
> > 4.(a) solving (5) doesn't seem to provide an update for the coefficients in inactive sets?
>
> The coefficients in the inactive set $\mathcal{S}_{\mathcal{Z}}$ remain at $0$; we utilized the key property $\hat{\beta}\_j\equiv0 ~(j\in\mathcal{S}\_{\mathcal{Z}})$ in our derivations.
>
> Sparse components have no impact on the output of the results, which is why they have long been referred to as "inactive" in literatures (e.g., [2]).
>
> ## Addressing your Concern
>
> It seems you have raised 2 main concerns regarding our work. Our responses are as follows:
>
> > The proposed algorithm seems to require high computational resources, e.g. memory, making it incompetent with existing algorithms
>
> Thank you for your feedback; as pointed out in Appendix C.2, our algorithm indeed uses more memory than existing batch algorithms. This property actually offers more benefits than drawbacks. It also aligns with a principle in efficient ML: "**Space-time trade-off**" (i.e., solving a problem in less time by using more storage space). In detail,
>
> + Using additional storage in training GLMs can initially seem costly in terms of resources. However, this cost is offset by the gains in model performance as well as prediction accuracy, which could be verified by our theory and experiments. Nowadays, **faster training and prediction times lead to quicker improvements and iterations, which is crucial in a fast-paced research or industrial environment** [3].
>
> + Our SAGO requires storing the entire matrix $X$ and maintaining a matrix $\widetilde{\mathcal{L}}\left(\hat{\beta},\lambda\right)$ of size $\mathcal{O}\left( |\mathcal{A}|^2\right)$, but this is a necessary choice to enable faster computation by following the optimizer's dynamics. **It should be noted that in contemporary machine learning systems, the space-time trade-off is extremely common** [4], e.g., by using additional memory, ML algorithms can pre-compute values (e.g. Gram matrix in kernel), or parallelize tasks (e.g. DataParallel in PyTorch), which dramatically reduce the required computation times.
>
> + In practice, we found that for some GLMs, certain key submatrices within their PDE and ODE systems **exhibit good structural properties, such as *sparsity* and *symmetry***. These properties could potentially be leveraged to **further save on memory usage** in future research. Moreover, when maintaining or updating matrices of size $\mathcal{O}\left( |\mathcal{A}|^2\right)$, we can employ ***low-rank approximation***, ***incremental update formula*** [5] or ***low-precision training*** to further save memory.
>
>
> --------
>
> (We are running out of allowed characters, please see the second part [2/3])

---

> ### Author Response · Authors · 2023-11-15
> **Author Response [2/3]**
>
> ## Addressing your Concern (*continued*)
>
>
> > Dimension of the datasets selected for empirical studies are relatively large, but are low in model dimension. It is unclear whether the proposed SAGO algorithm can handle modern datasets of interest
>
> Thank you for your feedback. Regarding the practicality of our algorithm and the robustness of experiments, we would like to point out
>
> + It's important to emphasize that GLMs assume a linear relationship between output and input through a link function, so **they are generally not used for high-dimensional, extra large datasets** (e.g., ImageNet [6]).  These 'modern' datasets contain unstructured data, such as images, audio and text. In modern machine learning, we often use overparameterized deep neural networks to capture the complex non-linear relationships in data, which **often exceeds the representation ability of classic GLMs**. This should *not* be considered a shortcoming of our algorithm. Additionally, it should be noted that the complexity of our algorithm depends on the cardinality of the active set $\lVert\beta\rVert_0$, **not the original dimensionality** $m$ of the observations applied. By discarding features with high noise or irrelevance, it works on a relatively **small set of selected variables** (i.e., $\lVert\beta\rVert\_0\ll m$), which is a *meaningful reduction* from the original large set of variables.
>
> + **In experiments, we used real-world datasets from various fields suitable for GLMs, not just simple synthetic datasets**. For instance, the `numerai28.6` dataset in Table 1 consists of global stock training data that has been cleaned and standardized. New market indicators may update every few minutes or seconds. We fed these real-time data into our online model to obtain timely predictions, which can lead to better returns for investors [7]. Hence, the practicality of our algorithm can be comprehensively tested. We kindly ask you to read Section 5 (page 8) and Appendix E (page 32) for more insights and in-depth discussion.
>
> + **Practically, we often use existing dimensionality reduction techniques to reduce the model size before feeding the (transformed) dataset into a GLM** for further analysis [8]. Directly applying an appropriate GLM to high-dimensional space typically relies on *strong prior knowledge*, such as how to set the $\mathcal{S}_k$ in (1), how to divide group-wise sparsity in $G$, etc. Well-known dimensionality reduction methods include approximating high-dimensional Hilbert spaces using random Fourier features. For this, **we have conducted some numerical simulations in Table 7 (Appendix F.5), which you can kindly refer to.**
>
> + Our SAGO can handle variant observations but not variant model dimensions (i.e., the number of features). Our method can efficiently deal with a wide dynamic range of $n$, as used in our experiments from $10$ to $10^7$. However, it is not as flexible when dealing with model dimensions due to the fact that **in real-world applications, changes in samples are common, while applications with changes in dimensions are relatively rare** [9]. But this would be an important direction for our future exploration (please see Appendices C.3 and C.4 for details). Given this characteristic, **this is why we used datasets with large sample sizes but relatively small model dimensions** in our experiments.
>
>
>
> `Extra empirical evidence`:  ***To further reinforce empirical verification regarding the *active set* and *high-dimensional* model, we have attached some extra results on the side for your review.***
> Considering the time limit of this phase, we will incorporate more detailed results in the final version of our paper.
>
>
>
> ---
>
> ## Minor Notation Issue
>
> > above Eq. (2), it should probably be $\sum\_{i|(\mathbf{x}\_i,y\_i) \in X^\diamond} $ in the definition of $\ell^\diamond$.
>
> Thank you a lot for pointing this out! We apologize for the incorrect notation. To ensure the elements match, they indeed should be the pairs of $\mathbf{x}_i$ and $y_i$. **We have revised this part of the manuscript as:**
>
> > For simplicity, we denote $\ell^{\diamond}\left(\beta\right):={\sum\_{\left\lbrace \rule{0pt}{5.8pt}i|\left\lbrace \mathbf{x}\_i,y\_i\right\rbrace \in X^{\diamond} \right\rbrace}}\ell\_i\left(\rule{0pt}{10pt}y\_i, f\left(\beta^{\top} \mathbf{x}_i\right) \right)$, $ ~\ell^{\textbf{+}}\left(\beta\right):={\sum\_{\left\lbrace \rule{0pt}{5.8pt}i|\left\lbrace \mathbf{x}\_i,y\_i\right\rbrace\in X^+ \right\rbrace}}\ell\_i\left(\rule{0pt}{10pt}y\_i, f\left(\beta^{\top} \mathbf{x}\_i\right) \right)$ and similarly, $\ell^{\textbf{--}}\hspace{-2pt}\left(\beta\right)$.
>
> ***We will continue to check for similar problem. Thank you once again on this observation.***
>
> --------
>
> (We are running out of allowed characters, please see the last part [3/3])

---

> ### Author Response · Authors · 2023-11-15
> **Extra empirical evidence**
>
> ### Large model dimension
>
> We followed the experimental settings and data partitions in Table 7 (See Appendix F.5 on page 36) and then mapped the features into **a feature space of size** $m=2000$. The SAGO and baseline algorithms were run in this large dimension environment, with each test being conducted $10$ times. The results are shown in the table below. **The best results are highlighted in bold.**
>
>
>
> | **Method** | **ACSIncome** | **Buzzinsocialmedia** | **fried** | **house_16H** |
> |------------|---------------|-----------------------|-----------|---------------|
> | **Round #1** | | | | |
> | ColdGrid | 226.9 ± 0.29 | 36.58 ± 0.59 | 5.031 ± 0.13 | 23.74 ± 0.62 |
> | ColdHyBi | 228.7 ± 0.58 | 37.42 ± 0.83 | 5.061 ± 0.68 | 24.74 ± 0.51 |
> | ColdByO | 226.2 ± 0.51 | 36.86 ± 0.25 | 4.856 ± 0.91 | 23.93 ± 0.54 |
> | SAGO | **225.0 ± 0.68** | **35.56 ± 0.54** | **4.152 ± 0.09** | **23.44 ± 0.36** |
> | **Round #2** | | | | |
> | WarmGrid | 225.7 ± 0.35 | 37.42 ± 0.82 | 4.607± 0.81 | 23.76 ± 0.46 |
> | WarmHyBi | 226.5 ± 0.52 | 36.46 ± 0.96 | 4.462 ± 0.47 | 23.38 ± 0.17 |
> | WarmByO | 226.7 ± 0.78 | 36.91 ± 0.47 | 4.238 ± 0.16 | 23.50 ± 0.17 |
> | SAGO | **225.5 ± 0.90** | **36.29 ± 0.27** | **4.068 ± 0.11** | **23.20 ± 0.36** |
> | **Round #3** | | | | |
> | ColdGrid | 225.7 ± 0.75 | 34.24 ± 0.87 | 3.996 ± 0.61 | 21.36 ± 0.29 |
> | ColdHyBi | 227.6 ± 0.22 | 34.37 ± 0.63 | 4.486 ± 0.85 | 22.23 ± 0.66 |
> | ColdByO | 226.8 ± 0.39 | 35.45 ± 0.43 | 4.880 ± 0.53 | 22.89 ± 0.87 |
> | SAGO | **225.4 ± 0.72** | **34.09 ± 0.39** | **3.976 ± 0.25** | **21.12 ± 0.35** |
> | **Round #4** | | | | |
> | WarmGrid | 226.8 ± 0.67 | 35.00 ± 0.63 | 4.640 ± 0.78 | 23.57 ± 0.94 |
> | WarmHyBi | 229.4 ± 0.66 | 34.63 ± 0.28 | 4.872 ± 0.95 | 23.62 ± 0.05 |
> | WarmByO | 226.7 ± 0.44 | 34.93 ± 0.51 | 4.558 ± 0.57 | 22.97 ± 0.28 |
> | SAGO | **225.8 ± 0.92** | **34.61 ± 0.04** | **4.370 ± 0.05** | **22.84 ± 0.44** |
>
>
>
> -----
>
>
>
> ### Active set comparing
>
> We followed the experimental settings and data partitions as performed in Table 2, then used a list to **record the different indices in two active sets** compared to the batch retraining after each inner loop. The recorded results were returned and then printed. Each test was run $20$ times. The results are shown in the table below.
>
> | Dataset           | GLM Type | Regularization Strength | Returned List |
> |-------------------|----------|------------------------|---------------|
> | ACSIncome         | (i)      | 0.02                   | ∅             |
> | ACSIncome         | (i)      | 0.05                   | ∅             |
> | Buzzinsocialmedia | (i)      | 0.02                   | ∅             |
> | Buzzinsocialmedia | (i)      | 0.05                   | ∅             |
> | fried             | (i)      | 0.02                   | ∅             |
> | fried             | (i)      | 0.05                   | ∅             |
> | house_16H         | (i)      | 0.02                   | ∅             |
> | house_16H         | (i)      | 0.05                   | ∅             |
> | creditcard        | (ii)     | 0.02                   | ∅             |
> | creditcard        | (ii)     | 0.05                   | ∅             |
> | MiniBooNE         | (ii)     | 0.02                   | ∅             |
> | MiniBooNE         | (ii)     | 0.05                   | ∅             |
> | higgs             | (ii)     | 0.02                   | ∅             |
> | higgs             | (ii)     | 0.05                   | ∅             |
> | numerai28.6       | (ii)     | 0.02                   | ∅             |
> | numerai28.6       | (ii)     | 0.05                   | ∅             |
>
> The $\emptyset$ represents that a returned list `array([])` is empty, indicating that **our SAGO and the existing batch methods agree on which predictors are active.**

---

> ### Author Response · Authors · 2023-11-15
> **Author Response [3/3]**
>
> ## Thresholding Condition
>
> > 3. In Algorithm 1, line 16-19: There seems to be a gap and I wonder how the thresholding condition can be generalized to a wider class of penalty functions
>
> > It's unclear how a coefficient can become active from inactive in SAGO in Algorithm 1
>
> Thank you for your constructive comment! Indeed, we only mentioned the $\ell_1$ case in the main body because it represents a classical *statistical property* of Lasso estimator. Despite its simplicity, $\ell_1$ serves as an excellent heuristic familiar to many researchers in sparse regression, making it readily acceptable. For GLMs, one only needs to compute (or just estimate) the $\omega_j$ value corresponding to each $\hat{\beta}_j$ according to our Equation (25) (part of **Lemma 3 on Page 22**). Note that the (25), despite appearing complex, is actually very simple to solve since it is just a first-order linear **scalar equation** in terms of $\omega_j$. Then the relationship is quite apparent based on our Lemma 3:
>
> $$
> \omega_j \in
> \begin{cases}
> \{-1\} & \text{if } \hat{\beta}_j < 0 \\\\
> [-1, 1] & \text{if } \hat{\beta}_j = 0 \\\\
> \{1\} & \text{if } \hat{\beta}_j > 0
> \end{cases}
> $$
>
>  All the $\omega_j$ here could be computed in a vectorized parallel. To detect a transition from inactive to active for $\beta_j ~(j\in\overline{A})$, we can monitor the value of $\omega_j$ and add $j$ to the active set. This operation can be supported by **boundary conditions pre-set in the numerical solver**.
> *We address below the revisions we have made concerning this gap*, and thank you again for pointing this out.
>
> > 4.(b) probably it's better to repeat explicitly the thresholding condition in line 18-19.
>
> Thank you for your extremely valuable suggestion! We have made modifications to the main body, not within Algorithm 1, but rather where we first introduce this concept (i.e., Section 3.3). We also add a cross reference to direct readers to Appendix C.2 to see how to derive and use the thresholding conditions under more general GLMs. **All changes have been marked in red (the revised version has been uploaded)**. As we have analyzed before, the rule itself is not complex, but more explanation will make it easier for readers to understand.
>
> ## Remaining Questions
>
> > 4.(b) When applying the thresholding condition, should one replace the $L$ in Lemma 1 by the validation loss $\mathcal{L}_{val}$?
>
> Thank you again for your insightful question! The choice between using $\mathcal{L}\_{val}$ or $\ell^+$ ($\ell^-$) in the thresholding condition **depends on which loop of Algorithm 1 we are currently in**. The thresholding conditions only provide a *general guideline* and does not strictly dictate the use of $\mathcal{L}\_{val}$ or $\ell^+$. Specifically, in implementation, we should use the loss on the training set when computing the gradient flow for the inner problem, and use the loss on the validation set for the outer problem. In response to this query, **we have further expanded the description in paper** w.r.t. thresholding conditions to better guide other readers.
>
> > 5. Minor: line 25 of Algorithm 1: should probably be arg zero.
>
> We have checked Algorithm 1 based on your comments but did not find any errors. We guess that you might be suggesting to set the value of the **hypergradient to zero** on line 25, similar to that in gradient descent. This is a common technique in conventional bilevel optimization but cannot be assured here. The hyperparameter space $\Lambda$ is determined by the user's input, making it theoretically challenging to guarantee a (local) optimum point, i.e., $\mathbf{0}\in\left\lbrace\nabla_\lambda\mathcal{L}_{val}\left(\lambda \right)|\lambda \in\Lambda\right\rbrace $ **does not always hold**. If you could provide further hints or descriptions, we are always open to continuing the discussion.
>
> ----
>
> **References**:
>
> [1] Peng Zhao, et al. "On model selection consistency of Lasso." Journal of Machine Learning Research, 2006.
>
> [2] Kelner, Jonathan, et al. "Feature Adaptation for Sparse Linear Regression." NeurIPS 2023.
>
> [3] Brink, Henrik, et al. "Real-world machine learning." Simon and Schuster, 2016.
>
> [4] Romero, Francisco, et al. "INFaaS: Automated model-less inference serving." USENIX 2021.
>
> [5] Duff, Iain S., and Christof Vömel. "Incremental norm estimation for dense and sparse matrices." Numerical Mathematics (2002): 300-322.
>
> [6] Deng, Jia, et al. "Imagenet: A large-scale hierarchical image database." CVPR 2009.
>
> [7] Das, Sanjiv R. "The future of fintech." Financial Management (2019): 981-1007.
>
> [8] Lindsey, James K. "Applying generalized linear models." Springer Science, 2000.
>
> [9] Gomes, Heitor Murilo, et al. "Machine learning for streaming data: state of the art, challenges, and opportunities." ACM SIGKDD Newsletter 2019.
>
>
> **We thank the Reviewer MPDs again for your detailed feedback. Please feel free to let us know if you still have any remaining concerns or questions, we will be happy to address them.**

---

> ### Author Response · Authors · 2023-11-21
> **A Kind Follow-up  (2 days left in discussion)**
>
> Dear reviewer MPDs,
>
> Thanks again for your thoughtful review. We would like to know if our response addresses your questions and concerns. We deeply appreciate the opportunity to engage further if needed. Thank you!
>
> Kind regards,
>
> Authors of #448

---

> > ### Comment · Reviewer_MPDs · 2023-11-21
> > **Thank you for addressing my questions**
> >
> > My questions have largely been addressed, although I am still not sure about the "space-time trade off" part because the space limitation sometimes forms a hard threshold so an algorithm can't be run. Nonetheless, I consider this to be relatively minor. I would raise my score.

---

> > > ### Author Response · Authors · 2023-11-21
> > > **Many thanks for your supportive review**
> > >
> > > Dear reviewer MPDs,
> > >
> > > Thank you for upgrading your score and timely reply!
> > >
> > > Your comment is constructive and aligns well with our thinking. Indeed, the size of memory can become a hard-threshold for SAGO. Suppose methods such as incremental updating are used; they would save a lot of memory usage but may affect the algorithm's *memory locality*, as frequent access operations can impact the efficiency. For fair comparisons, we did not consider such techniques in experiments, but we believe this is a valuable and important direction for *future research* (Appendix C.2), especially in environments with constrained computational resource like edge devices, etc.

---

### Official Review · Reviewer_oT9i · 2023-11-03

**Soundness:** 3 good
**Presentation:** 4 excellent
**Contribution:** 4 excellent
**Rating:** 8
**Confidence:** 2

**Summary:**

In this work, authors study GLMs with varying observations. Specifically, they develop an algorithm that can train optimal GLMs by only considering newly added (and removed) data points. This alleviates the need to train the entire model from scratch by only focusing on varied observations. They provide theoretical guarantees for their claim and conduct empirical studies to validate their theoretical contributions. Their algorithm outperforms the baselines in the numerical experiments.

**Strengths:**

I am not an expert in this specific research area, and it's important to take my opinions into consideration with that in mind.

The paper is well-written and presented. The theoretical results seem sound and the approach seems novel. As I perceive it, the introduction of the proposed bilevel optimization, coupled with the utilization of ordinary differential equations (ODEs) and partial differential equations (PDEs) to address these problems, represents a novel and inventive approach. While I have not conducted an exhaustive review of the mathematical details presented in the supplementary material, the findings in the main paper hold promise.

**Weaknesses:**

Please see questions.

**Questions:**

How easy is it to optimally solve the PDE-ODE procedure in both inner and outer problems? Are there existing solvers that solve them optimally with provable convergence guarantees and what are their computational/sample complexities? These can be answered for the models used in empirical study, i.e., for Sparse Poisson Regression and Logistic Group Lasso.

---

> ### Author Response · Authors · 2023-11-15
> **Many thanks for appreciating our research!**
>
> Dear reviewer oT9i,
>
> We extend our sincere gratitude for dedicating your valuable time to reviewing our paper and for expressing your appreciation and support for our work! In the following, we will provide a comprehensive response to your review comments in the order they were presented.
>
> -----
>
> > I am not an expert in this specific research area
>
> The ICLR is an expansive and inclusive community, with participants from a wide range of backgrounds. We believe that our research could have chances to appeal researchers from various areas. While this may not be your primary area of expertise, your insights are exceedingly valuable. **Your comments reflect a precise understanding of the many key facets of our paper**, and for that, we are particularly grateful.
>
> > the utilization of ODEs and PDEs ... ,represents a novel and inventive approach.
>
> > the findings in the main paper hold promise.
>
> Thank you for acknowledging our contributions! Indeed, in recent years, differential equations (including PDEs and ODEs), have demonstrated significant utility in machine learning community. This is evidenced by applications in gradient flow methods (e.g., [1]), Neural ODEs (e.g., [2]), and widely-used generative models such as Score-Based SDEs (e.g., [3]). We hope our work can provide further insights and provoke thoughtful consideration in these domains.
>
> > How easy is it to optimally solve the PDE-ODE procedure in both inner and outer problems?
>
> The solving process is pretty straightforward, provided that the researcher has a foundational understanding of machine learning. **Even those who are new to this field would be able to independently solve both the inner and outer problems by following the main body and the appendix of our paper**. We have *explicitly* derived the gradient flows for the inner and outer problems based on our theoretical analysis, and these have been formulated into a *standard* separable form that is compatible with numerical ODE and PDE solvers.
>
> The implementation of SAGO only involves two steps. The first is to choose appropriate parameterizers $\mu(t)$, $\nu\left(s\right)$, and manually determine the endpoint values of the solving interval $\underline{t}, \bar{t}, \underline{s}, \bar{s}$, as per Definition 2. This does not have a standard selection, although our experimental choices might serve as a useful reference. The second step involves deriving the concrete PDE and ODE systems for your loss function and regularization in GLMs based on Theorems 3 and 4. This will include the computation of simple derivatives such as $\nabla_{\beta} \ell^{\textbf{+}}(\hat{\beta})$, $\nabla_{\beta}P_\eta$ and $\widetilde{\mathcal{L}}$. Subsequently, one can directly employ **existing solvers** for the computation. For more details, please refer to our answer to the next question.
>
>
> > Are there existing solvers that solve them optimally with provable convergence guarantees？
>
> That's a good question and our answer is definitely **Yes**! For example, `MATLAB` offers the `ode45` function, which is well-known for its robustness and efficiency in handling a variety of ODEs. MATLAB’s numerical resources also include PDE solvers [4] that have built-in features to alert users when certain events, such as the stopping rules (of event) outlined in our Algorithm 1, are triggered. These features are essential for ensuring that the algorithm terminates appropriately and provides optimal solutions within the defined criteria. Besides, the `deSolve` package [5] in `R` language is another powerful tool that can be employed for this purpose. Furthermore, Python’s `SciPy` library also contains integration and solving functions, like well-known `solve_ivp`, which is a flexible and powerful solver that can handle a wide range of problems with robust convergence properties. It allows for event handling, dense output, and has multiple solver options, making it suitable for our context.
>
> Each of these solvers comes with theoretical guarantees for convergence under certain conditions, which are usually well-documented in their literature. The choice of solver may depend on the specific characteristics of the PDE/ODEs derived from our theorems. Notably, **we have a similar discussion on page 29** (Appendix C.2) of our paper. If you are interested, we warmly invite you to explore this part further!
>
> -----
>
> (We are running out of allowed characters, please see the next part)

---

> ### Author Response · Authors · 2023-11-15
> **Author Response [2/2]**
>
> > what are their computational/sample complexities? These can be answered for the models used in empirical study, i.e., for Sparse Poisson Regression and Logistic Group Lasso.
>
> Thank you for your insightful question! We kindly draw your attention to the fact that in addition to the classic GLMs like Poisson Regression and Logistic Group Lasso, we also conducted numerical **experiments on Gamma Regression and Inverse-Gaussian GLM, detailed in Appendix F.4**. The computational efficiency and improved generalization capabilities of SAGO over existing approaches can be found in Tables 5, 6 and Figure 13 on pages 35, 36.
>
> Addressing your question, the sample or computational complexity of numerical solvers for PDEs and ODEs **can indeed differ significantly** based on the employed method and the particularities of the equations at hand. Specifically,
>
> - Factors such as the system's stiffness and the length (or area) of the solving interval, can play a crucial role [6]. In the implementation of SAGO, the **choice of parameterizers influences the length of the integration interval**, while the specifics of the **loss function $\ell_i$ and (regularizer $P_\eta$) affect which solver is appropriate**, making it quite challenging (and perhaps *not suitable*) to provide a unified complexity analysis.
>
> - It is noteworthy that the quest for improved numerical solvers for differential equations remains a significant research direction within applied mathematics and statistical learning, i.e., **the community is still actively pursuing faster DE solving methods** [7]. Currently, we are witnessing the advent of more libraries with **parallel computing** capabilities, such as the `pbdR` packages in `R`, which reflect this ongoing endeavor.
>
> For these reasons, we have refrained from generalizing the overall complexity in Section 4. However, as any PDE or ODE solver recurrently evaluates the explicit (partial) derivatives (i.e., $\dfrac{d \hat{\beta}\_\mathcal{A}}{d \lambda}$, $\dfrac{\partial \hat{\beta}_{\mathcal{A}}\hspace{0.5pt}(s,t)}{\partial t}$ and $\dfrac{\partial \hat{\beta}\_{\mathcal{A}}\hspace{0.5pt}(s,t)}{\partial s}$) as given in (3) and (5), **we do analyzed the "*query complexity*" of solvers to provide some insight. We invite you to consult the detailed discussion in Section 4.2**, or *page 8* of the main body, for further details.
>
> ----
>
> **References**:
>
> [1] Barrett, David GT, and Benoit Dherin. "Implicit gradient regularization." ICLR 2021.
>
> [2] Zhu, Aiqing, et al. "On numerical integration in neural ordinary differential equations." ICML 2022.
>
> [3] Jo, Jaehyeong, et al. "Score-based generative modeling of graphs via the system of stochastic differential equations."  ICML 2022.
>
> [4] www.mathworks.com/help/matlab/math/partial-differential-equations.html
>
> [5] Soetaert, Karline, et al. "Solving differential equations in R: package deSolve." Journal of statistical software (2010): 1-25.
>
> [6] Verwer, Jan G. "Explicit Runge-Kutta methods for parabolic partial differential equations." Applied Numerical Mathematics (1996): 359-379.
>
> [7] Chen, Yifan, et al. "Solving and learning nonlinear PDEs with Gaussian processes." Journal of Computational Physics (2021): 110668.
>
>
> **We thank the Reviewer oT9i again for your insightful feedback and strong support of our submission! If you have any remaining concerns or further inquiries, please do not hesitate to let us know.**

---

> ### Author Response · Authors · 2023-11-21
> **A Kind Follow-up  (2 days left in discussion)**
>
> Dear reviewer oT9i,
>
> We are sincerely grateful for your strong support towards this research. If you desire further information regarding any aspect of our paper after reading the responses, please do not hesitate to let us know. Thanks!
>
> Kind regards,
>
> Authors of #448

---

### Author Response · Authors · 2023-11-15
**Highlighting Our Contributions [2/2]**

5. **This research reveals the dynamics of optimization paths for GLMs under varying observations, which is of independent intellectual interest** and also bears significant implications for machine learning. Notably, Arun et al. [2] connected *optimization paths* (e.g., gradient descent and mirror descent) to *regularization paths* (trajectory) and suggested that path-based studies could lead to better optimization algorithms with improved generalization bounds in the future. The idea of combining differential equation systems with optimization paths adds a significant tool to the current research community's toolbox, and provides a bridging interface between machine learning and applied mathematics.

-----

6. We also conducted numerical evaluations in simulated data-varying environments. **Our evaluations indeed demonstrate zero regret (i.e., solution equivalence) and better hyperparameter selection**. Please refer to Section 5, Appendix F.4, and our response to Reviewer MPDs for more information.

-----

References:

[1] Freund, et al. "Incremental forward stagewise regression: Computational complexity and connections to lasso." ROKS 2013.

[2] Suggala, et al. "Connecting optimization and regularization paths." NIPS 2018.

---

### Author Response · Authors · 2023-11-15
**Highlighting Our Contributions [1/2]**

There seems to be some confusion among the reviewers regarding our contributions. We would like to highlight our main contributions in the following. *We would be very happy to engage with the reviewers if they have any doubt or confusion*.

---

1. **We are the first to achieve an online update of the GLM as observations varying without retraining from scratch**, with accuracy comparable to batch training algorithms like coordinate descent and SGD. This process is underpinned by rigorous theoretical analysis, demonstrating that the **PDE-ODE process output is theoretically equivalent to that of batch methods**, converging to a first-order stationary point on the updated / changed dataset. Hence, our algorithm is no-regret, with zero regret value at each online round. Please see our response to Reviewer 244k, and for further details, refer to Section 2.3 and Appendix B.1. Traditionally, the only way to maintain generalization capabilities when adding or removing samples has been limited to retraining from scratch on whole dataset. Our contribution thus **addresses a long-standing gap in the field**, significant for both the GLMs and optimization communities.

-----

2. **Our algorithm is applicable to any sparse GLM, provided mild assumptions are met** (see Assumptions 1 and 2). This general applicability was previously unattainable in literature, which focused on specific data-varying GLMs. For example, Freund et al. [1] provided theoretical insights for the $\ell_1$-regularized quadratic loss (Lasso), i.e., if $\|\|\beta\|\|_1 \leq \delta$,

$$
\begin{align*}
& j_k \in \arg\max_{j \in \{1,\ldots,p\}} |(r^k)^T X_j|, ~~\bar{\alpha}_k \in (0,1), ~r^0 = \mathbf{y}, ~\beta^0 = \mathbf{0},  \\\\
& \beta^{k+1} \leftarrow (1 - \bar{\alpha}\_k)\beta^k + \bar{\alpha}\_k \delta \text{sgn}((r^k)^T X\_{j\_k})e\_{j\_k},  \\\\
&r^{k+1} \leftarrow (1 - \bar{\alpha}\_k)r^k - \bar{\alpha}\_k \delta\text{sgn}((r^k)^T X\_{j\_k})X\_{j\_k},
\end{align*}
$$

but benefits to researchers using other forms of GLMs were pretty limited, since **formulating similar rules for more complicated GLMs could be theoretically challenging**. Our method requires only weak assumptions for GLMs, enabling learning on varying observations for a range of significant and classic models like **Poisson Regression, Gamma Regression, and Logistic Group Lasso for the first time**. We offer a unified framework for this pivotal domain of GLMs, which is indeed one of our primary contributions.

-----

3. **Our framework is straightforward to implement and computationally efficient**. With theoretical analysis, we present explicit expressions for the gradient flows of both inner and outer problems, **homogenized and aligned with standard forms used by mainstream numerical libraries**, simplifying programming efforts. We pointed out the ease of algorithmic implementation in our response to Reviewer oT9i (also see Appendix C.2). Computationally, our solver adaptively selects step sizes within the solution interval to capture all events $E1$ / $E2$ and can swiftly adjust the active set $\mathcal{A}$ through preset thresholding condition. This feature is wll-supported by many solver libraries. Our algorithm requires no extensive iterations for reparameterization of the samples (in inner problem) or for traversing regularization parameters (in outer problem), contrasting sharply with conventional batch methods and search techniques. Our analysis also indicates that SAGO can **converge to first-order optimality conditions within a finite number of steps, avoiding oscillations around the optimum like SGD**.

-----

4. **Our Algorithm 1 adaptively adjusts regularization strength in each online round, which is of independent interest in the hyperparameter optimization field**. It is well-known that as observations varying, previously set hyperparameters may no longer be effective, so immediate adjustment of regularization parameter to optimal level provides the best statistical guarantees (see section 3.2). Even in *offline* (i.e., fixed-sample) scenarios, our algorithm can efficiently search the entire hyperparameter space for GLMs (not just for specific linear models) and optimizes its selection based on computed solution spectrum. **To the best of our knowledge, no current exploration for GLMs exists that does this** (random searches / Bayesian optimization can only evaluate a finite number of $\lambda$ values, not the entire space), making this work of broader interest.

---

### Author Response · Authors · 2023-11-15
**Gratitude to All the Reviewers**

Dear Reviewers,

Thanks for your time and constructive feedback on our work. *All the reviewers have agreed that the paper is well-written and technically sound.* We would like to highlight that the reviewers acknowledged several other strengths of our work.

**Reviewer oT9i:**

1. The paper is well-written and presented. The theoretical results seem sound and the approach seems novel;

2. the introduction of the proposed bilevel optimization, coupled with the utilization of ODEs and PDEs to address these problems, represents a novel and inventive approach;

3. the findings in the main paper hold promise.



**Reviewer MPDs:**

1. The merit of the SAGO is empirically tested; Improved accuracy was observed in numerical experiments;

2. Careful theoretical development.


**Reviewer 244k:**

1. The algorithm is correct and concrete in maths, and the technique that treats the discrete actions of adding and removing data in a continuous space is normal and inspiring;

2. It proposes a mathematical sound and practical algorithm and gives thorough analysis of the algorithm;

3. I do appreciate the discussion of the path of the optimizer in both inner loop and outer loop.


Meanwhile you raised some comments. In our detailed response below, we believe that we have fully answered them. We have modified the paper in light of these comments, and **all the changes are marked in red**. The major changes are summarized in the following:

* (*Reviewer MPDs*) **We have corrected the notations in $\ell^{\diamond}\left(\beta\right)$ and $\ell^{\textbf{+}}\left(\beta\right)$ on Page 4.**

* (*Reviewer MPDs*) **We have elaborated more on the usage of generalized thresholding conditions (Page 6,29) and have made remarks on the computational burden and the selection of loss functions (Page 29,30).**

* (*Reviewer 244k*) **We have corrected the notations in $\mathcal{S}\_{\mathcal{Z}}$ and $\mathcal{S}\_{\bar{\mathcal{Z}}}$ on Page 5.**

* (*Reviewer 244k*) **We have discussed the concept of no-regret in detail to emphasize our novelty and contributions (Page 2,5,28) and have added more explanations of the "solve" operation in Algorithm 1 (Page 30).**

We also kindly request you to consider stronger support for the paper if your concerns have been addressed. Please feel free to let us know if you still have any remaining concern, we will be happy to address them.

Thanks for your time and support!

Authors of \#448

---

### Meta-Review · Area_Chair_gZVk · 2023-12-05

**Metareview:**

This paper introduces an algorithm for learning Generalized Linear Models (GLMs) that can efficiently train optimal models with changing observations in an online (sequential) manner, without the need to retrain from scratch. Most existing approaches for sparse GLMs focus on offline batch updates, neglecting online solutions. The proposed algorithm can handle both the addition and removal of observations simultaneously and adaptively updates data-dependent regularization parameters for better statistical performance. The approach is based on bilevel optimization and explicit gradient flow, and it provides theoretical guarantees of consistency and finite convergence, with promising results demonstrated on real-world benchmarks.

Five reviewers have reviewed the paper, and their overall assessment of the paper was positive. I agree with the reviewers and believe that the paper proposes a novel approach for an important problem, with interesting and exciting new results. The paper is particularly very well written.

**Justification For Why Not Higher Score:**

While the paper is amazing, I believe it falls short of being ground-breaking. Therefore, I did not recommend Accept (Oral).

**Justification For Why Not Lower Score:**

The paper is well above the acceptance threshold for ICLR. Therefore, it deserves an Spotlight.

---

### Decision · Program_Chairs · 2024-01-16

Accept (spotlight)